# EF21-P and Friends: Improved Theoretical Communication Complexity for Distributed Optimization with Bidirectional Compression

## Abstract

The starting point of this paper is the discovery of a novel and simple error-feedback mechanism, which we call EF21-P, for dealing with the error introduced by a contractive compressor. Unlike all prior works on error feedback, where compression and correction operate in the *dual* space of gradients, our mechanism operates in the *primal* space of models. While we believe that EF21-P may be of interest in many situations where it is often advantageous to perform model perturbation prior to the computation of the gradient (e.g., randomized smoothing and generalization), in this work we focus our attention on its use as a key building block in the design of communication-efficient distributed optimization methods supporting *bidirectional* compression. In particular, we employ EF21-P as the mechanism for compressing and subsequently error-correcting the model broadcast by the server to the workers. By combining EF21-P with suitable methods performing worker-to-server compression, we obtain novel methods supporting bidirectional compression and enjoying new state-of-the-art theoretical communication complexity for convex and nonconvex problems. For example, our bounds are the first that manage to decouple the variance/error coming from the workers-to-server and server-to-workers compression, transforming a multiplicative dependence to an additive one. In the convex regime, we obtain the first bounds that match the theoretical communication complexity of gradient descent. Even in this convex regime, our algorithms work with biased gradient estimators, which is non-standard and requires new proof techniques that may be of independent interest. Finally, our theoretical results are corroborated through suitable experiments.

## 1 Introduction: Error Feedback in the Primal Space

The key moment which ultimately enabled the main results of this paper was our discovery of a new and simple error-feedback technique, which we call EF21-P, that operates in the primal space of the iterates/models instead of the prevalent approach to error-feedback (Stich & Karimireddy, 2019; Karimireddy et al., 2019; Gorbunov et al., 2020b; Beznosikov et al., 2020; Richtárik et al., 2021) which operates in the dual space of gradients[1]. To describe EF21-P, consider solving the optimization problem

$$\min_{x \in \mathbb{R}^d} f(x), \tag{1}$$

where $f : \mathbb{R}^d \to \mathbb{R}$ is a smooth but not necessarily convex function. Given a contractive compression operator $\mathcal{C} : \mathbb{R}^d \to \mathbb{R}^d$, i.e., a (possibly) randomized mapping satisfying the inequality

$$\mathrm{E}\left[\|\mathcal{C}(x) - x\|^2\right] \le (1 - \alpha)\|x\|^2, \qquad \forall x \in \mathbb{R}^d \tag{2}$$

for some constant $\alpha \in (0, 1]$, our EF21-P method aims to solve (1) via the iterative process

$$\begin{aligned} x^{t+1} &= x^t - \gamma \nabla f(w^t), \\ w^{t+1} &= w^t + \mathcal{C}^t(x^{t+1} - w^t), \end{aligned} \tag{3}$$

---

[1] Our method is inspired by the recently proposed error-feedback mechanism, EF21, of Richtárik et al. (2021), which compresses the dual vectors, i.e., the gradients. EF21 is currently the state-of-the-art error feedback mechanism in terms of its theoretical properties and practical performance (Fatkhullin et al., 2021). If we wish to explicitly highlight its dual nature, we could instead meaningfully call their method EF21-D.

where $\gamma > 0$ is a stepsize, $x^0 \in \mathbb{R}^d$ is the initial iterate, $w^0 = x^0 \in \mathbb{R}^d$ is the initial iterate *shift*, and $\mathcal{C}^t$ is an instantiation of a randomized contractive compressor satisfying (2) sampled at time $t$. Note that when $\mathcal{C}$ is the identity mapping ($\alpha = 1$), then $w^t = x^t$ for all $t$, and hence EF21-P reduces to vanilla gradient descent (GD). Otherwise, EF21-P is a new optimization method. Note that $\{x^t\}$ iteration of EF21-P can be equivalently written in the form of perturbed gradient descent

$$x^{t+1} = x^t - \gamma \nabla f(x^t + \zeta^t), \qquad \zeta^t = \mathcal{C}^{t-1}(x^t - w^{t-1}) - (x^t - w^{t-1}).$$

Note that the model perturbation $\zeta^t$ is not a zero mean random variable[2], and that in view of (2), the size of the perturbation can be bounded via

$$\mathrm{E}\left[\left\|\zeta^t\right\|^2 \mid x^t, w^{t-1}\right] \le (1 - \alpha)\left\|x^t - w^{t-1}\right\|^2. \tag{4}$$

From now on, we will write $\mathcal{C} \in \mathbb{B}(\alpha)$ to mean that $\mathcal{C}$ is a compressor satisfying (2).

### 1.1 EF21-P THEORY

If $f$ is $L$-smooth and $\mu$-strongly convex, we prove that both $x^t$ and $w^t$ converge to $x^* = \arg\min f$ at a linear rate, in $\mathcal{O}((L/\alpha\mu)\log 1/\varepsilon)$ iterations in expectation (see Section D). Intuitively speaking, this happens because the error-feedback mechanism embedded in EF21-P makes sure that the quantity on the right-hand side of (4) converges to zero, which forces the size of the error $\zeta^t$ caused by the perturbation to converge to zero as well. However, EF21-P can be analyzed in the smooth nonconvex regime as well, in which case it finds an $\varepsilon$-approximate stationary point. The precise convergence result, proof, as well as an extension that allows to replace $\nabla f(w^t)$ with a stochastic gradient under the general ABC inequality introduced by Khaled & Richtárik (2020) (which provably holds for various sources of stochasticity, including subsampling and gradient compression) can be found in Section E.

### 1.2 SUMMARY OF CONTRIBUTIONS

We believe that EF21-P and its analysis could be useful in various optimization and machine learning contexts in which some kind of iterate perturbation plays an important role, including randomized smoothing (Duchi et al., 2012), perturbed SGD (Vardhan & Stich, 2022), and generalization (Orvieto et al., 2022). In this work we do not venture into these potential application areas and instead focus all our attention on a single and important use case where, as we found out, EF21-P leads to new state-of-the-art methods and theory: the design of communication-efficient distributed optimization methods supporting *bidirectional* (i.e., workers-to-server *and* server-to-workers) compression.

In particular, we use EF21-P as the mechanism for compressing and subsequently error-correcting the model broadcast by the server to the workers. By combining EF21-P with suitable methods ("friends" in the title of the paper) performing worker-to-server compression, in particular, DIANA (Mishchenko et al., 2019; Horváth et al., 2022) or DCGD (Alistarh et al., 2017; Khirirat et al., 2018), we obtain novel methods, suggestively named EF21-P + DIANA (Algorithm 1) and EF21-P + DCGD (Algorithm 2), both supporting bidirectional compression, and both enjoying new state-of-the-art theoretical communication complexity for convex and nonconvex problems. While DIANA and DCGD were not designed to work with compressors from $\mathbb{B}(\alpha)$ to compress the workers-to-server communication, and can in principle diverge if used that way, they work well with the smaller class of randomized compression mappings $\mathcal{C} : \mathbb{R}^d \to \mathbb{R}^d$ characterized by

$$\mathrm{E}[\mathcal{C}(x)] = x, \qquad \mathrm{E}\left[\|\mathcal{C}(x) - x\|^2\right] \le \omega \|x\|^2, \qquad \forall x \in \mathbb{R}^d, \tag{5}$$

where $\omega \ge 0$ is a constant. We will write $\mathcal{C} \in \mathbb{U}(\omega)$ to mean that $\mathcal{C}$ satisfies (5). It is well known that if $\mathcal{C} \in \mathbb{U}(\omega)$, then $\mathcal{C}/(\omega+1) \in \mathbb{B}(1/(\omega+1))$, which means that the class $\mathbb{U}(\omega)$ is indeed more narrow.

⋄ **Convex setting.** EF21-P + DIANA provides new state-of-the-art convergence rates for distributed optimization tasks in the strongly convex (see Table 1) and general convex regimes. This is the first method enabling bidirectional compression that has the server-to-workers and workers-to-server communication complexity better than vanilla GD. When the workers calculate stochastic gradients (see Section 3.1), we prove that EF21-P + DIANA improves the rates of the existing methods.

---

[2]In fact, this only happens in the non-interesting case when $\mathcal{C}^{t-1}$ is identity with probability 1.

We prove that EF21-P + DCGD has an even better convergence rate than EF21-P + DIANA in the interpolation regime (see Section 3.2).

⋄ **Nonconvex setting.** In the nonconvex setting (see Section 4), EF21-P + DCGD is the first method using bidirectional compression whose convergence rate decouples the noises coming from the workers-to-server and server-to-workers compression from multiplicative to additive dependence (see Table 2). Moreover, EF21-P + DCGD provides the new state-of-the-art convergence rate in the low accuracy regimes ($\varepsilon$ is small or the # of workers $n$ is large). Further, we provide examples of optimization problems where EF21-P + DCGD outperforms previous state-of-the-art methods even in the high accuracy regime.

⋄ **Unified SGD analysis framework with the EF21-P mechanism.** Khaled & Richtárik (2020) provide a unified framework for the analysis of SGD-type methods for smooth nonconvex problems. Their framework helps to analyze SGD and DCGD under various assumptions, including i) strong and weak growth, ii) samplings strategies, e.g., importance sampling. Unfortunately, the theory relies heavily on the unbiasedness of stochastic gradients, as a result of which it is not applicable to our methods (in EF21-P + DCGD, $\mathrm{E}\left[g^t\right] = \nabla f(w^t) \neq \nabla f(x^t)$). Therefore, we decided to rebuild the theory from scratch. Our results inherit all previous achievements of (Khaled & Richtárik, 2020), and further generalize the unified framework to make it suitable for optimization methods where the iterates are perturbed using the EF21-P mechanism. We believe that this is a contribution with potential applications beyond the focus of this work (distributed optimization with bidirectional compression). This development is presented in Section E; our main results from Section 4.1–4.3 which cater to the nonconvex setting are simple corollaries of our general theory.

## 2 DISTRIBUTED OPTIMIZATION AND BIDIRECTIONAL COMPRESSION

In this paper, we consider distributed optimization problems in strongly convex, convex and non-convex settings. Such problems arise in federated learning (Konečný et al., 2016; McMahan et al., 2017) and in deep learning (Ramesh et al., 2021). In federated learning, a large number of workers/devices/nodes contain local data and communicate with a parameter-server that performs optimization of a function in a distributed fashion (Ramaswamy et al., 2019). Due to privacy concerns and the potentially large number of workers, the communication between the workers and the server is a bottleneck and requires specialized algorithms capable of reducing the communication overhead. Popular algorithms dealing with these kinds of problems are based on communication compression (Mishchenko et al., 2019; Richtárik et al., 2021; Tang et al., 2019). We consider the distributed optimization problem of the form

$$\min_{x \in \mathbb{R}^d} \left\{ f(x) := \frac{1}{n} \sum_{i=1}^{n} f_i(x) \right\}, \tag{6}$$

where $n$ is the number of workers and $f_i : \mathbb{R}^d \to \mathbb{R}$ are smooth (possibly nonconvex) functions for all $i \in [n] := \{1, \ldots, n\}$. We assume that the functions $f_i$ are stored on $n$ workers. Each of them is directly connected to a server that orchestrates the work of the devices (Kairouz et al., 2021), i.e., the workers perform some calculations and send the results to the server, after which the server does calculations and sends the results back to the workers and the whole process repeats.

Throughput the work we will refer to a subset of these assumptions:

**Assumption 2.1.** The function $f$ is $L$–smooth, i.e., $\|\nabla f(x) - \nabla f(y)\| \leq L \|x - y\| \; \forall x, y \in \mathbb{R}^d$.

**Assumption 2.2.** The functions $f_i$ are $L_i$–smooth for all $i \in [n]$. $\widehat{L}^2$ is a constant such that $\frac{1}{n} \sum_{i=1}^{n} \|\nabla f_i(x) - \nabla f_i(y)\|^2 \leq \widehat{L}^2 \|x - y\|^2$ for all $x, y \in \mathbb{R}^d$ and $L_{\max} := \max_{i \in [n]} L_i$.

**Assumption 2.3.** The functions $f_i$ are convex and the function $f$ is $\mu$-strongly convex with $\mu \geq 0$ and attains a minimum at some point $x^* \in \mathbb{R}^d$.

To avoid ambiguity, the constants $L, \widehat{L}$, and $L_i$ are the smallest such numbers.

**Lemma 2.4.** *If Assumptions 2.1, 2.2 and 2.3 hold, then $\widehat{L} \leq L_{\max} \leq nL$ and $L \leq \widehat{L} \leq \sqrt{n}L$.*

### 2.1 COMMUNICATION COMPLEXITY OF VANILLA GRADIENT DESCENT

Solving the aforementioned optimization problem involves two key steps: i) the workers send results to the server (server-to-workers communication), ii) the server sends results to the workers (workers-

to-server communication). Let us first consider how this procedure works in the case of GD:

$$x^{t+1} = x^t - \gamma \nabla f(x^t) = x^t - \gamma \frac{1}{n} \sum_{i=1}^{n} \nabla f_i(x^t).$$

It is well known that if the function $f$ is $L$-smooth and $\mu$-strongly convex (see Assumptions 2.1 and 2.3), then GD with stepsize $\gamma = 1/L$ returns an $\varepsilon$-solution after $\mathcal{O}\left(L/\mu \log 1/\varepsilon\right)$ steps. In distributed setting, GD would require i) the workers to send $\nabla f_i(x^t)$ to the server ii) the server to send $x^{t+1}$ to the workers or, alternatively, ii) the server to send $\frac{1}{n} \sum_{i=1}^{n} \nabla f_i(x^t)$ to the workers, depending on whether the iterates $x^t$ are updated on the server or on the workers. Assuming that the communication complexity is proportional to the number of coordinates, the server-to-workers and workers-to-server communication complexities are equal $\mathcal{O}\left(dL/\mu \log 1/\varepsilon\right)$.

## 2.2 WORKERS-TO-SERVER (=UPLINK) COMPRESSION

We now move on to more advanced algorithms that aim to improve the workers-to-server communication complexity. These algorithms assume that the server-to-workers communication complexity is negligible and focus exclusively on sending the message from devices to the server. Such an approach can be justified by the fact that broadcast operation may in some systems be much faster than gather operation (Mishchenko et al., 2019; Kairouz et al., 2021). Moreover, the server can be considered to be just an abstraction representing "all other nodes", in which case server-to-worker communication does not exist at all.

The primary tools that help reduce communication cost are compression operators, such as vector sparsification and quantization (Beznosikov et al., 2020). However, compression injects error/noise into the process, as formalized in (2) and (5). Two canonical examples of compressors belonging to these two classes are the Top$K \in \mathbb{B}(k/d)$ and Rand$K \in \mathbb{U}(d/k - 1)$ sparsifiers. The former retains the $K$ largest values of the input vector, while the latter takes $K$ random values of this vector scaled by $d/k$ (Beznosikov et al., 2020). Further examples of compressors belonging to $\mathbb{B}(\alpha)$ and $\mathbb{U}(\omega)$ can be found in (Beznosikov et al., 2020).

The theory of methods supporting workers-to-server compression is reasonably well developed. In the convex and strongly convex setting, the current state-of-the-art methods are DIANA (Mishchenko et al., 2019), ADIANA (Li et al., 2020), and CANITA (Li & Richtárik, 2021). In the nonconvex setting, the current state-of-the-art methods are DCGD (Khaled & Richtárik, 2020) (in the low accuracy regime) and MARINA, DASHA, FRECON, and EF21 (Gorbunov et al., 2021; Tyurin & Richtárik, 2022b;a; Zhao et al., 2021; Richtárik et al., 2021) (in the high accuracy regime).

To see that these types of algorithms can achieve workers-to-server communication complexity that is no worse than that of GD, let us consider the DIANA method. In the strongly convex case, DIANA (Khaled et al., 2020) has the convergence rate $\mathcal{O}\left(((1 + \omega/n) L_{\max}/\mu + \omega) \log 1/\varepsilon\right)$. Using the Rand$K$ compression operator with $K = d/n$, the workers-to-server complexity is not greater than

$$\mathcal{O}\left(\frac{d}{n} \times \left(\left(1 + \frac{\omega}{n}\right) \frac{L_{\max}}{\mu} + \omega\right) \log \frac{1}{\varepsilon}\right) = \mathcal{O}\left(\left(\frac{dL_{\max}}{n\mu} + d\right) \log \frac{1}{\varepsilon}\right),$$

meaning that DIANA's complexity is better than GD's complexity $\mathcal{O}\left(dL/\mu \log 1/\varepsilon\right)$ (recall that $L_{\max} \leq nL$). The same reasoning applies to other algorithms in the convex and nonconvex worlds.

## 2.3 BIDIRECTIONAL COMPRESSION

In the previous section, we showed that it is possible to improve workers-to-server communication complexity of GD. But what about the server-to-workers compression? Does there exist a method that would also compress the information sent from the server to the workers and obtain the workers-to-server and server-to-workers communication complexities at least as good as with the vanilla GD method? As far as we know, the current answer to the question is NO!

Bidirectional compression has been considered in many papers, including (Horváth et al., 2019; Tang et al., 2019; Liu et al., 2020; Philippenko & Dieuleveut, 2020; 2021; Fatkhullin et al., 2021). In Table 1, we provide a comparison of methods applying this type of compression in the strongly convex setting. Let us now take a closer look at the MCM method of Philippenko & Dieuleveut (2021). For simplicity, we assume that the server and the workers use Rand$K$ compressors with

---

**Algorithm 1** EF21-P + DIANA

---

1: **Parameters:** learning rates $\gamma > 0$ (for learning the model) and $\beta > 0$ (for learning the gradient shifts); initial model $x^0 \in \mathbb{R}^d$ (stored on the server and the workers); initial gradient shifts $h_1^0, \ldots, h_n^0 \in \mathbb{R}^d$ (stored on the workers); **average of the initial gradient shifts** $h^0 = \frac{1}{n} \sum_{i=1}^n h_i^0$ (stored on the server); initial model shift $w^0 = x^0 \in \mathbb{R}^d$ (stored on the server and the workers)
2: **for** $t = 0, 1, \ldots, T - 1$ **do**
3:     **for** $i = 1, \ldots, n$ in parallel **do**
4:         $m_i^t = \mathcal{C}_i^D(\nabla f_i(w^t) - h_i^t)$          Worker $i$ compresses the shifted gradient via the dual compressor $\mathcal{C}_i^D \in \mathbb{U}(\omega)$
5:          Send compressed message $m_i^t$ to the server
6:         $h_i^{t+1} = h_i^t + \beta m_i^t$          Worker $i$ updates its local gradient shift with stepsize $\beta$
7:     **end for**
8:     $m^t = \frac{1}{n} \sum_{i=1}^n m_i^t$          Server averages the $n$ messages received from the workers
9:     $h^{t+1} = h^t + \beta m^t$          Server updates the average gradient shift so that $h^t = \frac{1}{n} \sum_{i=1}^n h_i^t$
10:    $g^t = h^t + m^t$          Server computes the gradient estimator
11:    $x^{t+1} = x^t - \gamma g^t$          Server takes a gradient-type step with stepsize $\gamma$
12:    $p^{t+1} = \mathcal{C}^P\left(x^{t+1} - w^t\right)$          Server compresses the shifted model via the primal compressor $\mathcal{C}^P \in \mathbb{B}(\alpha)$
13:    $w^{t+1} = w^t + p^{t+1}$          Server updates the model shift
14:    Broadcast compressed message $p^{t+1}$ to all $n$ workers
15:    **for** $i = 1, \ldots, n$ in parallel **do**
16:        $w^{t+1} = w^t + p^{t+1}$          Worker $i$ updates its local copy of the model shift
17:    **end for**
18: **end for**

---

parameters $K_s$ and $K_w$, respectively. The server-to-workers communication complexity of MCM is not less than

$$\Omega\left(K_s \times \left(1 + \omega_s^{3/2} + \frac{\omega_s \omega_w^{1/2}}{\sqrt{n}} + \frac{\omega_w}{n}\right) \frac{L_{\max}}{\mu} \log \frac{1}{\varepsilon}\right) = \Omega\left(\frac{d^{3/2}}{K_s^{1/2}} \frac{L_{\max}}{\mu} \log \frac{1}{\varepsilon}\right).$$

Thus, for any $K_s \in [1, d]$, the server-to-workers communication complexity is worse than the GD's complexity $\mathcal{O}\left(\frac{dL}{\mu} \log 1/\varepsilon\right)$ by a factor of $d^{1/2}/K_s^{1/2}$. The same reasoning applies to Dore (Liu et al., 2020) and Artemis (Philippenko & Dieuleveut, 2020):

$$\Omega\left(K_s \times \left(\frac{\omega_s \omega_w}{n}\right) \frac{L_{\max}}{\mu} \log \frac{1}{\varepsilon}\right) = \Omega\left(\frac{d^2}{K_w n} \frac{L_{\max}}{\mu} \log \frac{1}{\varepsilon}\right).$$

It turns out that one can find an example of problem (6) with $L_{\max} = nL$. Therefore, in the worst case scenario, the server-to-workers communication complexity can be up to $d/K_w$ times worse than the GD's complexity for any $K_w \in [1, d]$.

### 2.4 NEW METHODS

We are now ready to present our main method EF21-P + DIANA (see Algorithm 1), which is a combination of our EF21-P mechanism described in Section 1 (and analyzed in Sections D and E) and the DIANA method of Mishchenko et al. (2019); Horváth et al. (2022); Gorbunov et al. (2020a). The pseudocode of Algorithm 1 should be self-explanatory. If the gradient shifts $\{h_i^t\}$ employed by DIANA are initialized to zeros, and we choose $\beta = 0$, then DIANA reduces to DCGD, and EF21-P + DIANA thus reduces to EF21-P + DCGD (see Algorithm 2). If we further choose the dual/gradient compressors $\mathcal{C}_i^D$ to be identity mappings, then EF21-P + DCGD further reduces to EF21-P.

## 3 ANALYSIS IN THE CONVEX SETTING

Let us first state the convergence theorem.

**Theorem 3.1.** *Suppose that Assumptions 2.1, 2.2 and 2.3 hold, $\beta = \frac{1}{\omega+1}$, set $x^0 = w^0$ and let*

$$\gamma \leq \min\left\{\frac{n}{160\omega L_{\max}}, \frac{\sqrt{n\alpha}}{20\sqrt{\omega}\widehat{L}}, \frac{\alpha}{100L}, \frac{1}{(\omega+1)\mu}\right\}. \text{ Then Algorithm 1 returns } x^T \text{ such that}$$

$$\frac{1}{2\gamma}\mathrm{E}\left[\left\|x^T - x^*\right\|^2\right] + \mathrm{E}\left[f(x^T) - f(x^*)\right] \leq \left(1 - \frac{\gamma\mu}{2}\right)^T V^0,$$

*where $V^0 := \frac{1}{2\gamma}\mathrm{E}\left[\left\|x^0 - x^*\right\|^2\right] + \left(f(x^0) - f(x^*)\right) + \frac{8\gamma\omega(\omega+1)}{n^2} \sum_{i=1}^n \left\|h_i^0 - \nabla f_i(x^*)\right\|^2$.*

Table 1: **Strongly Convex Case.** The number of communication rounds to get an $\varepsilon$-solution ($\mathrm{E}[\|\widehat{x} - x^*\|^2] \leq \varepsilon$) up to logarithmic factors. To make comparison easier, if a method works with a biased compressor, we assume that the biased compressor is formed from the unbiased compressors and the following relations hold: $\omega_{\mathrm{w}} + 1 = 1/\alpha_{\mathrm{w}}$ and $\omega_{\mathrm{s}} + 1 = 1/\alpha_{\mathrm{s}}$, where $\omega_{\mathrm{w}}$ and $\omega_{\mathrm{s}}$ are parameters of workers-to-server and server-to-workers compressors, accordingly.

| Method | # Communication Rounds | Limitations |
|---|---|---|
| DIANA (Mishchenko et al., 2019) | $\left(1 + \frac{\omega_{\mathrm{w}}}{n}\right) \frac{L_{\max}}{\mu} + \omega_{\mathrm{w}}$ | No server-to-worker compression. |
| Dore, Artemis (Liu et al., 2020) (Philippenko & Dieuleveut, 2020) | $\Omega\left(\frac{\omega_{\mathrm{s}}\omega_{\mathrm{w}}}{n} \frac{L_{\max}}{\mu}\right)$ | — |
| MCM (Philippenko & Dieuleveut, 2021) | $\Omega\left(\left(\omega_{\mathrm{s}}^{3/2} + \frac{\omega_{\mathrm{s}}\omega_{\mathrm{w}}^{1/2}}{\sqrt{n}} + \frac{\omega_{\mathrm{w}}}{n}\right) \frac{L_{\max}}{\mu}\right)$ | — |
| EF21-P + DIANA (new) (Theorem 3.1) | $(1 + \omega_{\mathrm{s}}) \frac{L}{\mu} + \sqrt{\frac{(\omega_{\mathrm{s}}+1)\omega_{\mathrm{w}}}{n}} \frac{\widehat{L}}{\mu} + \frac{\omega_{\mathrm{w}}}{n} \frac{L_{\max}}{\mu} + \omega_{\mathrm{w}}$ or presented a bit less accurately: $(1 + \omega_{\mathrm{s}} + \frac{\omega_{\mathrm{w}}}{n}) \frac{L_{\max}}{\mu} + \omega_{\mathrm{w}}$ | — |
| EF21-P + DCGD (new) (Theorem G.3) | $(1 + \omega_{\mathrm{s}}) \frac{L}{\mu} + \sqrt{\frac{(\omega_{\mathrm{s}}+1)\omega_{\mathrm{w}}}{n}} \frac{\widehat{L}}{\mu} + \frac{\omega_{\mathrm{w}}}{n} \frac{L_{\max}}{\mu}$ or presented a bit less accurately: $(1 + \omega_{\mathrm{s}} + \frac{\omega_{\mathrm{w}}}{n}) \frac{L_{\max}}{\mu}$ | Interpolation regime: $\nabla f_i(x^*) = 0$ |

The above result means that EF21-P + DIANA guarantees an $\varepsilon$-solution after

$$T_{\mathrm{NEW}} := \mathcal{O}\left(\left(\frac{L}{\alpha\mu} + \sqrt{\frac{\omega}{\alpha n}} \frac{\widehat{L}}{\mu} + \frac{\omega}{n} \frac{L_{\max}}{\mu} + \omega\right) \log \frac{1}{\varepsilon}\right)$$

steps. Noting that $L \leq \widehat{L} \leq L_{\max}$ and $\sqrt{\frac{\omega_{\mathrm{w}}}{\alpha n}} \leq \frac{1}{2\alpha} + \frac{\omega_{\mathrm{w}}}{2n}$, this gives

$$T_{\mathrm{NEW}} = \mathcal{O}\left(\left(\left(\frac{1}{\alpha} + \frac{\omega}{n}\right) \frac{L_{\max}}{\mu} + \omega\right) \log \frac{1}{\varepsilon}\right).$$

Comparing this rate with rates achieved by the existing algorithms (see Table 1), our method is the first one to guarantee the decoupling of noises $\alpha$ and $\omega$ coming from the server-to-workers and the workers-to-server compressors. Moreover, it is more general, as the server-to-workers compression can use biased compressors, including Top$K$ and Rank$K$ (Safaryan et al., 2021). These can in practice perform better than the unbiassed ones (Beznosikov et al., 2020; Vogels et al., 2019).

As promised, let us now show that EF21-P + DIANA has the communication complexity better than GD. For simplicity, we assume that the server and the workers use Top$K$ and Rand$K$ compressors respectively. Since under this assumption $\omega = d/K - 1$ and $\alpha = K/d$, the server-to-workers and the workers-to-server communication complexities equal

$$\mathcal{O}\left(K \times \left(\frac{L}{\alpha\mu} + \sqrt{\frac{\omega}{\alpha n}} \frac{\widehat{L}}{\mu} + \frac{\omega}{n} \frac{L_{\max}}{\mu} + \omega\right) \log \frac{1}{\varepsilon}\right) = \mathcal{O}\left(\left(d\frac{L}{\mu} + \frac{d}{\sqrt{n}} \frac{\widehat{L}}{\mu} + \frac{d}{n} \frac{L_{\max}}{\mu}\right) \log \frac{1}{\varepsilon}\right).$$

Note that $L_{\max} \leq nL$ and $\widehat{L} \leq \sqrt{n}L$, so this complexity is no worse than the GD's complexity for any $K \in [1, d]$. The general convex case is discussed in Section F.1.

## 3.1 STOCHASTIC GRADIENTS

In this section, we assume that the workers in EF21-P + DIANA calculate stochastic gradients instead of exact gradients:

**Assumption 3.2** (Stochastic gradients). *For all $x \in \mathbb{R}^d$, stochastic gradients $\widetilde{\nabla} f_i(x)$ are unbiased and have bounded variance, i.e., $\mathrm{E}[\widetilde{\nabla} f_i(x)] = \nabla f_i(x)$, and $\mathrm{E}[\|\widetilde{\nabla} f_i(x) - \nabla f_i(x)\|^2] \leq \sigma^2$ for all $i \in [n]$, where $\sigma^2 \geq 0$.*

We now provide a generalization of Theorem 3.1:

**Theorem 3.3.** *Let us consider Algorithm 1 using stochastic gradients $\widetilde{\nabla} f_i$ instead of exact gradients $\nabla f_i$ for all $i \in [n]$. Let Assumptions 2.1, 2.2, 2.3 and 3.2 hold, $\beta = \frac{1}{\omega+1}$, $x^0 = w^0$, and $\gamma \leq$*

$\min \left\{ \frac{n}{160\omega L_{\max}}, \frac{\sqrt{n\alpha}}{20\sqrt{\omega}\widehat{L}}, \frac{\alpha}{100L}, \frac{1}{(\omega+1)\mu} \right\}$. *Then Algorithm 1 returns $x^T$ such that*

$$\frac{1}{2\gamma}\mathrm{E}\left[\left\|x^T - x^*\right\|^2\right] + \mathrm{E}\left[f(x^T) - f(x^*)\right] \le \left(1 - \frac{\gamma\mu}{2}\right)^T V^0 + \frac{24(\omega+1)\sigma^2}{\mu n},$$

*where $V^0 := \frac{1}{2\gamma}\mathrm{E}\left[\left\|x^0 - x^*\right\|^2\right] + \left(f(x^0) - f(x^*)\right) + \frac{8\gamma\omega(\omega+1)}{n^2}\sum_{i=1}^n \left\|h_i^0 - \nabla f_i(x^*)\right\|^2.$*

For general convex case, we refer to Theorem F.4. Note that Theorem 3.3 has the same convergence rate as Theorem 3.1, except for the *statistical* term $\mathcal{O}\left(\frac{(\omega+1)\sigma^2}{\mu n}\right)$ that is the same as in DIANA (Gorbunov et al., 2020a; Khaled et al., 2020) and does not depend on $\alpha$.

### 3.2 EF21-P + DCGD AND INTERPOLATION REGIME

We also analyze a second method, EF21-P + DCGD, which is based on DCGD (Khaled & Richtárik, 2020; Alistarh et al., 2017). One can think of DCGD as DIANA with parameter $\beta = 0$. On one hand, the convergence of EF21-P + DCGD is faster (see Theorem G.3) comparing to EF21-P + DIANA (see Theorem 3.1). On the other hand, we can guarantee the convergence only to a $\mathcal{O}(1/n\sum_{i=1}^n \|\nabla f_i(x^*)\|^2)$ "neighborhood" of the solution. However, this "neighborhood" disappears in the interpolation regime, i.e., when $\nabla f_i(x^*) = 0$ for all $i \in [n]$. The interpolation regime is very common in modern deep learning tasks (Brown et al., 2020; Bubeck & Sellke, 2021).

### 3.3 WHY DO BIDIRECTIONAL METHODS WORK MUCH BETTER THAN GD?

Our analysis of EF21-P + DIANA covers the worst case scenario for the values of $L_{\max}$ and $\alpha$. Although $L_{\max}$ can be equal to $nL$, in practice it tends to be much smaller. Similarly, the assumed bound on the parameter $\alpha$ equal to $k/d$ for the TopK compressor is also very conservative and the "effective" $\alpha$ is much larger (Beznosikov et al., 2020; Vogels et al., 2019; Xu et al., 2021). Note that Algorithm 1 does not depend on $\alpha$! Our claims are also supported by experiments from Section 5.

## 4 ANALYSIS IN THE NONCONVEX SETTING

In the nonconvex case, existing bidirectional methods suffer from the same problem as those used in the convex case (see Section 2.3): they either do not provide server-to-workers compression at all, or the compressor errors/noises are coupled in a multiplicative fashion (see $\omega_w$ and $\omega_s$ in Table 2).

Instead of the convexity (see Assumption 2.3), we will need the following assumption:

**Assumption 4.1** (Lower boundedness). There exist $f^* \in \mathbb{R}$ and $f_1^*, \ldots, f_n^* \in \mathbb{R}$ such that $f(x) \ge f^*$ and $f_i(x) \ge f_i^*$ for all $x \in \mathbb{R}^d$ and for all $i \in [n]$.

As in the convex setting, the theory of methods that only use workers-to-server compression is well examined. In the high accuracy regimes, the current state-of-the-art methods are MARINA and DASHA (Gorbunov et al., 2021; Tyurin & Richtárik, 2022b); both return an $\varepsilon$-stationary point after $\mathcal{O}\left(\frac{\Delta_0 L}{\varepsilon} + \frac{\Delta_0\omega\widehat{L}}{\sqrt{n}\varepsilon}\right)$ iterations, where $\Delta_0 := f(x^0) - f^*$. In the low accuracy regimes, the current state-of-the-art method is DCGD (Khaled & Richtárik, 2020), with an iteration complexity $\mathcal{O}\left(\frac{\Delta_0 L}{\varepsilon} + \frac{\Delta_0(\Delta_0+\Delta^*)(1+\omega)LL_{\max}}{n\varepsilon^2}\right)$, where $\Delta^* := f^* - \frac{1}{n}\sum_{i=1}^n f_i^*$. Note that DCGD has worse dependence on $\varepsilon$, but it scales much better with the number of workers $n$.

We now investigate how EF21-P can help us in the general nonconvex case. Let us recall that in the convex case, decoupling of the noises coming from two compression schemes can be achieved by combining EF21-P with DIANA. In the nonconvex setting, we successfully combine EF21-P and DCGD. Moreover, we provide analysis of some particular cases where EF21-P + DCGD can be the method of choice in the high accuracy regimes.

Whether or not it is possible to achieve the decoupling by combining our method with MARINA or DASHA is not yet known and we leave it to future work[3].

---

[3] We did not try to get the convergence rate of EF21-P + DIANA in the nonconvex regime because it is well known that DIANA is a suboptimal method in the nonconvex case (Gorbunov et al., 2021).

Table 2: **General nonconvex Case.** The # of communication rounds to get an $\varepsilon$-stationary point $(\mathrm{E}[\|\nabla f(\widehat{x})\|^2] \leq \varepsilon)$. For simplicity, we assume that $f_i^* = f^*$ for all $i \in [n]$ and only the terms with respect to $\omega_\mathrm{w}$ and $\omega_\mathrm{s}$ are shown. The parameters $\omega_\mathrm{w}$ and $\omega_\mathrm{s}$ have the same meaning as in Table 1.

| Method | # Communication Rounds | Limitations |
|---|---|---|
| DCGD (Khaled & Richtárik, 2020) | $\frac{\Delta_0^2 \omega_\mathrm{w} L L_{\max}}{n \varepsilon^2}$ | No server-to-worker compression. |
| MARINA, DASHA (Gorbunov et al., 2021) (Tyurin & Richtárik, 2022b) | $\frac{\Delta_0 \omega_\mathrm{w} \widehat{L}}{\sqrt{n} \varepsilon}$ | No server-to-worker compression. |
| MCM (Philippenko & Dieuleveut, 2021) | $\Delta_0 \left( \frac{\omega_\mathrm{s}^{3/2}}{\varepsilon} + \frac{\omega_\mathrm{s} \omega_\mathrm{w}^{1/2}}{\sqrt{n}\varepsilon} + \frac{\omega_\mathrm{w}}{n\varepsilon} \right) L_{\max}$ | Only homogeneous case, i.e., $f_i = f$ for all $i \in [n]$. |
| CD-Adam (Wang et al., 2022) | $\Omega \left( \frac{\sqrt{d} \max\{\omega_\mathrm{s}, \omega_\mathrm{w}\}^4}{\varepsilon^2} \right)$ | Bounded gradient assumption. |
| EF21-BC (Fatkhullin et al., 2021) | $\frac{\Delta_0 \omega_\mathrm{w} \omega_\mathrm{s} \widehat{L}}{\varepsilon}$ | — |
| NEOLITHIC (Huang et al., 2022) | $\frac{\Delta_0 L_{\max}}{\varepsilon}$ | Does not compress vectors.[a] Bounded gradient assumption. |
| EF21-P + DCGD (new) | $\frac{\Delta_0^2 \omega_\mathrm{w} L L_{\max}}{n \varepsilon^2} + \frac{\Delta_0 \omega_\mathrm{s} L}{\varepsilon}$ | — |
| EF21-P + DCGD (new) | $\frac{\Delta_0 D \omega_\mathrm{w} L}{n \varepsilon} + \frac{\Delta_0 \omega_\mathrm{s} L}{\varepsilon}$ | Strong-growth assumption with parameter $D$. |

[a] In each communication round, NEOLITHIC sends the number of compressed vectors proportional to $1/\alpha$, where $\alpha$ is the parameter of a biased compressor. For $\mathrm{Top}K$ or $\mathrm{Rand}K$, it means that NEOLITHIC sends $\Theta(d/K)$ sparsified vectors with $K$ nonzero elements, meaning that, in total, $\Theta(d)$ values are sent in each communication round.

## 4.1 EF21-P + DCGD IN THE GENERAL NONCONVEX CASE

Without any restrictive assumptions, we can prove the following convergence result:

**Theorem 4.2.** *Consider Algorithm 2 and let Assumptions 2.1, 2.2 and 4.1 hold, $x^0 = w^0$, and $\gamma = \min\left\{ \frac{\alpha}{8L}, \frac{\sqrt{n}}{\sqrt{2\omega L L_{\max} T}}, \frac{n\varepsilon}{32\Delta^* \omega L L_{\max}} \right\}$. Then*

$$T \geq \frac{48\Delta_0 L}{\varepsilon} \max\left\{ \frac{8}{\alpha}, \frac{96\Delta_0 \omega L_{\max}}{n\varepsilon}, \frac{32\Delta^* \omega L_{\max}}{n\varepsilon} \right\} \quad \Rightarrow \quad \min_{0 \leq t \leq T-1} \mathrm{E}\left[\|\nabla f(x^t)\|^2\right] \leq \varepsilon.$$

*(The proof follows from Theorem E.3 and Proposition E.4 (Part 1)).*

We get the rate of DCGD (Khaled & Richtárik, 2020) plus an additional $\mathcal{O}\left( \frac{\Delta_0 L}{\alpha \varepsilon} \right)$ factor, thus obtaining the first method with bidirectional compression where the noises from the compressors are decoupled. Moreover, as noted before, this method provides the state-of-the-art rates when $\varepsilon$ is small or the number of workers $n$ is large.

## 4.2 STRONG GROWTH CONDITION

Here we analyze EF21-P + DCGD under the strong-growth condition (Schmidt & Roux, 2013).

**Assumption 4.3.** There exists $D > 0$ such that $\frac{1}{n} \sum_{i=1}^n \|\nabla f_i(x)\|^2 \leq D \|\nabla f(x)\|^2$ for all $x \in \mathbb{R}^d$.

While this assumption is restrictive and does not even hold for quadratic optimization problems, there exist numerous practical applications when it is reasonable. These include, for example, deep learning, where the number of parameters $d$ is so huge that the model can interpolate the training dataset (Schmidt & Roux, 2013; Vaswani et al., 2019; Meng et al., 2020). To train such models, engineers use distributed environments, in which case communication becomes the main a bottleneck (Ramesh et al., 2021). For these problems, our method is suitable and can be successfully applied.

**Theorem 4.4.** *Consider Algorithm 2, let Assumptions 2.1, 2.2, 4.1 and 4.3 hold, and choose $x^0 = w^0$ and $\gamma = \min\left\{ \frac{\alpha}{8L}, \frac{n}{4D\omega L} \right\}$. Then*

$$T \geq \frac{48\Delta_0 L}{\varepsilon} \max\left\{ \frac{8}{\alpha}, \frac{4D\omega}{n} \right\} \quad \Rightarrow \quad \min_{0 \leq t \leq T-1} \mathrm{E}\left[\|\nabla f(x^t)\|^2\right] \leq \varepsilon.$$

*(The proof follows from Theorem E.3 and Proposition E.4 (Part 2)).*

Comparing to Section 4.1, the above result shows an improved dependence on $\varepsilon$ under the strong growth assumption.

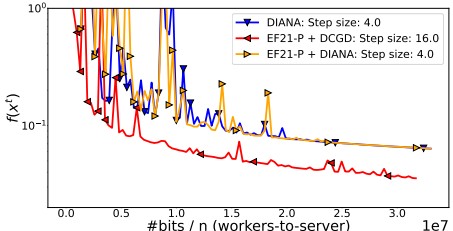 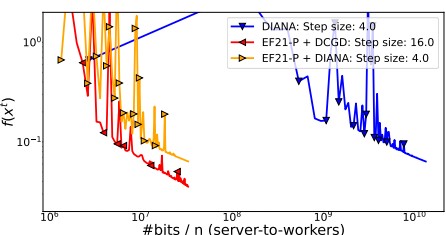

Figure 1: Logistic Regression with *real-sim* dataset. Number of workers $n = 100$. Sparsification level was set to $K = 100$ for all compressors.

### 4.3 HOMOGENEOUS FUNCTIONS

Another important problem where our method can be useful is distributed optimization with homogeneous (identical) functions and stochastic gradients. In particular, we consider the case when $f_i = f$ for all $i \in [n]$ and instead of exact gradients, stochastic gradients are used. This assumption holds, for instance, for distributed machine learning problems where every worker samples mini-batches from a large shared dataset (Recht et al., 2011; Goyal et al., 2017).

**Theorem 4.5.** *Let us consider Algorithm 2 with the stochastic gradients $\widetilde{\nabla} f$ instead of the exact gradients $\nabla f$. Suppose that Assumptions 2.1, 2.2, 3.2 and 4.1 hold and $f_i = f$ for all $i \in [n]$. Set $x^0 = w^0$ and let $\gamma = \min \left\{ \frac{\alpha}{8L}, \frac{1}{4\left(\frac{\omega}{n}+1\right)L}, \frac{n\varepsilon}{16(\omega+1)\sigma^2 L} \right\}$. Then*

$$T \geq \frac{48\Delta_0 L}{\varepsilon} \max\left\{ \frac{8}{\alpha}, 4\left(\frac{\omega}{n}+1\right), \frac{16(\omega+1)\sigma^2}{n\varepsilon} \right\} \quad \Rightarrow \quad \min_{0 \leq t \leq T-1} \mathrm{E}\left[\|\nabla f(x^t)\|^2\right] \leq \varepsilon.$$

*(The proof follows from Theorem E.3 and Proposition E.4 (Part 4)).*

Under exactly the same assumptions, MCM method by (Philippenko & Dieuleveut, 2020) with bidirectional compression guarantees the convergence rate $\Theta\left( \frac{\Delta_0 L}{\varepsilon} \max\left\{ \left(\frac{\omega_{\mathrm{w}}}{n}+1\right), \omega_{\mathrm{s}}^{3/2}, \frac{\omega_{\mathrm{s}}\omega_{\mathrm{w}}^{1/2}}{n}, \frac{(\omega_{\mathrm{w}}+1)\sigma^2}{n\varepsilon} \right\} \right)$. Comparing this with our result, the last *statistical* term $(\omega+1)\sigma^2/n\varepsilon$ is the same in both cases, but we significantly improve the other *communication* terms (take $\omega = \omega_{\mathrm{w}}$ and $\alpha = (\omega_{\mathrm{s}}+1)^{-1}$ in Theorem 4.5).

## 5 EXPERIMENTAL HIGHLIGHTS

We now provide a few highlights from our experiments. For more details and experiments, we refer to Section A, where we compare our algorithms with the previous state-of-the-art method MCM and solve a nonconvex task. We consider the logistic regression task with *real-sim* (# of features = 20,958, # of samples equals 72,309) from LIBSVM dataset (Chang & Lin, 2011). Each plot represents the relations between function values and the total number of coordinates transmitted from and to the server. In all algorithms, the Rand$K$ compressor is used to compress information from the workers to the server. In the case of EF21-P + DIANA and EF21-P + DCGD, we take Top$K$ compressor to compress from the server to the workers. The results are presented in Figure 1. The main conclusion from these experiments is that EF21-P + DIANA and EF21-P + DCGD converge to a solution not slower than DIANA, even though DIANA does not compress vectors sent from the server to the workers! This means that EF21-P + DIANA and EF21-P + DCGD can send $\times 400$ less values from the server to the workers for free! Moreover, we see that EF21-P + DCGD converges faster than its competitors.

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

CONTENTS

# A   FURTHER EXPERIMENTS

We now provide the results of our experiments on practical machine learning tasks with LIBSVM datasets (Chang & Lin, 2011) (under the 3-clause BSD license). Each plot represents the relations between function values and the total number of coordinates transmitted from and to the server. The parameters of the algorithms are as suggested by the theory, except for the stepsizes $\gamma$ that we finetune from a set $\{2^i \mid i \in [-10, 10]\}$.

We solve the logistic regression problem:

$$f_i(x_1, \ldots, x_c) := -\frac{1}{m} \sum_{j=1}^{m} \log \left( \frac{\exp\left(a_{ij}^\top x_{y_{ij}}\right)}{\sum_{y=1}^{c} \exp\left(a_{ij}^\top x_y\right)} \right),$$

where $x_1, \ldots, x_c \in \mathbb{R}^d$, $c$ is the number of unique labels, $a_{ij} \in \mathbb{R}^d$ is a feature of a sample on the $i^{\text{th}}$ worker, $y_{ij}$ is a corresponding label and $m$ is the number of samples located on the $i^{\text{th}}$ worker. In all algorithms, the Rand$K$ compressor is used to compress information from the workers to the server. In the case of EF21-P + DIANA and EF21-P + DCGD, we take Top$K$ compressor to compress from the server to the workers. The performance of algorithms is compared on *w8a* (# of features = 300, # of samples equals 49,749), CIFAR10 (Krizhevsky et al., 2009) (# of features = 3072, # of samples equals 50,000), and *real-sim* (# of features = 20958, # of samples equals 72,309) datasets.

The results are presented in Figures 1, 2 and 3. The conclusions are the same as in Section 5. One can see that EF21-P + DIANA and EF21-P + DCGD converge to a solution not slower than DIANA, even though DIANA does not compress vectors sent from the server to the workers! EF21-P + DIANA and EF21-P + DCGD send $\times 100 - \times 1000$ less values from the server to the workers!

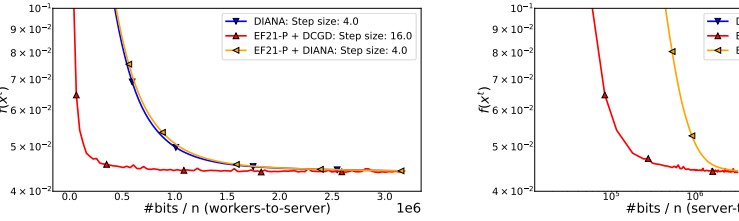

Figure 2: Logistic Regression with *w8a* dataset. # of workers $n = 10$. $K = 10$ in all compressors.

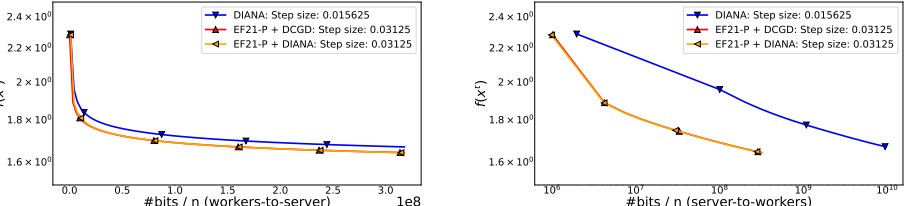

Figure 3: Logistic Regression with *CIFAR10* dataset. # of workers $n = 10$. $K = 1000$ in all compressors.

We also compare our algorithm to MCM. Since MCM does not support contractive compressors defined in 2, we use Rand$K$ instead of the Top$K$ compressor in the server-to-workers compression. Figure 4 shows that our new algorithms converge faster.

Finally, we provide experiments for the nonconvex setting and compare EF21-P + DCGD against EF21-BC (Fatkhullin et al., 2021) and DASHA (Tyurin & Richtárik, 2022b). We consider the logistic regression with a nonconvex regularizer

$$r(x_1, \ldots, x_c) := \lambda \sum_{y=1}^{c} \sum_{k=1}^{d} \frac{[x_y]_k^2}{1 + [x_y]_k^2},$$

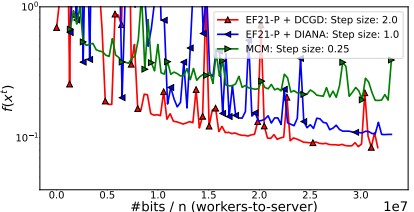 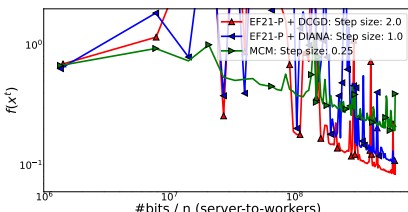

Figure 4: Logistic Regression with *real-sim* dataset. # of workers $n = 100$. The parameters of workers-to-server and server-to-workers compressors are $K_{\mathrm{w}} = 100$ and $K_{\mathrm{s}} = 2000$.

where $[\cdot]_k$ is an indexing operation of a vector and $\lambda = 0.001$. We use $\mathrm{Rand}K$ and $\mathrm{Top}K$ compressors for the workers-to-server and server-to-workers compressions, respectively. Note that in these experiments, the server-to-workers compression is only supported by EF21-P + DCGD and EF21-BC. In Figure 5, one can see that EF21-P + DCGD converges faster than other algorithms and outperforms DASHA, which does not compress vectors when transmitting them from the server to the workers.

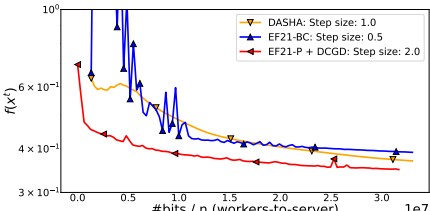 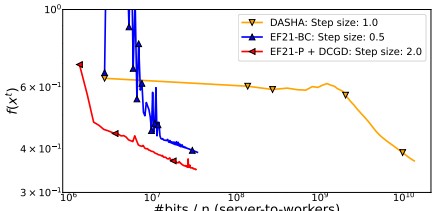

Figure 5: Logistic Regression with the nonconvex regularizer and *real-sim* dataset. # of workers $n = 100$. $K = 100$ in all compressors.

# B USEFUL IDENTITIES AND INEQUALITIES

For all $x, y, x_1, \ldots, x_n \in \mathbb{R}^d$, $s > 0$ and $\alpha \in (0, 1]$, we have:

$$\|x + y\|^2 \leq (1 + s) \|x\|^2 + (1 + s^{-1}) \|y\|^2, \tag{7}$$

$$\|x + y\|^2 \leq 2 \|x\|^2 + 2 \|y\|^2, \tag{8}$$

$$\langle x, y \rangle \leq \frac{\|x\|^2}{2s} + \frac{s \|y\|^2}{2}, \tag{9}$$

$$(1 - \alpha)\left(1 + \frac{\alpha}{2}\right) \leq 1 - \frac{\alpha}{2}, \tag{10}$$

$$(1 - \alpha)\left(1 + \frac{2}{\alpha}\right) \leq \frac{2}{\alpha}, \tag{11}$$

$$\langle a, b \rangle = \frac{1}{2}\left(\|a\|^2 + \|b\|^2 - \|a - b\|^2\right). \tag{12}$$

**Tower property:** For any random variables $X$ and $Y$, we have

$$\mathrm{E}\left[\mathrm{E}\left[X \mid Y\right]\right] = \mathrm{E}\left[X\right]. \tag{13}$$

**Variance decomposition:** For any random vector $X \in \mathbb{R}^d$ and any non-random $c \in \mathbb{R}^d$, we have

$$\mathrm{E}\left[\|X - c\|^2\right] = \mathrm{E}\left[\|X - \mathrm{E}\left[X\right]\|^2\right] + \|\mathrm{E}\left[X\right] - c\|^2. \tag{14}$$

**Lemma B.1** (Nesterov (2018)). *Let $f : \mathbb{R}^d \to \mathbb{R}$ be a function for which Assumptions 2.1 and 2.3 are satisfied. Then for all $x, y \in \mathbb{R}^d$ we have:*

$$\|\nabla f(x) - \nabla f(y)\|^2 \leq 2L(f(x) - f(y) - \langle \nabla f(y), x - y \rangle). \tag{15}$$

**Lemma B.2** (Khaled & Richtárik (2020)). *Let $f$ be a function for which Assumptions 2.1 and 4.1 are satisfied. Then for all $x, y \in \mathbb{R}^d$ we have:*

$$\|\nabla f(x)\|^2 \leq 2L(f(x) - f^*). \tag{16}$$

## C   PROOF OF LEMMA 2.4

**Lemma 2.4.** *If Assumptions 2.1, 2.2 and 2.3 hold, then $\widehat{L} \leq L_{\max} \leq nL$ and $L \leq \widehat{L} \leq \sqrt{n}L$.*

*Proof.* One can show (see (Nesterov, 2003)) that a convex function $f$ is $L$-smooth if and only if either of the two conditions below holds:

$$0 \leq \langle \nabla f(x) - \nabla f(x), x - y \rangle \leq L \left\| x - y \right\|^2, \quad \forall x, y \in \mathbb{R}^d,$$

$$\left\| \nabla f(x) - \nabla f(x) \right\|^2 \leq L \left\langle \nabla f(x) - \nabla f(x), x - y \right\rangle, \quad \forall x, y \in \mathbb{R}^d.$$

For any fixed $i \in [n]$, we have

$$
\begin{aligned}
\langle \nabla f_i(x) - \nabla f_i(y), x - y \rangle &\leq \sum_{i=1}^{n} \langle \nabla f_i(x) - \nabla f_i(y), x - y \rangle \\
&= n \frac{1}{n} \sum_{i=1}^{n} \langle \nabla f_i(x) - \nabla f_i(y), x - y \rangle \\
&= n \langle \nabla f(x) - \nabla f(y), x - y \rangle \\
&\leq n \left\| \nabla f(x) - \nabla f(y) \right\| \left\| x - y \right\| \\
&\overset{(2.1)}{\leq} nL \left\| x - y \right\|^2.
\end{aligned}
$$

Thus $L_i \leq nL$ and $L_{\max} \leq nL$. Next,

$$
\begin{aligned}
\frac{1}{n} \sum_{i=1}^{n} \left\| \nabla f_i(x) - \nabla f_i(y) \right\|^2 &\leq \frac{1}{n} \sum_{i=1}^{n} L_i \langle \nabla f_i(x) - \nabla f_i(y), x - y \rangle \\
&\leq L_{\max} \frac{1}{n} \sum_{i=1}^{n} \langle \nabla f_i(x) - \nabla f_i(y), x - y \rangle \\
&= L_{\max} \langle \nabla f(x) - \nabla f(y), x - y \rangle \\
&\leq L_{\max} \left\| \nabla f(x) - \nabla f(y) \right\| \left\| x - y \right\| \\
&\overset{(2.1)}{\leq} L_{\max} L \left\| x - y \right\|^2 \\
&\leq nL^2 \left\| x - y \right\|^2,
\end{aligned}
$$

and hence $\widehat{L} \leq \sqrt{n}L$. Using Jensen's inequality, we have

$$\left\| \nabla f(x) - \nabla f(y) \right\|^2 \leq \frac{1}{n} \sum_{i=1}^{n} \left\| \nabla f_i(x) - \nabla f_i(y) \right\|^2 \leq \widehat{L}^2 \left\| x - y \right\|^2.$$

Thus $L \leq \widehat{L}$. Finally, $\widehat{L} \leq L_{\max}$ follows from

$$\frac{1}{n} \sum_{i=1}^{n} \left\| \nabla f_i(x) - \nabla f_i(y) \right\|^2 \leq \frac{1}{n} \sum_{i=1}^{n} L_i^2 \left\| x - y \right\|^2 \leq L_{\max}^2 \left\| x - y \right\|^2.$$

$\square$

# D CONVERGENCE OF EF21-P IN THE STRONGLY CONVEX REGIME

We now provide the convergence rate of EF21-P from Section 1 in the strongly convex case.

**Theorem D.1.** *Let Assumptions 2.1 and 2.3 hold, set $w^0 = x^0$ and choose $\gamma \leq \frac{\alpha}{16L}$. Then EF21-P returns $x^T$ such that*

$$\frac{1}{2\gamma} \mathrm{E}\left[\left\|x^T - x^*\right\|^2\right] + \mathrm{E}\left[f(x^T) - f(x^*)\right] \leq \left(1 - \frac{\gamma\mu}{2}\right)^T \left(\frac{1}{2\gamma} \mathrm{E}\left[\left\|x^0 - x^*\right\|^2\right] + \left(f(x^0) - f(x^*)\right)\right).$$

*Moreover, $\mathrm{E}\left[\left\|w^t - x^*\right\|^2\right] \to 0$ as $t \to \infty$.*

Theorem D.1 states that EF21-P will return an $\varepsilon$-solution after $\mathcal{O}\left(\frac{L}{\alpha\mu} \log 1/\varepsilon\right)$ steps. Comparing to GD's rate $\mathcal{O}\left(\frac{L}{\mu} \log 1/\varepsilon\right)$, one can see that EF21-P converges $1/\alpha$ times slower.

*Proof.* First, let us note that

$$\left\|x^t - x^*\right\|^2 - \left\|x^{t+1} - x^*\right\|^2 - \left\|x^{t+1} - x^t\right\|^2$$
$$= \left\langle x^t - x^{t+1}, x^t - 2x^* + x^{t+1}\right\rangle - \left\langle x^{t+1} - x^t, x^{t+1} - x^t\right\rangle$$
$$= 2\left\langle x^t - x^{t+1}, x^{t+1} - x^*\right\rangle$$
$$= 2\gamma\left\langle \nabla f(w^t), x^{t+1} - x^*\right\rangle. \tag{17}$$

Using $L$-smoothness of $f$ (Assumption 2.1), we obtain

$$
\begin{aligned}
f(x^{t+1}) \;&\leq\; f(w^t) + \left\langle \nabla f(w^t), x^{t+1} - w^t\right\rangle + \frac{L}{2}\left\|x^{t+1} - w^t\right\|^2 \\
&\overset{\text{conv-ty}}{\leq}\; f(x^*) + \left\langle \nabla f(w^t), x^{t+1} - x^*\right\rangle - \frac{\mu}{2}\left\|w^t - x^*\right\|^2 + \frac{L}{2}\left\|x^{t+1} - w^t\right\|^2 \\
&\overset{(17)}{\leq}\; f(x^*) + \frac{1}{2\gamma}\left\|x^t - x^*\right\|^2 - \frac{1}{2\gamma}\left\|x^{t+1} - x^*\right\|^2 - \frac{1}{2\gamma}\left\|x^{t+1} - x^t\right\|^2 \\
&\qquad - \frac{\mu}{2}\left\|w^t - x^*\right\|^2 + \frac{L}{2}\left\|x^{t+1} - w^t\right\|^2.
\end{aligned}
$$

Using (8), we have

$$\frac{L}{2}\left\|x^{t+1} - w^t\right\|^2 \leq L\left\|x^{t+1} - x^t\right\|^2 + L\left\|w^t - x^t\right\|^2$$

and

$$\frac{\mu}{4}\left\|x^t - x^*\right\|^2 \leq \frac{\mu}{2}\left\|w^t - x^*\right\|^2 + \frac{\mu}{2}\left\|w^t - x^t\right\|^2 \leq \frac{\mu}{2}\left\|w^t - x^*\right\|^2 + L\left\|w^t - x^t\right\|^2,$$

where we used the fact that $\mu \leq L$. Hence

$$
\begin{aligned}
f(x^{t+1}) &\leq f(x^*) + \frac{1}{2\gamma}\left\|x^t - x^*\right\|^2 - \frac{1}{2\gamma}\left\|x^{t+1} - x^*\right\|^2 - \frac{1}{2\gamma}\left\|x^{t+1} - x^t\right\|^2 \\
&\qquad - \frac{\mu}{2}\left\|w^t - x^*\right\|^2 + \frac{L}{2}\left\|x^{t+1} - w^t\right\|^2 \\
&\leq f(x^*) + \frac{1}{2\gamma}\left\|x^t - x^*\right\|^2 - \frac{1}{2\gamma}\left\|x^{t+1} - x^*\right\|^2 - \frac{1}{2\gamma}\left\|x^{t+1} - x^t\right\|^2 \\
&\qquad + L\left\|w^t - x^t\right\|^2 - \frac{\mu}{4}\left\|x^t - x^*\right\|^2 + L\left\|x^{t+1} - x^t\right\|^2 + L\left\|w^t - x^t\right\|^2 \\
&= f(x^*) + \frac{1}{2\gamma}\left(1 - \frac{\gamma\mu}{2}\right)\left\|x^t - x^*\right\|^2 - \frac{1}{2\gamma}\left\|x^{t+1} - x^*\right\|^2 \\
&\qquad - \left(\frac{1}{2\gamma} - L\right)\left\|x^{t+1} - x^t\right\|^2 + 2L\left\|w^t - x^t\right\|^2 \\
&\leq f(x^*) + \frac{1}{2\gamma}\left(1 - \frac{\gamma\mu}{2}\right)\left\|x^t - x^*\right\|^2 - \frac{1}{2\gamma}\left\|x^{t+1} - x^*\right\|^2 + 2L\left\|w^t - x^t\right\|^2,
\end{aligned}
$$

where the last inequality follows from the fact that $\gamma \leq \frac{1}{2L}$. Let us denote by $\mathrm{E}_{t+1}\left[\cdot\right]$ the expectation conditioned on previous iterations $\{0, \ldots, t\}$. Then

$$
\mathrm{E}_{t+1}\left[f(x^{t+1})\right] \leq f(x^*) + \frac{1}{2\gamma}\left(1 - \frac{\gamma\mu}{2}\right)\left\|x^t - x^*\right\|^2
$$
$$
- \frac{1}{2\gamma}\mathrm{E}_{t+1}\left[\left\|x^{t+1} - x^*\right\|^2\right] + 2L\left\|w^t - x^t\right\|^2. \tag{18}
$$

It remains to bound $\mathrm{E}_{t+1}\left[\left\|w^{t+1} - x^{t+1}\right\|^2\right]$:

$$
\begin{aligned}
\mathrm{E}_{t+1}\left[\left\|w^{t+1} - x^{t+1}\right\|^2\right] &= \mathrm{E}_{t+1}\left[\left\|w^t + \mathcal{C}^p(x^{t+1} - w^t) - x^{t+1}\right\|^2\right]\\
&\overset{(2)}{\leq} (1-\alpha)\mathrm{E}_{t+1}\left[\left\|x^{t+1} - w^t\right\|^2\right]\\
&= (1-\alpha)\left\|x^t - \gamma\nabla f(w^t) - w^t\right\|^2\\
&\overset{(7)}{\leq} \left(1 - \frac{\alpha}{2}\right)\left\|w^t - x^t\right\|^2 + \frac{2\gamma^2}{\alpha}\left\|\nabla f(w^t)\right\|^2\\
&\overset{(8)}{\leq} \left(1 - \frac{\alpha}{2}\right)\left\|w^t - x^t\right\|^2 + \frac{4\gamma^2}{\alpha}\left\|\nabla f(w^t) - \nabla f(x^t)\right\|^2\\
&\quad + \frac{4\gamma^2}{\alpha}\left\|\nabla f(x^t) - \nabla f(x^*)\right\|^2\\
&\overset{(2.1),(B.1)}{\leq} \left(1 - \frac{\alpha}{2} + \frac{4\gamma^2 L^2}{\alpha}\right)\left\|w^t - x^t\right\|^2 + \frac{8\gamma^2 L}{\alpha}\left(f(x^t) - f(x^*)\right)\\
&\leq \left(1 - \frac{\alpha}{4}\right)\left\|w^t - x^t\right\|^2 + \frac{8\gamma^2 L}{\alpha}\left(f(x^t) - f(x^*)\right),
\end{aligned}
$$

where in the last step we assume that $\gamma \leq \frac{\alpha}{4L}$. Adding a $\frac{16L}{\alpha}$ multiple of the above inequality to (18), we obtain

$$
\mathrm{E}_{t+1}\left[f(x^{t+1})\right] + \frac{16L}{\alpha}\mathrm{E}_{t+1}\left[\left\|w^{t+1} - x^{t+1}\right\|^2\right] \leq f(x^*) + \frac{1}{2\gamma}\left(1 - \frac{\gamma\mu}{2}\right)\left\|x^t - x^*\right\|^2
$$
$$
- \frac{1}{2\gamma}\mathrm{E}_{t+1}\left[\left\|x^{t+1} - x^*\right\|^2\right] + \frac{16L}{\alpha}\left(1 - \frac{\alpha}{8}\right)\left\|w^t - x^t\right\|^2 + \frac{128\gamma^2 L^2}{\alpha^2}\left(f(x^t) - f(x^*)\right).
$$

Thus, taking full expectation over both sides of the inequality and considering $\gamma \leq \frac{\alpha}{16L} \leq \frac{\alpha}{4\mu}$ gives

$$
\mathrm{E}\left[f(x^{t+1}) - f(x^*)\right] + \frac{1}{2\gamma}\mathrm{E}\left[\left\|x^{t+1} - x^*\right\|^2\right] + \frac{16L}{\alpha}\mathrm{E}\left[\left\|w^{t+1} - x^{t+1}\right\|^2\right]
$$
$$
\leq \left(1 - \frac{\gamma\mu}{2}\right)\left(\mathrm{E}\left[f(x^t) - f(x^*)\right] + \frac{1}{2\gamma}\mathrm{E}\left[\left\|x^t - x^*\right\|^2\right] + \frac{16L}{\alpha}\mathrm{E}\left[\left\|w^t - x^t\right\|^2\right]\right).
$$

Applying this inequality iteratively and using the assumption $w^0 = x^0$ proves the result. $\qquad\square$

# E  CONVERGENCE OF EF21-P IN THE SMOOTH NONCONVEX REGIME

## E.1  GENERAL CONVERGENCE THEORY

We now move on to study how the EF21-P method can be used in the nonconvex regime. The analysis relies on the expected smoothness assumption introduced by Khaled & Richtárik (2020). In their work, they study SGD methods, performing iterations of the form

$$x^{t+1} = x^t - \gamma g^t,$$

where $g^t$ is an unbiased estimator of the true gradient $\nabla f(x^t)$. Following Khaled & Richtárik (2020), we shall assume that $\mathrm{E}\,[g(x)] = \nabla f(x)$. However, in our case, gradients will be evaluated at perturbed points, thus resulting in biased stochastic gradient estimators. In particular, we consider the following general update rule, where the stochastic gradients are calculated at points evolving according to the EF21-P mechanism, rather than at the current iterate:

$$x^{t+1} = x^t - \gamma g(w^t),$$
$$w^{t+1} = w^t + \mathcal{C}^P(x^{t+1} - w^t). \tag{19}$$

Our result covers a wide range of sources of stochasticity that may be present in $g$. For a detailed discussion of the topic, we refer the reader to the original paper (Khaled & Richtárik, 2020).

Throughout this section, we will rely on the following assumptions:

**Assumption E.1.** The stochastic gradient $g(x)$ is an unbiased estimator of the true gradient $\nabla f(x)$, i.e.,

$$\mathrm{E}\,[g(x)] = \nabla f(x)$$

for all $x \in \mathbb{R}^d$.

**Assumption E.2** (From Khaled & Richtárik (2020)). There exist constants $A, B, C \geq 0$ such that:

$$\mathrm{E}\left[\|g(x)\|^2\right] \leq 2A(f(x) - f^*) + B\,\|\nabla f(x)\|^2 + C$$

for all $x \in \mathbb{R}^d$.

We are ready to state the main theorem:

**Theorem E.3.** *Let Assumptions 2.1, 4.1, E.1 and E.2 hold and set $w^0 = x^0$. Fix $\varepsilon > 0$ and choose the stepsize*

$$\gamma = \min\left\{\frac{\alpha}{8L}, \frac{1}{4BL}, \frac{1}{\sqrt{2ALT}}, \frac{\varepsilon}{16CL}\right\}.$$

*Then*

$$T \geq \frac{48\Delta_0 L}{\varepsilon}\max\left\{\frac{8}{\alpha}, 4B, \frac{96\Delta_0 A}{\varepsilon}, \frac{16C}{\varepsilon}\right\} \quad \Rightarrow \quad \min_{0\leq t\leq T-1}\mathrm{E}\left[\|\nabla f(x^t)\|^2\right] \leq \varepsilon. \tag{20}$$

Note that by taking $A = C = 0$ and $B = 1$, one recovers the convergence of EF21-P in the nonconvex setting. Namely, under Assumptions 2.1 and 4.1, for $x^0 = w^0$ and $0 < \gamma \leq \frac{\alpha}{8L}$, we have $\min_{0\leq t\leq T-1}\mathrm{E}\left[\|\nabla f(x^t)\|^2\right] \leq \varepsilon$ as soon as $T \geq \frac{384\Delta_0 L}{\alpha\varepsilon}$.

We now apply the above result to the combination of EF21-P perturbation of the model and DCGD (Khaled & Richtárik, 2020) (EF21-P + DCGD). Suppose that the iterates follow the update (19) (see also Algorithm 2), where

$$g(x) = \frac{1}{n}\sum_{i=1}^n \mathcal{C}_i\,(g_i(x)) \tag{21}$$

and each stochastic gradient $g_i(x)$ is an unbiased estimator of the true gradient $\nabla f_i(x)$ (i.e., $\mathrm{E}\,[g_i(x)] = \nabla f_i(x)$).

**Proposition E.4.** *Suppose that the gradient estimator $g(x)$ is constructed via (21) and that Assumption 2.2 holds. Let $\Delta^* := \frac{1}{n}\sum_{i=1}^n(f^* - f_i^*)$. Then:*

1. *For $g_i(x) = \nabla f_i(x)$, Assumption E.2 is satisfied with $A = \frac{1}{n}\omega L_{max}$, $B = 1$ and $C = 2A\Delta^*$.*

2. *In the same setting as in part 1, assuming additionally that Assumption 4.3 holds, Assumption E.2 is satisfied with $A = C = 0$ and $B = \frac{D\omega}{n} + 1$.*

3. *Assume that each stochastic gradient $g_i$ has bounded variance, (i.e., $\mathrm{E}\left[\|g_i(x) - \nabla f_i(x)\|^2\right] \leq \sigma^2$). Then Assumption E.2 is satisfied with $A = \frac{1}{n}\omega L_{max}$, $B = 1$ and $C = 2A\Delta^* + \frac{\omega+1}{n}\sigma^2$.*

4. *Suppose that $\mathrm{E}\left[\|g_i(x) - \nabla f_i(x)\|^2\right] \leq \sigma^2$ and $f_i = f$ for all $i \in [n]$. Then Assumption E.2 is satisfied with $A = 0$, $B = \frac{\omega}{n} + 1$ and $C = \frac{\omega+1}{n}\sigma^2$.*

In Section 4, we apply Proposition E.4 and state the corresponding theorems.

### E.2 PROOF OF THE CONVERGENCE RESULT

We will need the following two lemmas:

**Lemma E.5.** *Consider sequences $(\delta^t)_t$, $(r^t)_t$ and $(s^t)_t$ such that $\delta^t, r^t, s^t \geq 0$ for all $t \geq 0$ and $s^0 = 0$. Suppose that*

$$\delta^{t+1} + as^{t+1} \leq b\delta^t + as^t - cr^t + d, \tag{22}$$

*where $a, b, c, d$ are non-negative constants and $b \geq 1$. Then for any $T \geq 1$*

$$\min_{0 \leq t \leq T-1} r^t \leq \frac{b^T}{cT}\delta^0 + \frac{d}{c}.$$

*Proof.* The proof follows similar steps as the proof of Lemma 2 of Khaled & Richtárik (2020) and we provide it for completeness. Let us fix $w_{-1} > 0$ and define $w_t = \frac{w_{t-1}}{b}$. Multiplying (22) by $w_t$ gives

$$w_t\delta^{t+1} + aw_ts^{t+1} \leq bw_t\delta^t + aw_ts^t - cw_tr^t + dw_t$$
$$\leq w_{t-1}\delta^t + aw_{t-1}s^t - cw_tr^t + dw_t.$$

Summing both sides of the inequality for $t = 0, \ldots, T-1$, we obtain

$$w_{T-1}\delta^T + aw_{T-1}s^T \leq w_{-1}\delta^0 + aw_{-1}s^0 - c\sum_{t=0}^{T-1} w_tr^t + d\sum_{t=0}^{T-1} w_t.$$

Rearranging and using the assumption that $s^0 = 0$ and non-negativity of $s^t$ gives

$$c\sum_{t=0}^{T-1} w_tr^t + w_{T-1}\delta^T \leq w_{-1}\delta^0 + aw_{-1}s^0 - aw_{T-1}s^T + d\sum_{t=0}^{T-1} w_t$$
$$\leq w_{-1}\delta^0 + d\sum_{t=0}^{T-1} w_t.$$

Next, using the non-negativity of $\delta^t$ and $w_t$, we have

$$c\sum_{t=0}^{T-1} w_tr^t \leq c\sum_{t=0}^{T-1} w_tr^t + w_{T-1}\delta^T \leq w_{-1}\delta^0 + d\sum_{t=0}^{T-1} w_t.$$

Letting $W_T := \sum_{t=0}^{T-1} w_t$ and dividing both sides of the inequality by $W_T$, we obtain

$$c\min_{0 \leq t \leq T-1} r^t \leq \frac{c}{W_T}\sum_{t=0}^{T-1} w_tr^t \leq \frac{w_{-1}}{W_T}\delta^0 + d.$$

Using the fact that

$$W_T = \sum_{t=0}^{T-1} w_t \geq \sum_{t=0}^{T-1} \min_{0 \leq t \leq T-1} w_t = Tw_{T-1} = \frac{Tw_{-1}}{b^T},$$

we can finish the proof. $\qquad\qquad\qquad\qquad\qquad\qquad\qquad\qquad\qquad\qquad\qquad\qquad\quad$ □

**Lemma E.6.** *Let Assumptions 2.1, 4.1, E.1 and E.2 hold, set $w^0 = x^0$, and choose*

$$\gamma \leq \min\left\{ \frac{1}{4A}, \frac{1}{4BL}, \frac{\alpha}{8L} \right\}.$$

*Then*

$$\min_{0 \leq t \leq T-1} \mathrm{E}\left[ \left\| \nabla f(x^t) \right\|^2 \right] \leq \frac{8 \left( 1 + 2AL\gamma^2 \right)^T}{\gamma T} \Delta_0 + 8CL\gamma. \tag{23}$$

*Proof.* First, $L$-smoothness of $f$ implies that

$$f(w^t) \leq f(x^t) + \langle \nabla f(x^t), w^t - x^t \rangle + \frac{L}{2} \left\| w^t - x^t \right\|^2$$
$$\overset{(9)}{\leq} f(x^t) + \frac{1}{2L} \left\| \nabla f(x^t) \right\|^2 + L \left\| w^t - x^t \right\|^2 \tag{24}$$

and

$$f(x^{t+1}) \leq f(x^t) + \langle \nabla f(x^t), x^{t+1} - x^t \rangle + \frac{L}{2} \| x^{t+1} - x^t \|^2$$
$$= f(x^t) - \gamma \langle \nabla f(x^t), g(w^t) \rangle + \frac{L\gamma^2}{2} \| g(w^t) \|^2.$$

Using the fact that $g(x)$ is an unbiased estimator of the true gradient, subtracting $f^*$ from both sides of the latter inequality and taking expectation given iterations $\{0, \ldots, t\}$, we obtain

$$\mathrm{E}_{t+1}\left[ f(x^{t+1}) - f^* \right] \leq f(x^t) - f^* - \gamma \langle \nabla f(x^t), \nabla f(w^t) \rangle + \frac{L\gamma^2}{2} \mathrm{E}_{t+1}\left[ \| g(w^t) \|^2 \right]$$

$$\overset{(\mathrm{E.2}),(12)}{\leq} f(x^t) - f^* - \frac{\gamma}{2} \| \nabla f(x^t) \|^2 - \frac{\gamma}{2} \| \nabla f(w^t) \|^2 + \frac{\gamma}{2} \| \nabla f(x^t) - \nabla f(w^t) \|^2$$
$$\qquad + \frac{L\gamma^2}{2} \left( 2A(f(w^t) - f^*) + B \| \nabla f(w^t) \|^2 + C \right)$$

$$\overset{(2.1)}{\leq} f(x^t) - f^* - \frac{\gamma}{2} \| \nabla f(x^t) \|^2 - \frac{\gamma}{2} \| \nabla f(w^t) \|^2 + \frac{\gamma L^2}{2} \| x^t - w^t \|^2$$
$$\qquad + AL\gamma^2 (f(w^t) - f^*) + \frac{BL\gamma^2}{2} \| \nabla f(w^t) \|^2 + \frac{CL\gamma^2}{2}$$

$$= f(x^t) - f^* - \frac{\gamma}{2} \| \nabla f(x^t) \|^2 - \frac{\gamma}{2} (1 - BL\gamma) \| \nabla f(w^t) \|^2 + \frac{L^2\gamma}{2} \| x^t - w^t \|^2$$
$$\qquad + AL\gamma^2 (f(w^t) - f^*) + \frac{CL\gamma^2}{2}$$

$$\overset{(24)}{\leq} f(x^t) - f^* - \frac{\gamma}{2} \| \nabla f(x^t) \|^2 - \frac{\gamma}{2} (1 - BL\gamma) \| \nabla f(w^t) \|^2 + \frac{L^2\gamma}{2} \| x^t - w^t \|^2$$
$$\qquad + AL\gamma^2 \left( f(x^t) + \frac{1}{2L} \| \nabla f(x^t) \|^2 + L \| w^t - x^t \|^2 - f^* \right) + \frac{CL\gamma^2}{2}$$

$$= (1 + AL\gamma^2) \left( f(x^t) - f^* \right) - \frac{\gamma}{2} (1 - A\gamma) \| \nabla f(x^t) \|^2 - \frac{\gamma}{2} (1 - BL\gamma) \| \nabla f(w^t) \|^2$$
$$\qquad + L^2\gamma \left( \frac{1}{2} + A\gamma \right) \| w^t - x^t \|^2 + \frac{CL\gamma^2}{2}.$$

Hence, taking full expectation, for $\gamma \leq \frac{1}{4A}$, we have

$$\mathrm{E}\left[ f(x^{t+1}) - f^* \right] \leq (1 + AL\gamma^2) \mathrm{E}\left[ f(x^t) - f^* \right] - \frac{\gamma}{4} \mathrm{E}\left[ \| \nabla f(x^t) \|^2 \right] \tag{25}$$

$$- \frac{\gamma}{2} (1 - BL\gamma) \, \mathrm{E} \left[ \left\| \nabla f(w^t) \right\|^2 \right] + L^2 \gamma \mathrm{E} \left[ \left\| w^t - x^t \right\|^2 \right] + \frac{CL\gamma^2}{2}.$$

Next, variance decomposition and Assumption E.2 gives

$$
\begin{aligned}
\mathrm{E} \left[ \left\| g(w^t) - \nabla f(w^t) \right\|^2 \right] &\overset{(14)}{=} \mathrm{E} \left[ \left\| g(w^t) \right\|^2 \right] - \left\| \nabla f(w^t) \right\|^2 \\
&\overset{(E.2)}{\leq} 2A(f(w^t) - f^*) + (B - 1) \left\| \nabla f(w^t) \right\|^2 + C \\
&\overset{(24)}{\leq} 2A \left( f(x^t) + \frac{1}{2L} \left\| \nabla f(x^t) \right\|^2 + L \left\| w^t - x^t \right\|^2 - f^* \right) \\
&\quad + (B - 1) \left\| \nabla f(w^t) \right\|^2 + C \\
&= 2A \left( f(x^t) - f^* \right) + \frac{A}{L} \left\| \nabla f(x^t) \right\|^2 + 2AL \left\| w^t - x^t \right\|^2 \\
&\quad + (B - 1) \left\| \nabla f(w^t) \right\|^2 + C.
\end{aligned}
\tag{26}
$$

Therefore, using the unbiasedness of $g(x)$, we can bound the expected distance between $w^{t+1}$ and $x^{t+1}$ as

$$
\begin{aligned}
\mathrm{E} \left[ \left\| w^{t+1} - x^{t+1} \right\|^2 \right] &= \mathrm{E} \left[ \left\| w^t + \mathcal{C}_p(x^{t+1} - w^t) - x^{t+1} \right\|^2 \right] \\
&\overset{(2)}{\leq} (1 - \alpha) \mathrm{E} \left[ \left\| x^{t+1} - w^t \right\|^2 \right] \\
&= (1 - \alpha) \, \mathrm{E} \left[ \left\| x^t - \gamma g^t - w^t \right\|^2 \right] \\
&\overset{(14)}{=} (1 - \alpha) \gamma^2 \mathrm{E} \left[ \left\| g^t - \nabla f(w^t) \right\|^2 \right] + (1 - \alpha) \, \mathrm{E} \left[ \left\| x^t - \gamma \nabla f(w^t) - w^t \right\|^2 \right] \\
&\overset{(7),(10),(11)}{\leq} (1 - \alpha) \gamma^2 \mathrm{E} \left[ \left\| g^t - \nabla f(w^t) \right\|^2 \right] + \left( 1 - \frac{\alpha}{2} \right) \mathrm{E} \left[ \left\| x^t - w^t \right\|^2 \right] \\
&\quad + \frac{2\gamma^2}{\alpha} \mathrm{E} \left[ \left\| \nabla f(w^t) \right\|^2 \right] \\
&\overset{(26)}{\leq} 2A (1 - \alpha) \gamma^2 \left( f(x^t) - f^* \right) + \frac{A (1 - \alpha) \gamma^2}{L} \left\| \nabla f(x^t) \right\|^2 \\
&\quad + 2AL (1 - \alpha) \gamma^2 \left\| w^t - x^t \right\|^2 + (B - 1) (1 - \alpha) \gamma^2 \left\| \nabla f(w^t) \right\|^2 \\
&\quad + C (1 - \alpha) \gamma^2 + \left( 1 - \frac{\alpha}{2} \right) \mathrm{E} \left[ \left\| x^t - w^t \right\|^2 \right] + \frac{2\gamma^2}{\alpha} \mathrm{E} \left[ \left\| \nabla f(w^t) \right\|^2 \right].
\end{aligned}
$$

Hence, taking expectation, for $\gamma \leq \sqrt{\frac{\alpha}{8AL(1-\alpha)}}$

$$
\begin{aligned}
\mathrm{E} \left[ \left\| w^{t+1} - x^{t+1} \right\|^2 \right] &\leq 2A (1 - \alpha) \gamma^2 \mathrm{E} \left[ f(x^t) - f^* \right] + \frac{A (1 - \alpha) \gamma^2}{L} \mathrm{E} \left[ \left\| \nabla f(x^t) \right\|^2 \right] \\
&\quad + \gamma^2 \left( \frac{2}{\alpha} + (B - 1) (1 - \alpha) \right) \mathrm{E} \left[ \left\| \nabla f(w^t) \right\|^2 \right] \\
&\quad + \left( 1 - \frac{\alpha}{2} + 2AL (1 - \alpha) \gamma^2 \right) \mathrm{E} \left[ \left\| x^t - w^t \right\|^2 \right] + C (1 - \alpha) \gamma^2 \\
&\leq 2A (1 - \alpha) \gamma^2 \mathrm{E} \left[ f(x^t) - f^* \right] + \frac{A (1 - \alpha) \gamma^2}{L} \mathrm{E} \left[ \left\| \nabla f(x^t) \right\|^2 \right] \\
&\quad + \gamma^2 \left( \frac{2}{\alpha} + (B - 1) (1 - \alpha) \right) \mathrm{E} \left[ \left\| \nabla f(w^t) \right\|^2 \right] \\
&\quad + \left( 1 - \frac{\alpha}{4} \right) \mathrm{E} \left[ \left\| x^t - w^t \right\|^2 \right] + C (1 - \alpha) \gamma^2.
\end{aligned}
\tag{27}
$$

Adding a $\frac{4L^2\gamma}{\alpha}$ multiple of (27) to (25), we obtain

$$\mathrm{E} \left[ f(x^{t+1}) - f^* \right] + \frac{4L^2\gamma}{\alpha} \mathrm{E} \left[ \left\| w^{t+1} - x^{t+1} \right\|^2 \right]$$

$$\leq \left(1 + AL\gamma^2 + \frac{8AL^2(1-\alpha)\gamma^3}{\alpha}\right) \mathrm{E}\left[f(x^t) - f^*\right]$$

$$- \frac{\gamma}{4}\left(1 - \frac{16AL\left(1-\alpha\right)\gamma^2}{\alpha}\right) \mathrm{E}\left[\left\|\nabla f(x^t)\right\|^2\right]$$

$$- \frac{\gamma}{2}\left(1 - BL\gamma - \frac{8L^2\gamma^2}{\alpha}\left(\frac{2}{\alpha} + (B-1)\left(1-\alpha\right)\right)\right) \mathrm{E}\left[\left\|\nabla f(w^t)\right\|^2\right]$$

$$+ \frac{4L^2\gamma}{\alpha}\mathrm{E}\left[\left\|w^t - x^t\right\|^2\right] + \frac{CL\gamma^2}{2} + \frac{4CL^2(1-\alpha)\gamma^3}{\alpha}.$$

Then, provided that

$$\gamma \leq \min\left\{\frac{1}{4BL}, \frac{\alpha}{8L}, \sqrt{\frac{\alpha}{32(B-1)(\alpha-1)L^2}}, \sqrt{\frac{\alpha}{32AL(1-\alpha)}},\right\}$$

$$= \min\left\{\frac{1}{4BL}, \frac{\alpha}{8L}, \sqrt{\frac{\alpha}{32AL(1-\alpha)}},\right\},$$

(where we used $\min\{a, b\} \leq \sqrt{ab}$ for all $a, b \in \mathbb{R}^+$), this gives

$$\mathrm{E}\left[f(x^{t+1}) - f^*\right] + \frac{4L^2\gamma}{\alpha}\mathrm{E}\left[\left\|w^{t+1} - x^{t+1}\right\|^2\right] \leq \left(1 + 2AL\gamma^2\right)\mathrm{E}\left[f(x^t) - f^*\right]$$

$$- \frac{\gamma}{8}\mathrm{E}\left[\left\|\nabla f(x^t)\right\|^2\right] + \frac{4L^2\gamma}{\alpha}\mathrm{E}\left[\left\|w^t - x^t\right\|^2\right] + CL\gamma^2.$$

Denoting $a := \frac{4L^2\gamma}{\alpha}$, $b := 1 + 2AL\gamma^2$, $c := \frac{\gamma}{8}$ and $d := CL\gamma^2$, this is equivalent to

$$\delta^{t+1} + as^{t+1} \leq b\delta^t + as^t - cr^t + d, \tag{28}$$

where $\delta^t := \mathrm{E}\left[f(x^t) - f^*\right]$, $r^t := \mathrm{E}\left[\left\|\nabla f(x^t)\right\|^2\right]$ and $s^t := \mathrm{E}\left[\left\|w^t - x^t\right\|^2\right]$. Hence, using Lemma E.5, for any $T \geq 1$

$$\min_{0 \leq t \leq T-1} r^t \leq \frac{b^T}{cT}\delta^0 + \frac{d}{c},$$

which proves (23). In the proof, we have the following constraints on $\gamma$:

$$\gamma \leq \min\left\{\frac{1}{4A}, \frac{1}{4BL}, \frac{\alpha}{8L}, \sqrt{\frac{\alpha}{32AL(1-\alpha)}}\right\}.$$

Using the inequality $\min\{a, b\} \leq \sqrt{ab}$ for all $a, b \in \mathbb{R}^+$, this can be simplified to

$$\gamma \leq \min\left\{\frac{1}{4A}, \frac{1}{4BL}, \frac{\alpha}{8L}\right\}.$$

$\square$

**Theorem E.3.** *Let Assumptions 2.1, 4.1, E.1 and E.2 hold and set $w^0 = x^0$. Fix $\varepsilon > 0$ and choose the stepsize*

$$\gamma = \min\left\{\frac{\alpha}{8L}, \frac{1}{4BL}, \frac{1}{\sqrt{2ALT}}, \frac{\varepsilon}{16CL}\right\}.$$

*Then*

$$T \geq \frac{48\Delta_0 L}{\varepsilon}\max\left\{\frac{8}{\alpha}, 4B, \frac{96\Delta_0 A}{\varepsilon}, \frac{16C}{\varepsilon}\right\} \quad \Rightarrow \quad \min_{0 \leq t \leq T-1}\mathrm{E}\left[\left\|\nabla f(x^t)\right\|^2\right] \leq \varepsilon. \tag{20}$$

*Proof.* By Lemma E.6, we have

$$\min_{0 \leq t \leq T-1}\mathrm{E}\left[\left\|\nabla f(x^t)\right\|^2\right] \leq \frac{8\left(1 + 2AL\gamma^2\right)^T}{\gamma T}\Delta_0 + 8CL\gamma$$

provided that $\gamma \leq \min\left\{\frac{1}{4A}, \frac{1}{4BL}, \frac{\alpha}{8L}\right\}$. Now, using the fact that $1 + x \leq e^x$ and the assumption $\gamma \leq \frac{1}{\sqrt{2ALT}}$, we obtain

$$\left(1 + 2AL\gamma^2\right)^T \leq \exp\left(2ALT\gamma^2\right) \leq \exp(1) < 3.$$

Hence

$$\min_{0 \leq t \leq T-1} \mathrm{E}\left[\left\|\nabla f(x^t)\right\|^2\right] \leq \frac{24}{\gamma T}\Delta_0 + 8CL\gamma.$$

In order to obtain $\frac{24}{\gamma T}\Delta_0 + 8CL\gamma \leq \varepsilon$, we require that both terms are no larger than $\frac{\varepsilon}{2}$, which is equivalent to

$$T \geq \frac{48\Delta_0}{\gamma\varepsilon}, \tag{29}$$

$$\gamma \leq \frac{\varepsilon}{16CL}. \tag{30}$$

We thus require that:

$$\gamma \leq \min\left\{\frac{1}{4A}, \frac{1}{4BL}, \frac{\alpha}{8L}, \frac{1}{\sqrt{2ALT}}, \frac{\varepsilon}{16CL}\right\}.$$

which, combined with (29) gives:

$$T \geq \frac{48\Delta_0}{\varepsilon}\max\left\{4A, 4BL, \frac{8L}{\alpha}, \frac{96\Delta_0 AL}{\varepsilon}, \frac{16CL}{\varepsilon}\right\}.$$

It remains to notice that the term $4A$ can be dropped, thus simplifying the constraints to

$$\gamma \leq \min\left\{\frac{1}{4BL}, \frac{\alpha}{8L}, \frac{1}{\sqrt{2ALT}}, \frac{\varepsilon}{16CL}\right\}.$$

and

$$T \geq \frac{48\Delta_0}{\varepsilon}\max\left\{4BL, \frac{8L}{\alpha}, \frac{96\Delta_0 AL}{\varepsilon}, \frac{16CL}{\varepsilon}\right\}.$$

Indeed, if $\left\|\nabla f(x^0)\right\|^2 \leq \varepsilon$, then (20) holds for any $\gamma > 0$. Let us now assume that $\left\|\nabla f(x^0)\right\|^2 > \varepsilon$. The above constraints imply that $\frac{1}{\sqrt{2ALT}} \leq \frac{\varepsilon}{96\Delta_0 AL}$. Moreover, from Lemma B.2, we know that $\varepsilon < \left\|\nabla f(x^0)\right\|^2 \leq 2L\Delta^0$. Thus $\frac{1}{\sqrt{2ALT}} \leq \frac{1}{48A}$. Similarly, we see that $\frac{96\Delta_0 AL}{\varepsilon} \geq 48A$. $\qquad\square$

### E.3 PROOF OF PROPOSITION E.4

*Proof.* 1. Using independence of $\mathcal{C}_1, \ldots, \mathcal{C}_n$, we have

$$\mathrm{E}\left[\|g(x)\|^2\right] = \mathrm{E}\left[\left\|\frac{1}{n}\sum_{i=1}^n \mathcal{C}_i\left(\nabla f_i(x)\right)\right\|^2\right]$$

$$\stackrel{(14)}{=} \mathrm{E}\left[\left\|\frac{1}{n}\sum_{i=1}^n \left(\mathcal{C}_i\left(\nabla f_i(x)\right) - \nabla f_i(x)\right)\right\|^2\right] + \|\nabla f(x)\|^2$$

$$= \frac{1}{n^2}\sum_{i=1}^n \mathrm{E}\left[\left\|\mathcal{C}_i\left(\nabla f_i(x)\right) - \nabla f_i(x)\right\|^2\right] + \|\nabla f(x)\|^2$$

$$\stackrel{(5)}{\leq} \frac{1}{n^2}\sum_{i=1}^n \omega\left\|\nabla f_i(x)\right\|^2 + \|\nabla f(x)\|^2$$

$$\stackrel{(B.2)}{\leq} \frac{\omega}{n^2}\sum_{i=1}^n 2L_i(f_i(x) - f_i^*) + \|\nabla f(x)\|^2$$

$$\leq \frac{2\omega L_{max}}{n^2} \sum_{i=1}^{n} (f_i(x) - f_i^*) + \|\nabla f(x)\|^2$$

$$= 2A(f(x) - f^*) + \|\nabla f(x)\|^2 + 2A\Delta^*,$$

where $A := \frac{\omega L_{max}}{n}$.

2. Starting as in part 1 of the proof, we obtain

$$\mathrm{E}\left[\|g(x)\|^2\right] \leq \frac{1}{n^2} \sum_{i=1}^{n} \omega \|\nabla f_i(x)\|^2 + \|\nabla f(x)\|^2 \overset{(4.3)}{\leq} \left(\frac{D\omega}{n} + 1\right) \|\nabla f(x)\|^2.$$

3. First let us note that

$$\mathrm{E}\left[\|g_i(x)\|^2\right] \overset{(14)}{=} \mathrm{E}\left[\|g_i(x) - \nabla f_i(x)\|^2\right] + \|\nabla f_i(x)\|^2 \leq \sigma^2 + \|\nabla f_i(x)\|^2.$$

Following steps similar to the proof of Proposition 4 of Khaled & Richtárik (2020), unbiasedness of the stochastic gradients gives

$$\mathrm{E}\left[\|g(x)\|^2\right] \overset{(13)}{=} \mathrm{E}\left[\mathrm{E}\left[\left\|\frac{1}{n}\sum_{i=1}^{n}\mathcal{C}_i(g_i(x))\right\|^2 \mid g_1(x), \dots, g_n(x)\right]\right]$$

$$\overset{(14)}{=} \mathrm{E}\left[\mathrm{E}\left[\left\|\frac{1}{n}\sum_{i=1}^{n}(\mathcal{C}_i(g_i(x)) - g_i(x))\right\|^2 \mid g_1(x), \dots, g_n(x)\right] + \left\|\frac{1}{n}\sum_{i=1}^{n}g_i(x)\right\|^2\right]$$

$$\overset{(14)}{=} \mathrm{E}\left[\frac{1}{n^2}\sum_{i=1}^{n}\mathrm{E}\left[\|\mathcal{C}_i(g_i(x)) - g_i(x)\|^2 \mid g_1(x), \dots, g_n(x)\right]\right]$$

$$+\mathrm{E}\left[\left\|\frac{1}{n}\sum_{i=1}^{n}(g_i(x) - \nabla f_i(x))\right\|^2\right] + \|\nabla f(x)\|^2$$

$$\leq \frac{\omega}{n^2}\sum_{i=1}^{n}\mathrm{E}\left[\|g_i(x)\|^2\right] + \mathrm{E}\left[\left\|\frac{1}{n}\sum_{i=1}^{n}(g_i(x) - \nabla f_i(x))\right\|^2\right] + \|\nabla f(x)\|^2$$

$$\leq \frac{\omega}{n^2}\sum_{i=1}^{n}\left(\|\nabla f_i(x)\|^2 + \sigma^2\right) + \frac{1}{n^2}\sum_{i=1}^{n}\mathrm{E}\left[\|g_i(x) - \nabla f_i(x)\|^2\right] + \|\nabla f(x)\|^2$$

$$\overset{(B.2)}{\leq} \frac{\omega}{n^2}\sum_{i=1}^{n}\left(2L_i(f_i(x) - f_i^*) + \sigma^2\right) + \frac{\sigma^2}{n} + \|\nabla f(x)\|^2$$

$$= 2A(f(x) - f^*) + \|\nabla f(x)\|^2 + C,$$

where $A := \frac{1}{n}\omega L_{max}$ and $C := 2A\Delta^* + \frac{\omega+1}{n}\sigma^2$.

4. Starting as in part 3 and using the assumption $f_i = f$, we have:

$$\mathrm{E}\left[\|g(x)\|^2\right] \leq \frac{\omega}{n^2}\sum_{i=1}^{n}\mathrm{E}\left[\|g_i(x)\|^2\right] + \mathrm{E}\left[\left\|\frac{1}{n}\sum_{i=1}^{n}g_i(x) - \nabla f(x)\right\|^2\right] + \|\nabla f(x)\|^2$$

$$\overset{(14)}{=} \frac{\omega}{n^2}\sum_{i=1}^{n}\left(\mathrm{E}\left[\|g_i(x) - \nabla f(x)\|^2\right] + \|\nabla f(x)\|^2\right)$$

$$+\frac{1}{n^2}\sum_{i=1}^{n}\mathrm{E}\left[\|g_i(x) - \nabla f(x)\|^2\right] + \|\nabla f(x)\|^2$$

$$\leq \frac{\omega+1}{n}\sigma^2 + \left(\frac{\omega}{n} + 1\right)\|\nabla f(x)\|^2.$$

$\square$

## F    PROOFS FOR EF21-P + DIANA IN THE CONVEX CASE

First, we prove an auxiliary theorem:

**Theorem F.1.** *Let us assume that Assumptions 2.1, 2.2 and 2.3 hold, $\beta \in \left[0, \frac{1}{\omega+1}\right]$, and*

$$\gamma \leq \min \left\{ \frac{n}{160\omega L_{\max}}, \frac{\sqrt{n\alpha}}{20\sqrt{\omega}\widehat{L}}, \frac{\alpha}{100L}, \frac{\beta}{\mu} \right\}. \tag{31}$$

*Then Algorithm 1 guarantees that*

$$\frac{1}{2\gamma}\mathrm{E}\left[\left\|x^{t+1} - x^*\right\|^2\right] + \mathrm{E}\left[f(x^{t+1}) - f(x^*)\right]$$

$$+ \kappa \frac{1}{n}\sum_{i=1}^{n}\mathrm{E}\left[\left\|h_i^{t+1} - \nabla f_i(x^*)\right\|^2\right] + \nu\mathrm{E}\left[\left\|w^{t+1} - x^{t+1}\right\|^2\right]$$

$$\leq \frac{1}{2\gamma}\left(1 - \frac{\gamma\mu}{2}\right)\mathrm{E}\left[\left\|x^t - x^*\right\|^2\right] + \frac{1}{2}\mathrm{E}\left[f(x^t) - f(x^*)\right]$$

$$+ \kappa\left(1 - \frac{\gamma\mu}{2}\right)\frac{1}{n}\sum_{i=1}^{n}\mathrm{E}\left[\left\|h_i^t - \nabla f_i(x^*)\right\|^2\right] + \nu\left(1 - \frac{\gamma\mu}{2}\right)\mathrm{E}\left[\left\|w^t - x^t\right\|^2\right], \tag{32}$$

*where $\kappa \leq \frac{8\gamma\omega}{n\beta}$ and $\nu \leq \frac{192\gamma\omega\widehat{L}^2}{n\alpha} + \frac{32L}{\alpha}$.*

*Proof.* From $L$-smoothness (Assumption 2.1) of the function $f$, we have

$$\begin{aligned} f(x^{t+1}) &\leq f(w^t) + \left\langle\nabla f(w^t), x^{t+1} - w^t\right\rangle + \frac{L}{2}\left\|x^{t+1} - w^t\right\|^2 \\ &\overset{\text{conv-ty}}{\leq} f(x^*) + \left\langle\nabla f(w^t), x^{t+1} - x^*\right\rangle - \frac{\mu}{2}\left\|w^t - x^*\right\|^2 + \frac{L}{2}\left\|x^{t+1} - w^t\right\|^2 \\ &= f(x^*) + \left\langle g^t, x^{t+1} - x^*\right\rangle + \left\langle\nabla f(w^t) - g^t, x^{t+1} - x^*\right\rangle \\ &\quad + \frac{L}{2}\left\|x^{t+1} - w^t\right\|^2 - \frac{\mu}{2}\left\|w^t - x^*\right\|^2. \end{aligned}$$

We now reprove a well-known equality from the convex world. Noting that $x^{t+1} = x^t - \gamma g^t$, we obtain

$$\begin{aligned} \left\|x^t - x^*\right\|^2 - &\left\|x^{t+1} - x^*\right\|^2 - \left\|x^{t+1} - x^t\right\|^2 \\ &= \left\langle x^t - x^{t+1}, x^t - 2x^* + x^{t+1}\right\rangle - \left\langle x^{t+1} - x^t, x^{t+1} - x^t\right\rangle \\ &= 2\left\langle x^t - x^{t+1}, x^{t+1} - x^*\right\rangle \\ &= 2\gamma\left\langle g^t, x^{t+1} - x^*\right\rangle. \end{aligned} \tag{33}$$

Substituting (33) in the inequality gives

$$\begin{aligned} f(x^{t+1}) &\leq f(x^*) + \left\langle\nabla f(w^t) - g^t, x^{t+1} - x^*\right\rangle \\ &\quad + \frac{1}{2\gamma}\left\|x^t - x^*\right\|^2 - \frac{1}{2\gamma}\left\|x^{t+1} - x^*\right\|^2 - \frac{1}{2\gamma}\left\|x^{t+1} - x^t\right\|^2 \\ &\quad + \frac{L}{2}\left\|x^{t+1} - w^t\right\|^2 - \frac{\mu}{2}\left\|w^t - x^*\right\|^2. \end{aligned}$$

Next, by (8), we have

$$\frac{L}{2}\left\|x^{t+1} - w^t\right\|^2 \leq L\left\|x^{t+1} - x^t\right\|^2 + L\left\|w^t - x^t\right\|^2$$

and

$$\frac{\mu}{4}\left\|x^t - x^*\right\|^2 \leq \frac{\mu}{2}\left\|w^t - x^*\right\|^2 + \frac{\mu}{2}\left\|w^t - x^t\right\|^2 \leq \frac{\mu}{2}\left\|w^t - x^*\right\|^2 + L\left\|w^t - x^t\right\|^2,$$

where we used $L \geq \mu$. Thus

$$
\begin{aligned}
f(x^{t+1}) \leq\ & f(x^*) + \left\langle \nabla f(w^t) - g^t, x^{t+1} - x^* \right\rangle \\
& + \frac{1}{2\gamma} \left\| x^t - x^* \right\|^2 - \frac{1}{2\gamma} \left\| x^{t+1} - x^* \right\|^2 - \frac{1}{2\gamma} \left\| x^{t+1} - x^t \right\|^2 \\
& + L \left\| x^{t+1} - x^t \right\|^2 + L \left\| w^t - x^t \right\|^2 - \frac{\mu}{4} \left\| x^t - x^* \right\|^2 + L \left\| w^t - x^t \right\|^2 \\
=\ & f(x^*) + \left\langle \nabla f(w^t) - g^t, x^{t+1} - x^* \right\rangle + \frac{1}{2\gamma} \left( 1 - \frac{\gamma\mu}{2} \right) \left\| x^t - x^* \right\|^2 \\
& - \frac{1}{2\gamma} \left\| x^{t+1} - x^* \right\|^2 - \left( \frac{1}{2\gamma} - L \right) \left\| x^{t+1} - x^t \right\|^2 + 2L \left\| w^t - x^t \right\|^2 \\
\leq\ & f(x^*) + \left\langle \nabla f(w^t) - g^t, x^{t+1} - x^* \right\rangle + \frac{1}{2\gamma} \left( 1 - \frac{\gamma\mu}{2} \right) \left\| x^t - x^* \right\|^2 \\
& - \frac{1}{2\gamma} \left\| x^{t+1} - x^* \right\|^2 + 2L \left\| w^t - x^t \right\|^2,
\end{aligned}
$$

where we used the fact that $\gamma \leq \frac{1}{2L}$. Then, taking expectation conditioned on previous iterations $\{0, \ldots, t\}$, we obtain

$$
\begin{aligned}
\mathrm{E}_{t+1} \left[ f(x^{t+1}) \right] \leq\ & f(x^*) + \mathrm{E}_{t+1} \left[ \left\langle \nabla f(w^t) - g^t, x^{t+1} - x^* \right\rangle \right] \\
& + \frac{1}{2\gamma} \left( 1 - \frac{\gamma\mu}{2} \right) \left\| x^t - x^* \right\|^2 - \frac{1}{2\gamma} \mathrm{E}_{t+1} \left[ \left\| x^{t+1} - x^* \right\|^2 \right] + 2L \left\| w^t - x^t \right\|^2.
\end{aligned}
$$

From the unbiasedness of the compressors $\mathcal{C}_i^D$, we have

$$
\mathrm{E}_{t+1} \left[ g^t \right] = \nabla f(w^t)
$$

and

$$
\begin{aligned}
\mathrm{E}_{t+1} \left[ \left\langle \nabla f(w^t) - g^t, x^{t+1} - x^* \right\rangle \right] &= \mathrm{E}_{t+1} \left[ \left\langle \nabla f(w^t) - g^t, x^t - \gamma g^t - x^* \right\rangle \right] \\
&= -\gamma \mathrm{E}_{t+1} \left[ \left\langle \nabla f(w^t) - g^t, g^t \right\rangle \right] \\
&= \gamma \mathrm{E}_{t+1} \left[ \left\| g^t \right\|^2 \right] - \gamma \left\| \nabla f(w^t) \right\|^2 \\
&\overset{(14)}{=} \gamma \mathrm{E}_{t+1} \left[ \left\| g^t - \nabla f(w^t) \right\|^2 \right].
\end{aligned}
$$

Therefore

$$
\begin{aligned}
\mathrm{E}_{t+1} \left[ f(x^{t+1}) \right] \leq\ & f(x^*) + \gamma \mathrm{E}_{t+1} \left[ \left\| g^t - \nabla f(w^t) \right\|^2 \right] \\
& + \frac{1}{2\gamma} \left( 1 - \frac{\gamma\mu}{2} \right) \left\| x^t - x^* \right\|^2 - \frac{1}{2\gamma} \mathrm{E}_{t+1} \left[ \left\| x^{t+1} - x^* \right\|^2 \right] + 2L \left\| w^t - x^t \right\|^2.
\end{aligned}
\tag{34}
$$

Now, we separately consider $\mathrm{E}_{t+1} \left[ \left\| g^t - \nabla f(w^t) \right\|^2 \right]$. From the independence of compressors, we have

$$
\begin{aligned}
& \mathrm{E}_{t+1} \left[ \left\| g^t - \nabla f(w^t) \right\|^2 \right] \\
&= \mathrm{E}_{t+1} \left[ \left\| h^t + \frac{1}{n} \sum_{i=1}^{n} \mathcal{C}_i^D (\nabla f_i(w^t) - h_i^t) - \nabla f(w^t) \right\|^2 \right] \\
&= \frac{1}{n^2} \sum_{i=1}^{n} \mathrm{E}_{t+1} \left[ \left\| \mathcal{C}_i^D (\nabla f_i(w^t) - h_i^t) - (\nabla f_i(w^t) - h_i^t) \right\|^2 \right] \\
&\leq \frac{\omega}{n^2} \sum_{i=1}^{n} \left\| \nabla f_i(w^t) - h_i^t \right\|^2 \\
&\leq \frac{2\omega}{n^2} \sum_{i=1}^{n} \left\| h_i^t - \nabla f_i(x^*) \right\|^2 + \frac{2\omega}{n^2} \sum_{i=1}^{n} \left\| \nabla f_i(w^t) - \nabla f_i(x^*) \right\|^2
\end{aligned}
$$

$$\leq \frac{2\omega}{n^2}\sum_{i=1}^{n}\left\|h_i^t - \nabla f_i(x^*)\right\|^2 + \frac{4\omega}{n^2}\sum_{i=1}^{n}\left\|\nabla f_i(w^t) - \nabla f_i(x^t)\right\|^2 + \frac{4\omega}{n^2}\sum_{i=1}^{n}\left\|\nabla f_i(x^t) - \nabla f_i(x^*)\right\|^2,$$

where in the last three inequalities, we used (5) and (8). Next, using Assumption 2.2 and Lemma B.1, we obtain

$$\mathrm{E}_{t+1}\left[\left\|g^t - \nabla f(w^t)\right\|^2\right]$$

$$\leq \frac{2\omega}{n^2}\sum_{i=1}^{n}\left\|h_i^t - \nabla f_i(x^*)\right\|^2 + \frac{4\omega\widehat{L}^2}{n}\left\|w^t - x^t\right\|^2 + \frac{8\omega L_{\max}}{n}\left(f(x^t) - f(x^*)\right). \quad (35)$$

To construct a Lyapunov function, it remains to bound $\frac{1}{n}\sum_{i=1}^{n}\left\|h_i^{t+1} - \nabla f_i(x^*)\right\|^2$ and $\left\|w^{t+1} - z^{t+1}\right\|^2$:

$$\frac{1}{n}\sum_{i=1}^{n}\mathrm{E}_{t+1}\left[\left\|h_i^{t+1} - \nabla f_i(x^*)\right\|^2\right]$$

$$= \frac{1}{n}\sum_{i=1}^{n}\mathrm{E}_{t+1}\left[\left\|h_i^t + \beta\mathcal{C}_i^D(\nabla f_i(w^t) - h_i^t) - \nabla f_i(x^*)\right\|^2\right]$$

$$= \frac{1}{n}\sum_{i=1}^{n}\left\|h_i^t - \nabla f_i(x^*)\right\|^2 + \frac{2\beta}{n}\sum_{i=1}^{n}\left\langle h_i^t - \nabla f_i(x^*), \mathrm{E}_{t+1}\left[\mathcal{C}_i^D(\nabla f_i(w^t) - h_i^t)\right]\right\rangle$$

$$+ \frac{\beta^2}{n}\sum_{i=1}^{n}\mathrm{E}_{t+1}\left[\left\|\mathcal{C}_i^D(\nabla f_i(w^t) - h_i^t)\right\|^2\right]$$

$$\stackrel{(5)}{\leq} \frac{1}{n}\sum_{i=1}^{n}\left\|h_i^t - \nabla f_i(x^*)\right\|^2 + \frac{2\beta}{n}\sum_{i=1}^{n}\left\langle h_i^t - \nabla f_i(x^*), \nabla f_i(w^t) - h_i^t\right\rangle$$

$$+ \frac{\beta^2(\omega + 1)}{n}\sum_{i=1}^{n}\left\|\nabla f_i(w^t) - h_i^t\right\|^2$$

$$\stackrel{(12)}{=} (1-\beta)\frac{1}{n}\sum_{i=1}^{n}\left\|h_i^t - \nabla f_i(x^*)\right\|^2 + \frac{\beta}{n}\sum_{i=1}^{n}\left\|\nabla f_i(w^t) - \nabla f_i(x^*)\right\|^2$$

$$+ \frac{\beta(\beta(\omega + 1) - 1)}{n}\sum_{i=1}^{n}\left\|\nabla f_i(w^t) - h_i^t\right\|^2$$

$$\leq (1-\beta)\frac{1}{n}\sum_{i=1}^{n}\left\|h_i^t - \nabla f_i(x^*)\right\|^2 + \frac{\beta}{n}\sum_{i=1}^{n}\left\|\nabla f_i(w^t) - \nabla f_i(x^*)\right\|^2,$$

where we use that $\beta \in \left[0, \frac{1}{\omega+1}\right]$. Thus, using (8), Assumption 2.2 and Lemma B.1, we have

$$\frac{1}{n}\sum_{i=1}^{n}\mathrm{E}_{t+1}\left[\left\|h_i^{t+1} - \nabla f_i(x^*)\right\|^2\right]$$

$$\leq (1-\beta)\frac{1}{n}\sum_{i=1}^{n}\left\|h_i^t - \nabla f_i(x^*)\right\|^2 + 2\beta\widehat{L}^2\left\|w^t - x^t\right\|^2 + 4\beta L_{\max}\left(f(x^t) - f(x^*)\right). \quad (36)$$

It remains to bound $\mathrm{E}_{t+1}\left[\left\|w^{t+1} - x^{t+1}\right\|^2\right]$:

$$\mathrm{E}_{t+1}\left[\left\|w^{t+1} - x^{t+1}\right\|^2\right] = \mathrm{E}_{t+1}\left[\left\|w^t + \mathcal{C}^p(x^{t+1} - w^t) - x^{t+1}\right\|^2\right]$$

$$\stackrel{(2)}{\leq} (1-\alpha)\mathrm{E}_{t+1}\left[\left\|x^{t+1} - w^t\right\|^2\right]$$

$$= (1-\alpha)\mathrm{E}_{t+1}\left[\left\|x^t - \gamma g^t - w^t\right\|^2\right]$$

$$\stackrel{(14)}{=} (1-\alpha)\gamma^2 \mathrm{E}_{t+1}\left[\left\|g^t - \nabla f(w^t)\right\|^2\right] + (1-\alpha)\left\|x^t - \gamma\nabla f(w^t) - w^t\right\|^2$$

$$\stackrel{(7)}{\leq} \gamma^2 \mathrm{E}_{t+1}\left[\left\|g^t - \nabla f(w^t)\right\|^2\right] + \left(1-\frac{\alpha}{2}\right)\left\|w^t - x^t\right\|^2 + \frac{2\gamma^2}{\alpha}\left\|\nabla f(w^t)\right\|^2$$

$$\stackrel{(8)}{\leq} \gamma^2 \mathrm{E}_{t+1}\left[\left\|g^t - \nabla f(w^t)\right\|^2\right] + \left(1-\frac{\alpha}{2}\right)\left\|w^t - x^t\right\|^2$$
$$+ \frac{4\gamma^2}{\alpha}\left\|\nabla f(w^t) - \nabla f(x^t)\right\|^2 + \frac{4\gamma^2}{\alpha}\left\|\nabla f(x^t) - \nabla f(x^*)\right\|^2.$$

Using Assumption 2.1 and Lemma B.1, we obtain

$$\mathrm{E}_{t+1}\left[\left\|w^{t+1} - x^{t+1}\right\|^2\right] \leq \gamma^2 \mathrm{E}_{t+1}\left[\left\|g^t - \nabla f(w^t)\right\|^2\right]$$
$$+ \left(1 - \frac{\alpha}{2} + \frac{4\gamma^2 L^2}{\alpha}\right)\left\|w^t - x^t\right\|^2 + \frac{8\gamma^2 L}{\alpha}\left(f(x^t) - f(x^*)\right)$$

$$\stackrel{(35)}{\leq} \gamma^2\left(\frac{2\omega}{n^2}\sum_{i=1}^n\left\|h_i^t - \nabla f_i(x^*)\right\|^2 + \frac{4\omega\widehat{L}^2}{n}\left\|w^t - x^t\right\|^2 + \frac{8\omega L_{\max}}{n}\left(f(x^t) - f(x^*)\right)\right)$$
$$+ \left(1 - \frac{\alpha}{2} + \frac{4\gamma^2 L^2}{\alpha}\right)\left\|w^t - x^t\right\|^2 + \frac{8\gamma^2 L}{\alpha}\left(f(x^t) - f(x^*)\right)$$

$$= \left(1 - \frac{\alpha}{2} + \frac{4\gamma^2 L^2}{\alpha} + \frac{4\gamma^2\omega\widehat{L}^2}{n}\right)\left\|w^t - x^t\right\|^2 + \frac{2\gamma^2\omega}{n^2}\sum_{i=1}^n\left\|h_i^t - \nabla f_i(x^*)\right\|^2$$
$$+ \left(\frac{8\gamma^2\omega L_{\max}}{n} + \frac{8\gamma^2 L}{\alpha}\right)\left(f(x^t) - f(x^*)\right)$$

$$\leq \left(1 - \frac{\alpha}{4}\right)\left\|w^t - x^t\right\|^2 + \frac{2\gamma^2\omega}{n^2}\sum_{i=1}^n\left\|h_i^t - \nabla f_i(x^*)\right\|^2$$
$$+ \left(\frac{8\gamma^2\omega L_{\max}}{n} + \frac{8\gamma^2 L}{\alpha}\right)\left(f(x^t) - f(x^*)\right),$$

where we assume that $\gamma \leq \frac{\alpha}{\sqrt{32}L}$ and $\gamma \leq \frac{\sqrt{\alpha n}}{\sqrt{32\omega}\widehat{L}}$.

Let us fix some constants $\kappa \geq 0$ and $\nu \geq 0$. We now combine the above inequality with (34), (35) and (36) to obtain

$$\mathrm{E}_{t+1}\left[f(x^{t+1})\right] + \kappa\frac{1}{n}\sum_{i=1}^n \mathrm{E}_{t+1}\left[\left\|h_i^{t+1} - \nabla f_i(x^*)\right\|^2\right] + \nu\mathrm{E}_{t+1}\left[\left\|w^{t+1} - x^{t+1}\right\|^2\right]$$

$$\leq f(x^*) + \gamma\left(\frac{2\omega}{n^2}\sum_{i=1}^n\left\|h_i^t - \nabla f_i(x^*)\right\|^2 + \frac{4\omega\widehat{L}^2}{n}\left\|w^t - x^t\right\|^2 + \frac{8\omega L_{\max}}{n}\left(f(x^t) - f(x^*)\right)\right)$$
$$+ \frac{1}{2\gamma}\left(1 - \frac{\gamma\mu}{2}\right)\left\|x^t - x^*\right\|^2 - \frac{1}{2\gamma}\mathrm{E}_{t+1}\left[\left\|x^{t+1} - x^*\right\|^2\right] + 2L\left\|w^t - x^t\right\|^2$$
$$+ \kappa\left((1-\beta)\frac{1}{n}\sum_{i=1}^n\left\|h_i^t - \nabla f_i(x^*)\right\|^2 + 2\beta\widehat{L}^2\left\|w^t - x^t\right\|^2 + 4\beta L_{\max}\left(f(x^t) - f(x^*)\right)\right)$$
$$+ \nu\left(\left(1 - \frac{\alpha}{4}\right)\left\|w^t - x^t\right\|^2 + \frac{2\gamma^2\omega}{n^2}\sum_{i=1}^n\left\|h_i^t - \nabla f_i(x^*)\right\|^2 + \left(\frac{8\gamma^2\omega L_{\max}}{n} + \frac{8\gamma^2 L}{\alpha}\right)\left(f(x^t) - f(x^*)\right)\right).$$

Rearranging the last inequality, one can get

$$\frac{1}{2\gamma}\mathrm{E}_{t+1}\left[\left\|x^{t+1} - x^*\right\|^2\right] + \mathrm{E}_{t+1}\left[f(x^{t+1}) - f(x^*)\right]$$

$$+ \kappa\frac{1}{n}\sum_{i=1}^n \mathrm{E}_{t+1}\left[\left\|h_i^{t+1} - \nabla f_i(x^*)\right\|^2\right] + \nu\mathrm{E}_{t+1}\left[\left\|w^{t+1} - x^{t+1}\right\|^2\right]$$

$$\leq \frac{1}{2\gamma} \left(1 - \frac{\gamma\mu}{2}\right) \|x^t - x^*\|^2$$

$$+ \left(\frac{8\gamma\omega L_{\max}}{n} + \kappa 4\beta L_{\max} + \nu \left(\frac{8\gamma^2\omega L_{\max}}{n} + \frac{8\gamma^2 L}{\alpha}\right)\right) (f(x^t) - f(x^*))$$

$$+ \left(\frac{2\gamma\omega}{n} + \nu \frac{2\gamma^2\omega}{n} + \kappa(1-\beta)\right) \frac{1}{n} \sum_{i=1}^{n} \|h_i^t - \nabla f_i(x^*)\|^2$$

$$+ \left(\frac{4\gamma\omega\widehat{L}^2}{n} + 2L + \kappa 2\beta\widehat{L}^2 + \nu\left(1 - \frac{\alpha}{4}\right)\right) \|w^t - x^t\|^2. \tag{37}$$

Our final goal is to find $\kappa$ and $\nu$ such that

$$\frac{2\gamma\omega}{n} + \nu \frac{2\gamma^2\omega}{n} + \kappa(1-\beta) = \kappa\left(1 - \frac{\beta}{2}\right)$$

and

$$\frac{4\gamma\omega\widehat{L}^2}{n} + 2L + \kappa 2\beta\widehat{L}^2 + \nu\left(1 - \frac{\alpha}{4}\right) \leq \nu\left(1 - \frac{\alpha}{8}\right).$$

The last inequality is equivalent to

$$\frac{32\gamma\omega\widehat{L}^2}{n\alpha} + \frac{16L}{\alpha} + \kappa \frac{16\beta\widehat{L}^2}{\alpha} \leq \nu. \tag{38}$$

From the first equality we get $\kappa = \frac{4\gamma\omega}{n\beta} + \nu\frac{4\gamma^2\omega}{n\beta}$. Thus

$$\frac{32\gamma\omega\widehat{L}^2}{n\alpha} + \frac{16L}{\alpha} + \kappa\frac{16\beta\widehat{L}^2}{\alpha} = \frac{32\gamma\omega\widehat{L}^2}{n\alpha} + \frac{16L}{\alpha} + \left(\frac{4\gamma\omega}{n\beta} + \nu\frac{4\gamma^2\omega}{n\beta}\right)\frac{16\beta\widehat{L}^2}{\alpha}$$

$$= \frac{96\gamma\omega\widehat{L}^2}{n\alpha} + \frac{16L}{\alpha} + \nu\frac{64\gamma^2\omega\widehat{L}^2}{n\alpha} \leq \frac{96\gamma\omega\widehat{L}^2}{n\alpha} + \frac{16L}{\alpha} + \nu\frac{1}{2},$$

where we used that $\gamma \leq \frac{\sqrt{n\alpha}}{\sqrt{128\omega}\widehat{L}}$. It means that we can take $\nu = \frac{192\gamma\omega\widehat{L}^2}{n\alpha} + \frac{32L}{\alpha}$ to ensure that (38) holds. Thus

$$\kappa = \frac{4\gamma\omega}{n\beta} + \left(\frac{192\gamma\omega\widehat{L}^2}{n\alpha} + \frac{32L}{\alpha}\right)\frac{4\gamma^2\omega}{n\beta} = \frac{4\gamma\omega}{n\beta} + \frac{768\gamma^3\omega^2\widehat{L}^2}{n^2\alpha\beta} + \frac{128\gamma^2\omega L}{n\beta\alpha}.$$

Let us now substitute these values of $\kappa$ and $\nu$ in inequality (37):

$$\frac{1}{2\gamma}\mathrm{E}_{t+1}\left[\|x^{t+1} - x^*\|^2\right] + \mathrm{E}_{t+1}\left[f(x^{t+1}) - f(x^*)\right]$$

$$+ \kappa\frac{1}{n}\sum_{i=1}^{n}\mathrm{E}_{t+1}\left[\|h_i^{t+1} - \nabla f_i(x^*)\|^2\right] + \nu\mathrm{E}_{t+1}\left[\|w^{t+1} - x^{t+1}\|^2\right]$$

$$\leq \frac{1}{2\gamma}\left(1 - \frac{\gamma\mu}{2}\right)\|x^t - x^*\|^2 + \kappa\left(1 - \frac{\beta}{2}\right)\frac{1}{n}\sum_{i=1}^{n}\|h_i^t - \nabla f_i(x^*)\|^2 + \nu\left(1 - \frac{\alpha}{8}\right)\|w^t - x^t\|^2$$

$$+ \left(\frac{8\gamma\omega L_{\max}}{n} + \left(\frac{4\gamma\omega}{n\beta} + \frac{768\gamma^3\omega^2\widehat{L}^2}{n^2\alpha\beta} + \frac{128\gamma^2\omega L}{n\beta\alpha}\right)4\beta L_{\max}\right.$$

$$\left. + \left(\frac{192\gamma\omega\widehat{L}^2}{n\alpha} + \frac{32L}{\alpha}\right)\left(\frac{8\gamma^2\omega L_{\max}}{n} + \frac{8\gamma^2 L}{\alpha}\right)\right)(f(x^t) - f(x^*))$$

$$= \frac{1}{2\gamma}\left(1 - \frac{\gamma\mu}{2}\right)\|x^t - x^*\|^2 + \kappa\left(1 - \frac{\beta}{2}\right)\frac{1}{n}\sum_{i=1}^{n}\|h_i^t - \nabla f_i(x^*)\|^2 + \nu\left(1 - \frac{\alpha}{8}\right)\|w^t - x^t\|^2$$

$$+ \left(\frac{24\gamma\omega L_{\max}}{n} + \frac{4608\gamma^3\omega^2\widehat{L}^2 L_{\max}}{n^2\alpha} + \frac{768\gamma^2\omega L L_{\max}}{n\alpha} + \frac{1536\gamma^3\omega L\widehat{L}^2}{n\alpha^2} + \frac{256\gamma^2 L^2}{\alpha^2}\right)(f(x^t) - f(x^*)).$$

Using the assumptions on $\gamma$, we have

$$\frac{24\gamma\omega L_{\max}}{n} \leq \frac{1}{10},$$

$$\frac{4608\gamma^3\omega^2\widehat{L}^2 L_{\max}}{n^2\alpha} \leq \frac{20\gamma^2\omega\widehat{L}^2}{n\alpha} \leq \frac{1}{10},$$

$$\frac{768\gamma^2\omega L L_{\max}}{n\alpha} \leq \frac{4\gamma L}{\alpha} \leq \frac{1}{10},$$

$$\frac{1536\gamma^3\omega L\widehat{L}^2}{n\alpha^2} \leq \frac{40\gamma^2\omega\widehat{L}^2}{n\alpha} \leq \frac{1}{10},$$

$$\frac{256\gamma^2 L^2}{\alpha^2} \leq \frac{1}{10}.$$

Finally, considering $\gamma \leq \frac{\beta}{\mu}$ and $\gamma \leq \frac{\alpha}{4\mu}$ gives

$$\frac{1}{2\gamma}\mathrm{E}_{t+1}\left[\left\|x^{t+1} - x^*\right\|^2\right] + \mathrm{E}_{t+1}\left[f(x^{t+1}) - f(x^*)\right]$$

$$+ \kappa\frac{1}{n}\sum_{i=1}^{n}\mathrm{E}_{t+1}\left[\left\|h_i^{t+1} - \nabla f_i(x^*)\right\|^2\right] + \nu\mathrm{E}_{t+1}\left[\left\|w^{t+1} - x^{t+1}\right\|^2\right]$$

$$\leq \frac{1}{2\gamma}\left(1 - \frac{\gamma\mu}{2}\right)\left\|x^t - x^*\right\|^2 + \kappa\left(1 - \frac{\gamma\mu}{2}\right)\frac{1}{n}\sum_{i=1}^{n}\left\|h_i^t - \nabla f_i(x^*)\right\|^2$$

$$+ \nu\left(1 - \frac{\gamma\mu}{2}\right)\left\|w^t - x^t\right\|^2 + \frac{1}{2}\left(f(x^t) - f(x^*)\right).$$

Note that $\kappa = \frac{4\gamma\omega}{n\beta} + \frac{768\gamma^3\omega^2\widehat{L}^2}{n^2\alpha\beta} + \frac{128\gamma^2\omega L}{n\beta\alpha} \leq \frac{8\gamma\omega}{n\beta}$. $\qquad\square$

We now prove a theorem for the general convex case:

**Theorem F.2.** *Let us assume that Assumptions 2.1, 2.2 and 2.3 hold, the strong convexity parameter satisfies $\mu = 0$, $\beta = \frac{1}{\omega+1}$, $x^0 = w^0$ and*

$$\gamma \leq \min\left\{\frac{n}{160\omega L_{\max}}, \frac{\sqrt{n\alpha}}{20\sqrt{\omega\widehat{L}}}, \frac{\alpha}{100L}\right\}.$$

*Then Algorithm 1 guarantees a convergence rate*

$$f\left(\frac{1}{T}\sum_{t=1}^{T}x^t\right) - f(x^*) \leq \frac{1}{\gamma T}\left\|x^0 - x^*\right\|^2 + \frac{f(x^0) - \nabla f(x^*)}{T} + \frac{16\gamma\omega(\omega+1)}{Tn^2}\sum_{i=1}^{n}\left\|h_i^0 - \nabla f_i(x^*)\right\|^2.$$

(39)

*Proof.* Let us bound (32):

$$\frac{1}{2\gamma}\mathrm{E}\left[\left\|x^{t+1} - x^*\right\|^2\right] + \mathrm{E}\left[f(x^{t+1}) - f(x^*)\right]$$

$$+ \kappa\frac{1}{n}\sum_{i=1}^{n}\mathrm{E}\left[\left\|h_i^{t+1} - \nabla f_i(x^*)\right\|^2\right] + \nu\mathrm{E}\left[\left\|w^{t+1} - x^{t+1}\right\|^2\right]$$

$$\leq \frac{1}{2\gamma}\left(1 - \frac{\gamma\mu}{2}\right)\mathrm{E}\left[\left\|x^t - x^*\right\|^2\right] + \frac{1}{2}\mathrm{E}\left[f(x^t) - f(x^*)\right]$$

$$+ \kappa\left(1 - \frac{\gamma\mu}{2}\right)\frac{1}{n}\sum_{i=1}^{n}\mathrm{E}\left[\left\|h_i^t - \nabla f_i(x^*)\right\|^2\right] + \nu\left(1 - \frac{\gamma\mu}{2}\right)\mathrm{E}\left[\left\|w^t - x^t\right\|^2\right]$$

$$\leq \frac{1}{2\gamma}\mathrm{E}\left[\left\|x^t - x^*\right\|^2\right] + \frac{1}{2}\mathrm{E}\left[f(x^t) - f(x^*)\right] + \kappa\frac{1}{n}\sum_{i=1}^{n}\mathrm{E}\left[\left\|h_i^t - \nabla f_i(x^*)\right\|^2\right] + \nu\mathrm{E}\left[\left\|w^t - x^t\right\|^2\right].$$

We now sum the inequality for $t \in \{0, \ldots, T-1\}$ and obtain

$$
\frac{1}{2\gamma}\mathrm{E}\left[\left\|x^T - x^*\right\|^2\right] + \frac{1}{2}\mathrm{E}\left[f(x^T) - f(x^*)\right] + \frac{1}{2}\sum_{t=1}^{T}\mathrm{E}\left[f(x^t) - f(x^*)\right]
$$

$$
+ \kappa\frac{1}{n}\sum_{i=1}^{n}\mathrm{E}\left[\left\|h_i^T - \nabla f_i(x^*)\right\|^2\right] + \nu\mathrm{E}\left[\left\|w^T - x^T\right\|^2\right]
$$

$$
\leq \frac{1}{2\gamma}\left\|x^0 - x^*\right\|^2 + \frac{1}{2}\left(f(x^0) - f(x^*)\right) + \kappa\frac{1}{n}\sum_{i=1}^{n}\left\|h_i^0 - \nabla f_i(x^*)\right\|^2 + \nu\left\|w^0 - x^0\right\|^2
$$

$$
\leq \frac{1}{2\gamma}\left\|x^0 - x^*\right\|^2 + \frac{1}{2}\left(f(x^0) - f(x^*)\right) + \frac{8\gamma\omega}{n^2\beta}\sum_{i=1}^{n}\left\|h_i^0 - \nabla f_i(x^*)\right\|^2,
$$

where we used the assumption $x^0 = w^0$ and the bound on $\kappa$. Using nonnegativity of the terms and convexity, we then have

$$
f\left(\frac{1}{T}\sum_{t=1}^{T}x^t\right) - f(x^*) \leq \frac{1}{\gamma T}\left\|x^0 - x^*\right\|^2 + \frac{f(x^0) - \nabla f(x^*)}{T} + \frac{16\gamma\omega}{Tn^2\beta}\sum_{i=1}^{n}\left\|h_i^0 - \nabla f_i(x^*)\right\|^2.
$$

$\square$

We now prove a theorem for the strongly convex case:

**Theorem 3.1.** *Suppose that Assumptions 2.1, 2.2 and 2.3 hold, $\beta = \frac{1}{\omega+1}$, set $x^0 = w^0$ and let $\gamma \leq \min\left\{\frac{n}{160\omega L_{\max}}, \frac{\sqrt{n\alpha}}{20\sqrt{\omega}\widehat{L}}, \frac{\alpha}{100L}, \frac{1}{(\omega+1)\mu}\right\}$. Then Algorithm 1 returns $x^T$ such that*

$$
\tfrac{1}{2\gamma}\mathrm{E}\left[\left\|x^T - x^*\right\|^2\right] + \mathrm{E}\left[f(x^T) - f(x^*)\right] \leq \left(1 - \tfrac{\gamma\mu}{2}\right)^T V^0,
$$

*where $V^0 := \frac{1}{2\gamma}\mathrm{E}\left[\left\|x^0 - x^*\right\|^2\right] + \left(f(x^0) - f(x^*)\right) + \frac{8\gamma\omega(\omega+1)}{n^2}\sum_{i=1}^{n}\left\|h_i^0 - \nabla f_i(x^*)\right\|^2.$*

*Proof.* Using $\gamma \leq \frac{\alpha}{100L} \leq \frac{1}{\mu}$, let us bound (32):

$$
\frac{1}{2\gamma}\mathrm{E}\left[\left\|x^{t+1} - x^*\right\|^2\right] + \mathrm{E}\left[f(x^{t+1}) - f(x^*)\right]
$$

$$
+ \kappa\frac{1}{n}\sum_{i=1}^{n}\mathrm{E}\left[\left\|h_i^{t+1} - \nabla f_i(x^*)\right\|^2\right] + \nu\mathrm{E}\left[\left\|w^{t+1} - x^{t+1}\right\|^2\right]
$$

$$
\leq \frac{1}{2\gamma}\left(1 - \frac{\gamma\mu}{2}\right)\mathrm{E}\left[\left\|x^t - x^*\right\|^2\right] + \frac{1}{2}\mathrm{E}\left[f(x^t) - f(x^*)\right]
$$

$$
+ \kappa\left(1 - \frac{\gamma\mu}{2}\right)\frac{1}{n}\sum_{i=1}^{n}\mathrm{E}\left[\left\|h_i^t - \nabla f_i(x^*)\right\|^2\right] + \nu\left(1 - \frac{\gamma\mu}{2}\right)\mathrm{E}\left[\left\|w^t - x^t\right\|^2\right]
$$

$$
\leq \frac{1}{2\gamma}\left(1 - \frac{\gamma\mu}{2}\right)\mathrm{E}\left[\left\|x^t - x^*\right\|^2\right] + \left(1 - \frac{\gamma\mu}{2}\right)\mathrm{E}\left[f(x^t) - f(x^*)\right]
$$

$$
+ \kappa\left(1 - \frac{\gamma\mu}{2}\right)\frac{1}{n}\sum_{i=1}^{n}\mathrm{E}\left[\left\|h_i^t - \nabla f_i(x^*)\right\|^2\right] + \nu\left(1 - \frac{\gamma\mu}{2}\right)\mathrm{E}\left[\left\|w^t - x^t\right\|^2\right]
$$

$$
= \left(1 - \frac{\gamma\mu}{2}\right)\left(\frac{1}{2\gamma}\mathrm{E}\left[\left\|x^t - x^*\right\|^2\right] + \mathrm{E}\left[f(x^t) - f(x^*)\right] + \kappa\frac{1}{n}\sum_{i=1}^{n}\mathrm{E}\left[\left\|h_i^t - \nabla f_i(x^*)\right\|^2\right] + \nu\mathrm{E}\left[\left\|w^t - x^t\right\|^2\right]\right).
$$

Recursively applying the last inequality and using $x^0 = w^0$, one can get that

$$
\frac{1}{2\gamma}\mathrm{E}\left[\left\|x^T - x^*\right\|^2\right] + \mathrm{E}\left[f(x^T) - f(x^*)\right] + \kappa\frac{1}{n}\sum_{i=1}^{n}\mathrm{E}\left[\left\|h_i^T - \nabla f_i(x^*)\right\|^2\right] + \nu\mathrm{E}\left[\left\|w^T - x^T\right\|^2\right]
$$

$$\leq \left(1 - \frac{\gamma\mu}{2}\right)^T \left(\frac{1}{2\gamma}\mathrm{E}\left[\left\|x^0 - x^*\right\|^2\right] + (f(x^0) - f(x^*)) + \kappa\frac{1}{n}\sum_{i=1}^n \left\|h_i^0 - \nabla f_i(x^*)\right\|^2\right).$$

Using the nonnegativity of the terms and the bound on $\kappa$, we obtain

$$\frac{1}{2\gamma}\mathrm{E}\left[\left\|x^T - x^*\right\|^2\right] + \mathrm{E}\left[f(x^T) - f(x^*)\right]$$

$$\leq \left(1 - \frac{\gamma\mu}{2}\right)^T \left(\frac{1}{2\gamma}\mathrm{E}\left[\left\|x^0 - x^*\right\|^2\right] + (f(x^0) - f(x^*)) + \frac{8\gamma\omega}{n^2\beta}\sum_{i=1}^n \left\|h_i^0 - \nabla f_i(x^*)\right\|^2\right).$$

$\square$

### F.1 Communication Complexities in the General Convex Case

We now derive the communication complexities for the general convex case. From Theorem F.2, we know that EF21-P + DIANA has the following convergence rate:

$$f\left(\frac{1}{T}\sum_{t=1}^T x^t\right) - f(x^*) \leq \frac{1}{\gamma T}\left\|x^0 - x^*\right\|^2 + \frac{f(x^0) - \nabla f(x^*)}{T} + \frac{16\gamma\omega(\omega+1)}{Tn^2}\sum_{i=1}^n \left\|h_i^0 - \nabla f_i(x^*)\right\|^2.$$

Let us take $h_i^0 = \nabla f_i(x^0)$ for all $i \in [n]$. Using Assumptions 2.1 and 2.2, we have

$$f\left(\frac{1}{T}\sum_{t=1}^T x^t\right) - f(x^*) \leq \frac{1}{\gamma T}\left\|x^0 - x^*\right\|^2 + \frac{L\left\|x^0 - x^*\right\|^2}{2T}$$

$$+ \frac{16\gamma\omega(\omega+1)}{Tn^2}\sum_{i=1}^n \left\|\nabla f_i(x^0) - \nabla f_i(x^*)\right\|^2$$

$$\leq \frac{1}{\gamma T}\left\|x^0 - x^*\right\|^2 + \frac{L\left\|x^0 - x^*\right\|^2}{2T} + \frac{16\gamma\omega(\omega+1)\widehat{L}^2\left\|x^0 - x^*\right\|^2}{Tn}.$$

Using the bound on $\gamma$, we obtain that EF21-P + DIANA returns an $\varepsilon$-solution after

$$\mathcal{O}\left(\frac{\omega L_{\max}}{n\varepsilon} + \frac{\sqrt{\omega}\widehat{L}}{\sqrt{n\alpha}\varepsilon} + \frac{L}{\alpha\varepsilon} + \frac{L}{\varepsilon} + \frac{\gamma\omega(\omega+1)\widehat{L}^2}{n\varepsilon}\right)$$

steps. For simplicity, we assume that the server and the workers use $\mathrm{Top}K$ and $\mathrm{Rand}K$ compressors, respectively. Thus the server-to-workers and the workers-to-server communication complexities equal

$$\mathcal{O}\left(K \times \left(\frac{\omega L_{\max}}{n\varepsilon} + \frac{\sqrt{\omega}\widehat{L}}{\sqrt{n\alpha}\varepsilon} + \frac{L}{\alpha\varepsilon} + \frac{L}{\varepsilon} + \frac{\gamma\omega(\omega+1)\widehat{L}^2}{n\varepsilon}\right)\right)$$

$$= \mathcal{O}\left(\frac{dL_{\max}}{n\varepsilon} + \frac{d\widehat{L}}{\sqrt{n}\varepsilon} + \frac{dL}{\varepsilon} + \frac{KL}{\varepsilon} + \frac{d\gamma\omega\widehat{L}^2}{n\varepsilon}\right).$$

Note that $\gamma \leq \frac{\sqrt{n\alpha}}{20\sqrt{\omega}\widehat{L}} = \frac{\sqrt{n}}{20\sqrt{\omega(\omega+1)}\widehat{L}}$. Thus

$$\mathcal{O}\left(K \times \left(\frac{\omega L_{\max}}{n\varepsilon} + \frac{\sqrt{\omega}\widehat{L}}{\sqrt{n\alpha}\varepsilon} + \frac{L}{\alpha\varepsilon} + \frac{L}{\varepsilon} + \frac{\gamma\omega(\omega+1)\widehat{L}^2}{n\varepsilon}\right)\right)$$

$$= \mathcal{O}\left(\frac{dL_{\max}}{n\varepsilon} + \frac{d\widehat{L}}{\sqrt{n}\varepsilon} + \frac{dL}{\varepsilon} + \frac{KL}{\varepsilon} + \frac{d\widehat{L}}{\sqrt{n}\varepsilon}\right)$$

$$= \mathcal{O}\left(\frac{dL_{\max}}{n\varepsilon} + \frac{d\widehat{L}}{\sqrt{n}\varepsilon} + \frac{dL}{\varepsilon}\right).$$

Since $L_{\max} \leq nL$ and $\widehat{L} \leq \sqrt{n}L$, this complexity is not worse than the GD's complexity $\mathcal{O}\left(\frac{dL}{\varepsilon}\right)$ for any $K \in [1, d]$.

### F.2 PROOFS FOR EF21-P + DIANA WITH STOCHASTIC GRADIENTS

First, we prove the following auxiliary theorem:

**Theorem F.3.** *Let us consider Algorithm 1 using the stochastic gradients $\widetilde{\nabla} f_i$ instead of the exact gradients $\nabla f_i$ for all $i \in [n]$. Assume that Assumptions 2.1, 2.2, 2.3 and 3.2 hold, $\beta \in \left[0, \frac{1}{\omega+1}\right]$, and*

$$\gamma \leq \min\left\{ \frac{n}{160\omega L_{\max}}, \frac{\sqrt{n\alpha}}{20\sqrt{\omega}\widehat{L}}, \frac{\alpha}{100L}, \frac{\beta}{\mu} \right\}.$$

*Then Algorithm 1 guarantees that*

$$\frac{1}{2\gamma}\mathrm{E}\left[\left\|x^{t+1} - x^*\right\|^2\right] + \mathrm{E}\left[f(x^{t+1}) - f(x^*)\right]$$

$$+ \kappa\frac{1}{n}\sum_{i=1}^{n}\mathrm{E}\left[\left\|h_i^{t+1} - \nabla f_i(x^*)\right\|^2\right] + \nu\mathrm{E}\left[\left\|w^{t+1} - x^{t+1}\right\|^2\right]$$

$$\leq \frac{1}{2\gamma}\left(1 - \frac{\gamma\mu}{2}\right)\mathrm{E}\left[\left\|x^t - x^*\right\|^2\right] + \frac{1}{2}\mathrm{E}\left[f(x^t) - f(x^*)\right]$$

$$+ \kappa\left(1 - \frac{\gamma\mu}{2}\right)\frac{1}{n}\sum_{i=1}^{n}\mathrm{E}\left[\left\|h_i^t - \nabla f_i(x^*)\right\|^2\right] + \nu\left(1 - \frac{\gamma\mu}{2}\right)\mathrm{E}\left[\left\|w^t - x^t\right\|^2\right] + \frac{12\gamma(\omega+1)\sigma^2}{n}, \tag{40}$$

*where $\kappa \leq \frac{8\gamma\omega}{n\beta}$ and $\nu \leq \frac{192\gamma\omega\widehat{L}^2}{n\alpha} + \frac{32L}{\alpha}$.*

*Proof.* First, we bound $\mathrm{E}_{t+1}\left[\left\|g^t - \nabla f(w^t)\right\|^2\right]$, $\frac{1}{n}\sum_{i=1}^{n}\mathrm{E}_{t+1}\left[\left\|h_i^{t+1} - \nabla f_i(x^*)\right\|^2\right]$ and $\mathrm{E}_{t+1}\left[\left\|w^{t+1} - x^{t+1}\right\|^2\right]$. Using the independence of compressors, we have

$$\mathrm{E}_{t+1}\left[\left\|g^t - \nabla f(w^t)\right\|^2\right]$$

$$= \mathrm{E}_{t+1}\left[\left\|h^t + \frac{1}{n}\sum_{i=1}^{n}\mathcal{C}_i^D(\widetilde{\nabla} f_i(w^t) - h_i^t) - \nabla f(w^t)\right\|^2\right]$$

$$= \frac{1}{n^2}\sum_{i=1}^{n}\mathrm{E}_{t+1}\left[\left\|\mathcal{C}_i^D(\widetilde{\nabla} f_i(w^t) - h_i^t) - \left(\nabla f_i(w^t) - h_i^t\right)\right\|^2\right]$$

$$\overset{(14)}{=} \frac{1}{n^2}\sum_{i=1}^{n}\left(\mathrm{E}_{t+1}\left[\left\|\mathcal{C}_i^D(\widetilde{\nabla} f_i(w^t) - h_i^t) - \left(\widetilde{\nabla} f_i(w^t) - h_i^t\right)\right\|^2\right] + \mathrm{E}_{t+1}\left[\left\|\widetilde{\nabla} f_i(w^t) - \nabla f_i(w^t)\right\|^2\right]\right)$$

$$\leq \frac{\omega}{n^2}\sum_{i=1}^{n}\mathrm{E}_{t+1}\left[\left\|\widetilde{\nabla} f_i(w^t) - h_i^t\right\|^2\right] + \frac{1}{n^2}\sum_{i=1}^{n}\mathrm{E}_{t+1}\left[\left\|\widetilde{\nabla} f_i(w^t) - \nabla f_i(w^t)\right\|^2\right]$$

$$\overset{(14)}{=} \frac{\omega}{n^2}\sum_{i=1}^{n}\left\|\nabla f_i(w^t) - h_i^t\right\|^2 + \frac{\omega+1}{n^2}\sum_{i=1}^{n}\mathrm{E}_{t+1}\left[\left\|\widetilde{\nabla} f_i(w^t) - \nabla f_i(w^t)\right\|^2\right]$$

$$\leq \frac{\omega}{n^2}\sum_{i=1}^{n}\left\|\nabla f_i(w^t) - h_i^t\right\|^2 + \frac{(\omega+1)\sigma^2}{n}$$

$$\leq \frac{2\omega}{n^2}\sum_{i=1}^{n}\left\|h_i^t - \nabla f_i(x^*)\right\|^2 + \frac{2\omega}{n^2}\sum_{i=1}^{n}\left\|\nabla f_i(w^t) - \nabla f_i(x^*)\right\|^2 + \frac{(\omega+1)\sigma^2}{n}$$

$$\leq \frac{2\omega}{n^2}\sum_{i=1}^{n}\left\|h_i^t - \nabla f_i(x^*)\right\|^2 + \frac{4\omega}{n^2}\sum_{i=1}^{n}\left\|\nabla f_i(w^t) - \nabla f_i(x^t)\right\|^2$$

$$+ \frac{4\omega}{n^2}\sum_{i=1}^{n}\left\|\nabla f_i(x^t) - \nabla f_i(x^*)\right\|^2 + \frac{(\omega+1)\sigma^2}{n},$$

where in the last three inequalities, we used (5) and (8). Using Assumption 2.2 and Lemma B.1, we obtain

$$
\mathrm{E}_{t+1}\left[\left\|g^t - \nabla f(w^t)\right\|^2\right]
$$

$$
\leq \frac{2\omega}{n^2}\sum_{i=1}^{n}\left\|h_i^t - \nabla f_i(x^*)\right\|^2 + \frac{4\omega\widehat{L}^2}{n}\left\|w^t - x^t\right\|^2 + \frac{8\omega L_{\max}}{n}\left(f(x^t) - f(x^*)\right) + \frac{(\omega+1)\sigma^2}{n}.
$$

Next, we bound $\frac{1}{n}\sum_{i=1}^{n}\left\|h_i^{t+1} - \nabla f_i(x^*)\right\|^2$ to construct a Lyapunov function:

$$
\frac{1}{n}\sum_{i=1}^{n}\mathrm{E}_{t+1}\left[\left\|h_i^{t+1} - \nabla f_i(x^*)\right\|^2\right]
$$

$$
= \frac{1}{n}\sum_{i=1}^{n}\mathrm{E}_{t+1}\left[\left\|h_i^t + \beta\mathcal{C}_i^D(\widetilde{\nabla}f_i(w^t) - h_i^t) - \nabla f_i(x^*)\right\|^2\right]
$$

$$
= \frac{1}{n}\sum_{i=1}^{n}\left\|h_i^t - \nabla f_i(x^*)\right\|^2 + \frac{2\beta}{n}\sum_{i=1}^{n}\left\langle h_i^t - \nabla f_i(x^*), \mathrm{E}_{t+1}\left[\mathcal{C}_i^D(\widetilde{\nabla}f_i(w^t) - h_i^t)\right]\right\rangle
$$

$$
+ \frac{\beta^2}{n}\sum_{i=1}^{n}\mathrm{E}_{t+1}\left[\left\|\mathcal{C}_i^D(\widetilde{\nabla}f_i(w^t) - h_i^t)\right\|^2\right]
$$

$$
\overset{(5)}{\leq} \frac{1}{n}\sum_{i=1}^{n}\left\|h_i^t - \nabla f_i(x^*)\right\|^2 + \frac{2\beta}{n}\sum_{i=1}^{n}\left\langle h_i^t - \nabla f_i(x^*), \nabla f_i(w^t) - h_i^t\right\rangle
$$

$$
+ \frac{\beta^2(\omega+1)}{n}\sum_{i=1}^{n}\mathrm{E}_{t+1}\left[\left\|\widetilde{\nabla}f_i(w^t) - h_i^t\right\|^2\right]
$$

$$
\overset{(14)}{=} \frac{1}{n}\sum_{i=1}^{n}\left\|h_i^t - \nabla f_i(x^*)\right\|^2 + \frac{2\beta}{n}\sum_{i=1}^{n}\left\langle h_i^t - \nabla f_i(x^*), \nabla f_i(w^t) - h_i^t\right\rangle
$$

$$
+ \frac{\beta^2(\omega+1)}{n}\sum_{i=1}^{n}\mathrm{E}_{t+1}\left[\left\|\nabla f_i(w^t) - h_i^t\right\|^2\right] + \frac{\beta^2(\omega+1)}{n}\sum_{i=1}^{n}\mathrm{E}_{t+1}\left[\left\|\widetilde{\nabla}f_i(w^t) - \nabla f_i(w^t)\right\|^2\right]
$$

$$
\leq \frac{1}{n}\sum_{i=1}^{n}\left\|h_i^t - \nabla f_i(x^*)\right\|^2 + \frac{2\beta}{n}\sum_{i=1}^{n}\left\langle h_i^t - \nabla f_i(x^*), \nabla f_i(w^t) - h_i^t\right\rangle
$$

$$
+ \frac{\beta^2(\omega+1)}{n}\sum_{i=1}^{n}\mathrm{E}_{t+1}\left[\left\|\nabla f_i(w^t) - h_i^t\right\|^2\right] + \beta^2(\omega+1)\sigma^2
$$

$$
\overset{(12)}{=} (1-\beta)\frac{1}{n}\sum_{i=1}^{n}\left\|h_i^t - \nabla f_i(x^*)\right\|^2 + \frac{\beta}{n}\sum_{i=1}^{n}\left\|\nabla f_i(w^t) - \nabla f_i(x^*)\right\|^2
$$

$$
+ \frac{\beta(\beta(\omega+1)-1)}{n}\sum_{i=1}^{n}\left\|\nabla f_i(w^t) - h_i^t\right\|^2 + \beta^2(\omega+1)\sigma^2
$$

$$
\leq (1-\beta)\frac{1}{n}\sum_{i=1}^{n}\left\|h_i^t - \nabla f_i(x^*)\right\|^2 + \frac{\beta}{n}\sum_{i=1}^{n}\left\|\nabla f_i(w^t) - \nabla f_i(x^*)\right\|^2 + \beta^2(\omega+1)\sigma^2,
$$

where we use the assumption $\beta \in \left[0, \frac{1}{\omega+1}\right]$. Using (8), Assumption 2.2 and Lemma B.1, we have

$$
\frac{1}{n}\sum_{i=1}^{n}\mathrm{E}_{t+1}\left[\left\|h_i^{t+1} - \nabla f_i(x^*)\right\|^2\right] \leq (1-\beta)\frac{1}{n}\sum_{i=1}^{n}\left\|h_i^t - \nabla f_i(x^*)\right\|^2
$$

$$
+ 2\beta\widehat{L}^2\left\|w^t - x^t\right\|^2 + 4\beta L_{\max}\left(f(x^t) - f(x^*)\right) + \beta^2(\omega+1)\sigma^2.
$$

It remains to bound $\mathrm{E}_{t+1}\left[\left\|w^{t+1} - x^{t+1}\right\|^2\right]$ :

$$
\mathrm{E}_{t+1}\left[\left\|w^{t+1} - x^{t+1}\right\|^2\right] = \mathrm{E}_{t+1}\left[\left\|w^t + \mathcal{C}^p(x^{t+1} - w^t) - x^{t+1}\right\|^2\right]
$$

$$\overset{(2)}{\leq} (1-\alpha)\mathrm{E}_{t+1}\left[\left\|x^{t+1}-w^t\right\|^2\right]$$

$$= (1-\alpha)\mathrm{E}_{t+1}\left[\left\|x^t-\gamma g^t-w^t\right\|^2\right]$$

$$\overset{(14)}{=} (1-\alpha)\gamma^2\mathrm{E}_{t+1}\left[\left\|g^t-\nabla f(w^t)\right\|^2\right] + (1-\alpha)\left\|x^t-\gamma\nabla f(w^t)-w^t\right\|^2$$

$$\overset{(7)}{\leq} \gamma^2\mathrm{E}_{t+1}\left[\left\|g^t-\nabla f(w^t)\right\|^2\right] + \left(1-\frac{\alpha}{2}\right)\left\|w^t-x^t\right\|^2 + \frac{2\gamma^2}{\alpha}\left\|\nabla f(w^t)\right\|^2$$

$$\overset{(8)}{\leq} \gamma^2\mathrm{E}_{t+1}\left[\left\|g^t-\nabla f(w^t)\right\|^2\right] + \left(1-\frac{\alpha}{2}\right)\left\|w^t-x^t\right\|^2$$
$$+ \frac{4\gamma^2}{\alpha}\left\|\nabla f(w^t)-\nabla f(x^t)\right\|^2 + \frac{4\gamma^2}{\alpha}\left\|\nabla f(x^t)-\nabla f(x^*)\right\|^2.$$

Using Assumption 2.1 and Lemma B.1, we obtain

$$\mathrm{E}_{t+1}\left[\left\|w^{t+1}-x^{t+1}\right\|^2\right] \leq \gamma^2\mathrm{E}_{t+1}\left[\left\|g^t-\nabla f(w^t)\right\|^2\right]$$
$$+ \left(1-\frac{\alpha}{2}+\frac{4\gamma^2 L^2}{\alpha}\right)\left\|w^t-x^t\right\|^2 + \frac{8\gamma^2 L}{\alpha}\left(f(x^t)-f(x^*)\right)$$

$$\leq \gamma^2\left(\frac{2\omega}{n^2}\sum_{i=1}^{n}\left\|h_i^t-\nabla f_i(x^*)\right\|^2 + \frac{4\omega\widehat{L}^2}{n}\left\|w^t-x^t\right\|^2 + \frac{8\omega L_{\max}}{n}\left(f(x^t)-f(x^*)\right) + \frac{(\omega+1)\sigma^2}{n}\right)$$
$$+ \left(1-\frac{\alpha}{2}+\frac{4\gamma^2 L^2}{\alpha}\right)\left\|w^t-x^t\right\|^2 + \frac{8\gamma^2 L}{\alpha}\left(f(x^t)-f(x^*)\right)$$

$$= \frac{2\gamma^2\omega}{n^2}\sum_{i=1}^{n}\left\|h_i^t-\nabla f_i(x^*)\right\|^2 + \left(1-\frac{\alpha}{2}+\frac{4\gamma^2 L^2}{\alpha}+\frac{4\gamma^2\omega\widehat{L}^2}{n}\right)\left\|w^t-x^t\right\|^2$$
$$+ \left(\frac{8\gamma^2\omega L_{\max}}{n}+\frac{8\gamma^2 L}{\alpha}\right)\left(f(x^t)-f(x^*)\right) + \frac{\gamma^2(\omega+1)\sigma^2}{n}$$

$$\leq \left(1-\frac{\alpha}{4}\right)\left\|w^t-x^t\right\|^2 + \frac{2\gamma^2\omega}{n^2}\sum_{i=1}^{n}\left\|h_i^t-\nabla f_i(x^*)\right\|^2$$
$$+ \left(\frac{8\gamma^2\omega L_{\max}}{n}+\frac{8\gamma^2 L}{\alpha}\right)\left(f(x^t)-f(x^*)\right) + \frac{\gamma^2(\omega+1)\sigma^2}{n},$$

where we assume that $\gamma \leq \frac{\alpha}{\sqrt{32}L}$ and $\gamma \leq \frac{\sqrt{\alpha n}}{\sqrt{32\omega}\widehat{L}}$. Let us fix some constants $\kappa \geq 0$ and $\nu \geq 0$. In the proof of (34) in Theorem F.1, we do not use the structure of $g^t$. Hence we can reuse (34) here and combine it with the above inequalities to obtain

$$\mathrm{E}_{t+1}\left[f(x^{t+1})\right] + \kappa\frac{1}{n}\sum_{i=1}^{n}\mathrm{E}_{t+1}\left[\left\|h_i^{t+1}-\nabla f_i(x^*)\right\|^2\right] + \nu\mathrm{E}_{t+1}\left[\left\|w^{t+1}-x^{t+1}\right\|^2\right]$$

$$\leq f(x^*) + \gamma\left(\frac{2\omega}{n^2}\sum_{i=1}^{n}\left\|h_i^t-\nabla f_i(x^*)\right\|^2 + \frac{4\omega\widehat{L}^2}{n}\left\|w^t-x^t\right\|^2 + \frac{8\omega L_{\max}}{n}\left(f(x^t)-f(x^*)\right) + \frac{(\omega+1)\sigma^2}{n}\right)$$

$$+ \frac{1}{2\gamma}\left(1-\frac{\gamma\mu}{2}\right)\left\|x^t-x^*\right\|^2 - \frac{1}{2\gamma}\mathrm{E}_{t+1}\left[\left\|x^{t+1}-x^*\right\|^2\right] + 2L\left\|w^t-x^t\right\|^2$$

$$+ \kappa\left((1-\beta)\frac{1}{n}\sum_{i=1}^{n}\left\|h_i^t-\nabla f_i(x^*)\right\|^2 + 2\beta\widehat{L}^2\left\|w^t-x^t\right\|^2 + 4\beta L_{\max}\left(f(x^t)-f(x^*)\right) + \beta^2(\omega+1)\sigma^2\right)$$

$$+ \nu\left(\left(1-\frac{\alpha}{4}\right)\left\|w^t-x^t\right\|^2 + \frac{2\gamma^2\omega}{n^2}\sum_{i=1}^{n}\left\|h_i^t-\nabla f_i(x^*)\right\|^2 + \left(\frac{8\gamma^2\omega L_{\max}}{n}+\frac{8\gamma^2 L}{\alpha}\right)\left(f(x^t)-f(x^*)\right) + \frac{\gamma^2(\omega+1)\sigma^2}{n}\right)$$

Rearranging the last inequality, one can get

$$\frac{1}{2\gamma}\mathrm{E}_{t+1}\left[\left\|x^{t+1}-x^*\right\|^2\right] + \mathrm{E}_{t+1}\left[f(x^{t+1})-f(x^*)\right]$$

$$+ \kappa \frac{1}{n} \sum_{i=1}^{n} \mathrm{E}_{t+1} \left[ \left\| h_i^{t+1} - \nabla f_i(x^*) \right\|^2 \right] + \nu \mathrm{E}_{t+1} \left[ \left\| w^{t+1} - x^{t+1} \right\|^2 \right]$$

$$\leq \frac{1}{2\gamma} \left( 1 - \frac{\gamma\mu}{2} \right) \left\| x^t - x^* \right\|^2$$
$$+ \left( \frac{8\gamma\omega L_{\max}}{n} + \kappa 4\beta L_{\max} + \nu \left( \frac{8\gamma^2\omega L_{\max}}{n} + \frac{8\gamma^2 L}{\alpha} \right) \right) \left( f(x^t) - f(x^*) \right)$$
$$+ \left( \frac{2\gamma\omega}{n} + \nu \frac{2\gamma^2\omega}{n} + \kappa\left(1 - \beta\right) \right) \frac{1}{n} \sum_{i=1}^{n} \left\| h_i^t - \nabla f_i(x^*) \right\|^2$$
$$+ \left( \frac{4\gamma\omega\widehat{L}^2}{n} + 2L + \kappa 2\beta\widehat{L}^2 + \nu\left(1 - \frac{\alpha}{4}\right) \right) \left\| w^t - x^t \right\|^2$$
$$+ \frac{\gamma(\omega + 1)\sigma^2}{n} + \kappa\beta^2(\omega + 1)\sigma^2 + \nu\frac{\gamma^2(\omega + 1)\sigma^2}{n}.$$

Using the same reasoning as in the proof of Theorem F.1, we have

$$\frac{1}{2\gamma}\mathrm{E}_{t+1}\left[ \left\| x^{t+1} - x^* \right\|^2 \right] + \mathrm{E}_{t+1}\left[ f(x^{t+1}) - f(x^*) \right]$$
$$+ \kappa\frac{1}{n}\sum_{i=1}^{n}\mathrm{E}_{t+1}\left[ \left\| h_i^{t+1} - \nabla f_i(x^*) \right\|^2 \right] + \nu\mathrm{E}_{t+1}\left[ \left\| w^{t+1} - x^{t+1} \right\|^2 \right]$$
$$\leq \frac{1}{2\gamma}\left(1 - \frac{\gamma\mu}{2}\right)\left\| x^t - x^* \right\|^2 + \kappa\left(1 - \frac{\gamma\mu}{2}\right)\frac{1}{n}\sum_{i=1}^{n}\left\| h_i^t - \nabla f_i(x^*) \right\|^2 + \nu\left(1 - \frac{\gamma\mu}{2}\right)\left\| w^t - x^t \right\|^2$$
$$+ \frac{1}{2}\left( f(x^t) - f(x^*) \right) + \frac{\gamma(\omega + 1)\sigma^2}{n} + \kappa\beta^2(\omega + 1)\sigma^2 + \nu\frac{\gamma^2(\omega + 1)\sigma^2}{n}$$

for some $\kappa \leq \frac{8\gamma\omega}{n\beta}$ and $\nu \leq \frac{192\gamma\omega\widehat{L}^2}{n\alpha} + \frac{32L}{\alpha}$. Thus

$$\frac{1}{2\gamma}\mathrm{E}_{t+1}\left[ \left\| x^{t+1} - x^* \right\|^2 \right] + \mathrm{E}_{t+1}\left[ f(x^{t+1}) - f(x^*) \right]$$
$$+ \kappa\frac{1}{n}\sum_{i=1}^{n}\mathrm{E}_{t+1}\left[ \left\| h_i^{t+1} - \nabla f_i(x^*) \right\|^2 \right] + \nu\mathrm{E}_{t+1}\left[ \left\| w^{t+1} - x^{t+1} \right\|^2 \right]$$
$$\leq \frac{1}{2\gamma}\left(1 - \frac{\gamma\mu}{2}\right)\left\| x^t - x^* \right\|^2 + \kappa\left(1 - \frac{\gamma\mu}{2}\right)\frac{1}{n}\sum_{i=1}^{n}\left\| h_i^t - \nabla f_i(x^*) \right\|^2 + \nu\left(1 - \frac{\gamma\mu}{2}\right)\left\| w^t - x^t \right\|^2$$
$$+ \frac{1}{2}\left( f(x^t) - f(x^*) \right) + \frac{\gamma(\omega + 1)\sigma^2}{n}$$
$$+ \frac{8\gamma\beta\omega(\omega + 1)\sigma^2}{n} + \frac{192\gamma^3\omega(\omega + 1)\widehat{L}^2\sigma^2}{n^2\alpha} + \frac{32\gamma^2(\omega + 1)L\sigma^2}{n\alpha}$$
$$\leq \frac{1}{2\gamma}\left(1 - \frac{\gamma\mu}{2}\right)\left\| x^t - x^* \right\|^2 + \kappa\left(1 - \frac{\gamma\mu}{2}\right)\frac{1}{n}\sum_{i=1}^{n}\left\| h_i^t - \nabla f_i(x^*) \right\|^2 + \nu\left(1 - \frac{\gamma\mu}{2}\right)\left\| w^t - x^t \right\|^2$$
$$+ \frac{1}{2}\left( f(x^t) - f(x^*) \right) + \frac{12\gamma(\omega + 1)\sigma^2}{n},$$

where used the bounds on $\gamma$ and $\beta$. $\qquad\square$

**Theorem 3.3.** *Let us consider Algorithm 1 using stochastic gradients $\widetilde{\nabla} f_i$ instead of exact gradients $\nabla f_i$ for all $i \in [n]$. Let Assumptions 2.1, 2.2, 2.3 and 3.2 hold, $\beta = \frac{1}{\omega + 1}$, $x^0 = w^0$, and $\gamma \leq \min\left\{ \frac{n}{160\omega L_{\max}}, \frac{\sqrt{n\alpha}}{20\sqrt{\omega}\widehat{L}}, \frac{\alpha}{100L}, \frac{1}{(\omega + 1)\mu} \right\}$. Then Algorithm 1 returns $x^T$ such that*

$$\frac{1}{2\gamma}\mathrm{E}\left[ \left\| x^T - x^* \right\|^2 \right] + \mathrm{E}\left[ f(x^T) - f(x^*) \right] \leq \left(1 - \frac{\gamma\mu}{2}\right)^T V^0 + \frac{24(\omega + 1)\sigma^2}{\mu n},$$

*where $V^0 := \frac{1}{2\gamma}\mathrm{E}\left[ \left\| x^0 - x^* \right\|^2 \right] + \left( f(x^0) - f(x^*) \right) + \frac{8\gamma\omega(\omega + 1)}{n^2}\sum_{i=1}^{n}\left\| h_i^0 - \nabla f_i(x^*) \right\|^2.$*

*Proof.* Using $\gamma \leq \frac{\alpha}{100L} \leq \frac{1}{\mu}$, we can bound (40) as follows:

$$\frac{1}{2\gamma} \mathrm{E}\left[\left\|x^{t+1} - x^*\right\|^2\right] + \mathrm{E}\left[f(x^{t+1}) - f(x^*)\right]$$

$$+ \kappa \frac{1}{n} \sum_{i=1}^{n} \mathrm{E}\left[\left\|h_i^{t+1} - \nabla f_i(x^*)\right\|^2\right] + \nu \mathrm{E}\left[\left\|w^{t+1} - x^{t+1}\right\|^2\right]$$

$$\leq \frac{1}{2\gamma}\left(1 - \frac{\gamma\mu}{2}\right) \mathrm{E}\left[\left\|x^t - x^*\right\|^2\right] + \frac{1}{2}\mathrm{E}\left[f(x^t) - f(x^*)\right]$$

$$+ \kappa\left(1 - \frac{\gamma\mu}{2}\right) \frac{1}{n} \sum_{i=1}^{n} \mathrm{E}\left[\left\|h_i^t - \nabla f_i(x^*)\right\|^2\right] + \nu\left(1 - \frac{\gamma\mu}{2}\right) \mathrm{E}\left[\left\|w^t - x^t\right\|^2\right] + \frac{12\gamma(\omega + 1)\sigma^2}{n}$$

$$\leq \frac{1}{2\gamma}\left(1 - \frac{\gamma\mu}{2}\right) \mathrm{E}\left[\left\|x^t - x^*\right\|^2\right] + \left(1 - \frac{\gamma\mu}{2}\right) \mathrm{E}\left[f(x^t) - f(x^*)\right]$$

$$+ \kappa\left(1 - \frac{\gamma\mu}{2}\right) \frac{1}{n} \sum_{i=1}^{n} \mathrm{E}\left[\left\|h_i^t - \nabla f_i(x^*)\right\|^2\right] + \nu\left(1 - \frac{\gamma\mu}{2}\right) \mathrm{E}\left[\left\|w^t - x^t\right\|^2\right] + \frac{12\gamma(\omega + 1)\sigma^2}{n}$$

$$= \left(1 - \frac{\gamma\mu}{2}\right)\left(\frac{1}{2\gamma} \mathrm{E}\left[\left\|x^t - x^*\right\|^2\right] + \mathrm{E}\left[f(x^t) - f(x^*)\right] + \kappa \frac{1}{n} \sum_{i=1}^{n} \mathrm{E}\left[\left\|h_i^t - \nabla f_i(x^*)\right\|^2\right] + \nu \mathrm{E}\left[\left\|w^t - x^t\right\|^2\right]\right)$$

$$+ \frac{12\gamma(\omega + 1)\sigma^2}{n}.$$

Recursively applying the last inequality and using the assumption $x^0 = w^0$, one can get that

$$\frac{1}{2\gamma} \mathrm{E}\left[\left\|x^T - x^*\right\|^2\right] + \mathrm{E}\left[f(x^T) - f(x^*)\right] + \kappa \frac{1}{n} \sum_{i=1}^{n} \mathrm{E}\left[\left\|h_i^T - \nabla f_i(x^*)\right\|^2\right] + \nu \mathrm{E}\left[\left\|w^T - x^T\right\|^2\right]$$

$$\leq \left(1 - \frac{\gamma\mu}{2}\right)^T \left(\frac{1}{2\gamma} \mathrm{E}\left[\left\|x^0 - x^*\right\|^2\right] + \left(f(x^0) - f(x^*)\right) + \kappa \frac{1}{n} \sum_{i=1}^{n} \left\|h_i^0 - \nabla f_i(x^*)\right\|^2\right)$$

$$+ \sum_{i=0}^{T-1} \left(1 - \frac{\gamma\mu}{2}\right)^i \frac{12\gamma(\omega + 1)\sigma^2}{n}$$

$$\leq \left(1 - \frac{\gamma\mu}{2}\right)^T \left(\frac{1}{2\gamma} \mathrm{E}\left[\left\|x^0 - x^*\right\|^2\right] + \left(f(x^0) - f(x^*)\right) + \kappa \frac{1}{n} \sum_{i=1}^{n} \left\|h_i^0 - \nabla f_i(x^*)\right\|^2\right)$$

$$+ \frac{24(\omega + 1)\sigma^2}{n\mu}$$

Using the nonnegativity of the terms and the bound on $\kappa$, we obtain

$$\frac{1}{2\gamma} \mathrm{E}\left[\left\|x^T - x^*\right\|^2\right] + \mathrm{E}\left[f(x^T) - f(x^*)\right]$$

$$\leq \left(1 - \frac{\gamma\mu}{2}\right)^T \left(\frac{1}{2\gamma} \mathrm{E}\left[\left\|x^0 - x^*\right\|^2\right] + \left(f(x^0) - f(x^*)\right) + \frac{8\gamma\omega}{n^2\beta} \sum_{i=1}^{n} \left\|h_i^0 - \nabla f_i(x^*)\right\|^2\right)$$

$$+ \frac{24(\omega + 1)\sigma^2}{n\mu}.$$

$\square$

**Theorem F.4.** *Let us consider Algorithm 1 using stochastic gradients $\widetilde{\nabla} f_i$ instead of the exact gradients $\nabla f_i$ for all $i \in [n]$. Let us assume that Assumptions 2.1, 2.2, 2.3 and 3.2 hold, the strong convexity parameter satisfies $\mu = 0$, $\beta = \frac{1}{\omega+1}$, $x^0 = w^0$, and*

$$\gamma \leq \min\left\{\frac{n}{160\omega L_{\max}}, \frac{\sqrt{n\alpha}}{20\sqrt{\omega}\widehat{L}}, \frac{\alpha}{100L}\right\}.$$

*Then Algorithm 1 guarantees the following convergence rate:*

$$f\left(\frac{1}{T}\sum_{t=1}^{T}x^t\right) - f(x^*) \leq \frac{1}{\gamma T}\left\|x^0 - x^*\right\|^2 + \frac{f(x^0) - \nabla f(x^*)}{T}$$

$$+ \frac{16\gamma\omega(\omega+1)}{Tn^2}\sum_{i=1}^{n}\left\|h_i^0 - \nabla f_i(x^*)\right\|^2 + \frac{24\gamma(\omega+1)\sigma^2}{n}.$$

*Proof.* Let us bound (40):

$$\frac{1}{2\gamma}\mathrm{E}\left[\left\|x^{t+1} - x^*\right\|^2\right] + \mathrm{E}\left[f(x^{t+1}) - f(x^*)\right]$$

$$+ \kappa\frac{1}{n}\sum_{i=1}^{n}\mathrm{E}\left[\left\|h_i^{t+1} - \nabla f_i(x^*)\right\|^2\right] + \nu\mathrm{E}\left[\left\|w^{t+1} - x^{t+1}\right\|^2\right]$$

$$\leq \frac{1}{2\gamma}\left(1 - \frac{\gamma\mu}{2}\right)\mathrm{E}\left[\left\|x^t - x^*\right\|^2\right] + \frac{1}{2}\mathrm{E}\left[f(x^t) - f(x^*)\right]$$

$$+ \kappa\left(1 - \frac{\gamma\mu}{2}\right)\frac{1}{n}\sum_{i=1}^{n}\mathrm{E}\left[\left\|h_i^t - \nabla f_i(x^*)\right\|^2\right] + \nu\left(1 - \frac{\gamma\mu}{2}\right)\mathrm{E}\left[\left\|w^t - x^t\right\|^2\right] + \frac{12\gamma(\omega+1)\sigma^2}{n}$$

$$\leq \frac{1}{2\gamma}\mathrm{E}\left[\left\|x^t - x^*\right\|^2\right] + \frac{1}{2}\mathrm{E}\left[f(x^t) - f(x^*)\right] + \kappa\frac{1}{n}\sum_{i=1}^{n}\mathrm{E}\left[\left\|h_i^t - \nabla f_i(x^*)\right\|^2\right]$$

$$+ \nu\mathrm{E}\left[\left\|w^t - x^t\right\|^2\right] + \frac{12\gamma(\omega+1)\sigma^2}{n}.$$

Summing the inequality for $t \in \{0,\ldots,T-1\}$ gives

$$\frac{1}{2\gamma}\mathrm{E}\left[\left\|x^T - x^*\right\|^2\right] + \frac{1}{2}\mathrm{E}\left[f(x^T) - f(x^*)\right] + \frac{1}{2}\sum_{t=1}^{T}\mathrm{E}\left[f(x^t) - f(x^*)\right]$$

$$+ \kappa\frac{1}{n}\sum_{i=1}^{n}\mathrm{E}\left[\left\|h_i^T - \nabla f_i(x^*)\right\|^2\right] + \nu\mathrm{E}\left[\left\|w^T - x^T\right\|^2\right]$$

$$\leq \frac{1}{2\gamma}\left\|x^0 - x^*\right\|^2 + \frac{1}{2}\left(f(x^0) - f(x^*)\right) + \kappa\frac{1}{n}\sum_{i=1}^{n}\left\|h_i^0 - \nabla f_i(x^*)\right\|^2$$

$$+ \nu\left\|w^0 - x^0\right\|^2 + \frac{12T\gamma(\omega+1)\sigma^2}{n}$$

$$\leq \frac{1}{2\gamma}\left\|x^0 - x^*\right\|^2 + \frac{1}{2}\left(f(x^0) - f(x^*)\right) + \frac{8\gamma\omega}{n^2\beta}\sum_{i=1}^{n}\left\|h_i^0 - \nabla f_i(x^*)\right\|^2 + \frac{12T\gamma(\omega+1)\sigma^2}{n},$$

where we used the fact that $x^0 = w^0$ and the bound on $\kappa$. Using nonnegativity of the terms and convexity, we have

$$f\left(\frac{1}{T}\sum_{t=1}^{T}x^t\right) - f(x^*) \leq \frac{1}{\gamma T}\left\|x^0 - x^*\right\|^2 + \frac{f(x^0) - \nabla f(x^*)}{T}$$

$$+ \frac{16\gamma\omega}{Tn^2\beta}\sum_{i=1}^{n}\left\|h_i^0 - \nabla f_i(x^*)\right\|^2 + \frac{24\gamma(\omega+1)\sigma^2}{n}.$$

$\square$

## G   PROOFS FOR EF21-P + DCGD IN THE CONVEX CASE

As mentioned before, EF21-P + DCGD arises a special case of EF21-P + DIANA if we do not attempt to learn any local gradient shifts $h_i^t$ and instead set them to 0 throughout. This can be achieved by setting $\beta = 0$.

---

**Algorithm 2** EF21-P + DCGD

1: **Parameters:** learning rate $\gamma > 0$; initial iterate $x^0 \in \mathbb{R}^d$ (stored on the server and the workers); initial iterate shift $w^0 = x^0 \in \mathbb{R}^d$ (stored on the server and the workers)
2: **for** $t = 0, 1, \ldots, T-1$ **do**
3:     **for** $i = 1, \ldots, n$ **in parallel do**
4:         $g_i^t = \mathcal{C}_i^D(\nabla f_i(w^t))$                      Compress gradient via $\mathcal{C}_i^D \in \mathbb{U}(\omega)$
5:         Send message $g_i^t$ to the server
6:     **end for**
7:     $g^t = \frac{1}{n} \sum_{i=1}^n g_i^t$                              Compute gradient estimator
8:     $x^{t+1} = x^t - \gamma g^t$                              Take gradient-type step
9:     $p^{t+1} = \mathcal{C}^P\left(x^{t+1} - w^t\right)$        Compress shifted model on the server via $\mathcal{C}^P \in \mathbb{B}\left(\alpha\right)$
10:    $w^{t+1} = w^t + p^{t+1}$                           Update model shift
11:    Broadcast $p^{t+1}$ to all workers
12:    **for** $i = 1, \ldots, n$ **in parallel do**
13:        $w^{t+1} = w^t + p^{t+1}$                       Update model shift
14:    **end for**
15: **end for**

---

The proofs in this section almost repeat the proofs from Section F.

**Theorem G.1.** *Let us assume that Assumptions 2.1, 2.2 and 2.3 hold and choose*

$$\gamma \le \min\left\{\frac{n}{160\omega L_{\max}}, \frac{\sqrt{n\alpha}}{20\sqrt{\omega}\widehat{L}}, \frac{\alpha}{100L}\right\}.$$

*Then Algorithm 2 guarantees that*

$$\frac{1}{2\gamma}\mathrm{E}\left[\left\|x^{t+1} - x^*\right\|^2\right] + \mathrm{E}\left[f(x^{t+1}) - f(x^*)\right] + \nu\mathrm{E}\left[\left\|w^{t+1} - x^{t+1}\right\|^2\right]$$

$$\le \frac{1}{2\gamma}\left(1 - \frac{\gamma\mu}{2}\right)\mathrm{E}\left[\left\|x^t - x^*\right\|^2\right] + \frac{1}{2}\mathrm{E}\left[f(x^t) - f(x^*)\right]$$

$$+ \nu\left(1 - \frac{\gamma\mu}{2}\right)\mathrm{E}\left[\left\|w^t - x^t\right\|^2\right] + \frac{4\gamma\omega}{n}\left(\frac{1}{n}\sum_{i=1}^n \left\|\nabla f_i(x^*)\right\|^2\right), \tag{41}$$

*where $\nu \le \frac{32\gamma\omega\widehat{L}^2}{n\alpha} + \frac{16L}{\alpha}$.*

*Proof.* Note that EF21-P + DCGD is EF21-P + DIANA with $\beta = 0$ and $h_i^t = 0$ for all $i \in [n]$ and $t \ge 0$. Up to (37), we can reuse the proof of Theorem F.1 and obtain

$$\frac{1}{2\gamma}\mathrm{E}_{t+1}\left[\left\|x^{t+1} - x^*\right\|^2\right] + \mathrm{E}_{t+1}\left[f(x^{t+1}) - f(x^*)\right]$$

$$+ \kappa\frac{1}{n}\sum_{i=1}^n \mathrm{E}_{t+1}\left[\left\|h_i^{t+1} - \nabla f_i(x^*)\right\|^2\right] + \nu\mathrm{E}_{t+1}\left[\left\|w^{t+1} - x^{t+1}\right\|^2\right]$$

$$\le \frac{1}{2\gamma}\left(1 - \frac{\gamma\mu}{2}\right)\left\|x^t - x^*\right\|^2$$

$$+ \left(\frac{8\gamma\omega L_{\max}}{n} + \kappa 4\beta L_{\max} + \nu\left(\frac{8\gamma^2\omega L_{\max}}{n} + \frac{8\gamma^2 L}{\alpha}\right)\right)\left(f(x^t) - f(x^*)\right)$$

$$+ \left(\frac{2\gamma\omega}{n} + \nu\frac{2\gamma^2\omega}{n} + \kappa\left(1 - \beta\right)\right)\frac{1}{n}\sum_{i=1}^n \left\|h_i^t - \nabla f_i(x^*)\right\|^2$$

$$+ \left( \frac{4\gamma\omega\widehat{L}^2}{n} + 2L + \kappa 2\beta\widehat{L}^2 + \nu\left(1 - \frac{\alpha}{4}\right) \right) \left\| w^t - x^t \right\|^2.$$

Due to $\beta = 0$, we have

$$\frac{1}{2\gamma}\mathrm{E}_{t+1}\left[\left\| x^{t+1} - x^* \right\|^2\right] + \mathrm{E}_{t+1}\left[f(x^{t+1}) - f(x^*)\right]$$

$$+ \kappa\frac{1}{n}\sum_{i=1}^{n}\mathrm{E}_{t+1}\left[\left\| h_i^{t+1} - \nabla f_i(x^*) \right\|^2\right] + \nu\mathrm{E}_{t+1}\left[\left\| w^{t+1} - x^{t+1} \right\|^2\right]$$

$$\leq \frac{1}{2\gamma}\left(1 - \frac{\gamma\mu}{2}\right)\left\| x^t - x^* \right\|^2$$

$$+ \left(\frac{8\gamma\omega L_{\max}}{n} + \nu\left(\frac{8\gamma^2\omega L_{\max}}{n} + \frac{8\gamma^2 L}{\alpha}\right)\right)\left(f(x^t) - f(x^*)\right)$$

$$+ \left(\frac{2\gamma\omega}{n} + \nu\frac{2\gamma^2\omega}{n} + \kappa\right)\frac{1}{n}\sum_{i=1}^{n}\left\| h_i^t - \nabla f_i(x^*) \right\|^2$$

$$+ \left(\frac{4\gamma\omega\widehat{L}^2}{n} + 2L + \nu\left(1 - \frac{\alpha}{4}\right)\right)\left\| w^t - x^t \right\|^2.$$

Taking $\kappa = 0$ and $\nu = \frac{32\gamma\omega\widehat{L}^2}{\alpha n} + \frac{16L}{\alpha}$, we obtain

$$\frac{1}{2\gamma}\mathrm{E}_{t+1}\left[\left\| x^{t+1} - x^* \right\|^2\right] + \mathrm{E}_{t+1}\left[f(x^{t+1}) - f(x^*)\right] + \nu\mathrm{E}_{t+1}\left[\left\| w^{t+1} - x^{t+1} \right\|^2\right]$$

$$\leq \frac{1}{2\gamma}\left(1 - \frac{\gamma\mu}{2}\right)\left\| x^t - x^* \right\|^2 + \nu\left(1 - \frac{\alpha}{8}\right)\left\| w^t - x^t \right\|^2$$

$$+ \left(\frac{8\gamma\omega L_{\max}}{n} + \left(\frac{32\gamma\omega\widehat{L}^2}{\alpha n} + \frac{16L}{\alpha}\right)\left(\frac{8\gamma^2\omega L_{\max}}{n} + \frac{8\gamma^2 L}{\alpha}\right)\right)\left(f(x^t) - f(x^*)\right)$$

$$+ \left(\frac{2\gamma\omega}{n} + \nu\frac{2\gamma^2\omega}{n}\right)\frac{1}{n}\sum_{i=1}^{n}\left\| h_i^t - \nabla f_i(x^*) \right\|^2$$

$$= \frac{1}{2\gamma}\left(1 - \frac{\gamma\mu}{2}\right)\left\| x^t - x^* \right\|^2 + \nu\left(1 - \frac{\alpha}{8}\right)\left\| w^t - x^t \right\|^2$$

$$+ \left(\frac{8\gamma\omega L_{\max}}{n} + \frac{256\gamma^3\omega^2\widehat{L}^2 L_{\max}}{n^2\alpha} + \frac{256\gamma^3\omega L\widehat{L}^2}{n\alpha^2} + \frac{128\gamma^2\omega L L_{\max}}{n\alpha} + \frac{128\gamma^2 L^2}{\alpha^2}\right)\left(f(x^t) - f(x^*)\right)$$

$$+ \left(\frac{2\gamma\omega}{n} + \nu\frac{2\gamma^2\omega}{n}\right)\frac{1}{n}\sum_{i=1}^{n}\left\| h_i^t - \nabla f_i(x^*) \right\|^2.$$

Using the assumptions on $\gamma$, we have

$$\frac{8\gamma\omega L_{\max}}{n} \leq \frac{1}{10},$$

$$\frac{256\gamma^3\omega^2\widehat{L}^2 L_{\max}}{n^2\alpha} \leq \frac{20\gamma^2\omega\widehat{L}^2}{n\alpha} \leq \frac{1}{10},$$

$$\frac{128\gamma^2\omega L L_{\max}}{n\alpha} \leq \frac{4\gamma L}{\alpha} \leq \frac{1}{10},$$

$$\frac{256\gamma^3\omega L\widehat{L}^2}{n\alpha^2} \leq \frac{40\gamma^2\omega\widehat{L}^2}{n\alpha} \leq \frac{1}{10},$$

$$\frac{128\gamma^2 L^2}{\alpha^2} \leq \frac{1}{10}.$$

Considering $\gamma \leq \frac{\alpha}{4\mu}$, we obtain

$$\frac{1}{2\gamma}\mathrm{E}_{t+1}\left[\left\| x^{t+1} - x^* \right\|^2\right] + \mathrm{E}_{t+1}\left[f(x^{t+1}) - f(x^*)\right] + \nu\mathrm{E}_{t+1}\left[\left\| w^{t+1} - x^{t+1} \right\|^2\right]$$

$$\leq \frac{1}{2\gamma}\left(1 - \frac{\gamma\mu}{2}\right)\left\|x^t - x^*\right\|^2 + \frac{1}{2}\left(f(x^t) - f(x^*)\right) + \nu\left(1 - \frac{\gamma\mu}{2}\right)\left\|w^t - x^t\right\|^2$$

$$+ \left(\frac{2\gamma\omega}{n} + \nu\frac{2\gamma^2\omega}{n}\right)\frac{1}{n}\sum_{i=1}^{n}\left\|h_i^t - \nabla f_i(x^*)\right\|^2.$$

From the assumptions on $\gamma$, we have

$$\frac{2\gamma\omega}{n} + \nu\frac{2\gamma^2\omega}{n} \leq \frac{2\gamma\omega}{n} + \left(\frac{32\gamma\omega\widehat{L}^2}{\alpha n} + \frac{16L}{\alpha}\right)\frac{2\gamma^2\omega}{n} \leq \frac{4\gamma\omega}{n}$$

and hence

$$\frac{1}{2\gamma}\mathrm{E}_{t+1}\left[\left\|x^{t+1} - x^*\right\|^2\right] + \mathrm{E}_{t+1}\left[f(x^{t+1}) - f(x^*)\right] + \nu\mathrm{E}_{t+1}\left[\left\|w^{t+1} - x^{t+1}\right\|^2\right]$$

$$\leq \frac{1}{2\gamma}\left(1 - \frac{\gamma\mu}{2}\right)\left\|x^t - x^*\right\|^2 + \frac{1}{2}\left(f(x^t) - f(x^*)\right) + \nu\left(1 - \frac{\gamma\mu}{2}\right)\left\|w^t - x^t\right\|^2$$

$$+ \frac{4\gamma\omega}{n}\frac{1}{n}\sum_{i=1}^{n}\left\|h_i^t - \nabla f_i(x^*)\right\|^2.$$

Taking the full expectation, we obtain

$$\frac{1}{2\gamma}\mathrm{E}\left[\left\|x^{t+1} - x^*\right\|^2\right] + \mathrm{E}\left[f(x^{t+1}) - f(x^*)\right] + \nu\mathrm{E}\left[\left\|w^{t+1} - x^{t+1}\right\|^2\right]$$

$$\leq \frac{1}{2\gamma}\left(1 - \frac{\gamma\mu}{2}\right)\mathrm{E}\left[\left\|x^t - x^*\right\|^2\right] + \frac{1}{2}\mathrm{E}\left[f(x^t) - f(x^*)\right] + \nu\left(1 - \frac{\gamma\mu}{2}\right)\mathrm{E}\left[\left\|w^t - x^t\right\|^2\right]$$

$$+ \frac{4\gamma\omega}{n}\frac{1}{n}\sum_{i=1}^{n}\mathrm{E}\left[\left\|h_i^t - \nabla f_i(x^*)\right\|^2\right].$$

It remains to use (36) with $\beta = 0$ to finish the proof of the theorem. $\qquad\square$

**Theorem G.2.** *Let us assume that Assumptions 2.1, 2.2 and 2.3 hold, the strong convexity parameter satisfies $\mu = 0$, $x^0 = w^0$ and*

$$\gamma \leq \min\left\{\frac{n}{160\omega L_{\max}}, \frac{\sqrt{n\alpha}}{20\sqrt{\omega}\widehat{L}}, \frac{\alpha}{100L}\right\}.$$

*Then Algorithm 2 guarantees that*

$$f\left(\frac{1}{T}\sum_{t=1}^{T}x^t\right) - f(x^*) \leq \frac{1}{\gamma T}\left\|x^0 - x^*\right\|^2 + \frac{f(x^0) - \nabla f(x^*)}{T} + \frac{8\gamma\omega}{n}\left(\frac{1}{n}\sum_{i=1}^{n}\|\nabla f_i(x^*)\|^2\right).$$

*Proof.* Let us bound (41):

$$\frac{1}{2\gamma}\mathrm{E}\left[\left\|x^{t+1} - x^*\right\|^2\right] + \mathrm{E}\left[f(x^{t+1}) - f(x^*)\right] + \nu\mathrm{E}\left[\left\|w^{t+1} - x^{t+1}\right\|^2\right]$$

$$\leq \frac{1}{2\gamma}\left(1 - \frac{\gamma\mu}{2}\right)\mathrm{E}\left[\left\|x^t - x^*\right\|^2\right] + \frac{1}{2}\mathrm{E}\left[f(x^t) - f(x^*)\right] + \nu\left(1 - \frac{\gamma\mu}{2}\right)\mathrm{E}\left[\left\|w^t - x^t\right\|^2\right]$$

$$+ \frac{4\gamma\omega}{n}\left(\frac{1}{n}\sum_{i=1}^{n}\|\nabla f_i(x^*)\|^2\right)$$

$$\leq \frac{1}{2\gamma}\mathrm{E}\left[\left\|x^t - x^*\right\|^2\right] + \frac{1}{2}\mathrm{E}\left[f(x^t) - f(x^*)\right] + \nu\mathrm{E}\left[\left\|w^t - x^t\right\|^2\right] + \frac{4\gamma\omega}{n}\left(\frac{1}{n}\sum_{i=1}^{n}\|\nabla f_i(x^*)\|^2\right).$$

We now sum the inequality for $t \in \{0, \ldots, T-1\}$ and obtain

$$\frac{1}{2\gamma}\mathrm{E}\left[\left\|x^T - x^*\right\|^2\right] + \frac{1}{2}\mathrm{E}\left[f(x^T) - f(x^*)\right] + \frac{1}{2}\sum_{t=1}^{T}\mathrm{E}\left[f(x^t) - f(x^*)\right] + \nu\mathrm{E}\left[\left\|w^T - x^T\right\|^2\right]$$

$$\leq \frac{1}{2\gamma} \left\| x^0 - x^* \right\|^2 + \frac{1}{2} \left( f(x^0) - f(x^*) \right) + \nu \left\| w^0 - x^0 \right\|^2 + T \frac{4\gamma\omega}{n} \left( \frac{1}{n} \sum_{i=1}^{n} \|\nabla f_i(x^*)\|^2 \right)$$

$$= \frac{1}{2\gamma} \left\| x^0 - x^* \right\|^2 + \frac{1}{2} \left( f(x^0) - f(x^*) \right) + T \frac{4\gamma\omega}{n} \left( \frac{1}{n} \sum_{i=1}^{n} \|\nabla f_i(x^*)\|^2 \right).$$

where we used the assumption $x^0 = w^0$. Non-negativity of the terms and convexity gives

$$f \left( \frac{1}{T} \sum_{t=1}^{T} x^t \right) - f(x^*) \leq \frac{1}{\gamma T} \left\| x^0 - x^* \right\|^2 + \frac{f(x^0) - \nabla f(x^*)}{T} + \frac{8\gamma\omega}{n} \left( \frac{1}{n} \sum_{i=1}^{n} \|\nabla f_i(x^*)\|^2 \right).$$

$\square$

**Theorem G.3.** *Let us assume that Assumptions 2.1, 2.2 and 2.3 hold, $x^0 = w^0$, and*

$$\gamma \leq \min \left\{ \frac{n}{160\omega L_{\max}}, \frac{\sqrt{n\alpha}}{20\sqrt{\omega \widehat{L}}}, \frac{\alpha}{100L} \right\}.$$

*Then Algorithm 2 guarantees that*

$$\frac{1}{2\gamma} \mathrm{E} \left[ \left\| x^T - x^* \right\|^2 \right] + \mathrm{E} \left[ f(x^T) - f(x^*) \right]$$

$$\leq \left( 1 - \frac{\gamma\mu}{2} \right)^T \left( \frac{1}{2\gamma} \mathrm{E} \left[ \left\| x^0 - x^* \right\|^2 \right] + \left( f(x^0) - f(x^*) \right) \right) + \frac{8\omega}{n\mu} \left( \frac{1}{n} \sum_{i=1}^{n} \|\nabla f_i(x^*)\|^2 \right).$$

*Proof.* Using $\gamma \leq \frac{\alpha}{100L} \leq \frac{1}{\mu}$, let us bound (41):

$$\frac{1}{2\gamma} \mathrm{E} \left[ \left\| x^{t+1} - x^* \right\|^2 \right] + \mathrm{E} \left[ f(x^{t+1}) - f(x^*) \right] + \nu \mathrm{E} \left[ \left\| w^{t+1} - x^{t+1} \right\|^2 \right]$$

$$\leq \frac{1}{2\gamma} \left( 1 - \frac{\gamma\mu}{2} \right) \mathrm{E} \left[ \left\| x^t - x^* \right\|^2 \right] + \frac{1}{2} \mathrm{E} \left[ f(x^t) - f(x^*) \right] + \nu \left( 1 - \frac{\gamma\mu}{2} \right) \mathrm{E} \left[ \left\| w^t - x^t \right\|^2 \right]$$

$$+ \frac{4\gamma\omega}{n} \left( \frac{1}{n} \sum_{i=1}^{n} \|\nabla f_i(x^*)\|^2 \right)$$

$$\leq \frac{1}{2\gamma} \left( 1 - \frac{\gamma\mu}{2} \right) \mathrm{E} \left[ \left\| x^t - x^* \right\|^2 \right] + \left( 1 - \frac{\gamma\mu}{2} \right) \mathrm{E} \left[ f(x^t) - f(x^*) \right] + \nu \left( 1 - \frac{\gamma\mu}{2} \right) \mathrm{E} \left[ \left\| w^t - x^t \right\|^2 \right]$$

$$+ \frac{4\gamma\omega}{n} \left( \frac{1}{n} \sum_{i=1}^{n} \|\nabla f_i(x^*)\|^2 \right)$$

$$= \left( 1 - \frac{\gamma\mu}{2} \right) \left( \frac{1}{2\gamma} \mathrm{E} \left[ \left\| x^t - x^* \right\|^2 \right] + \mathrm{E} \left[ f(x^t) - f(x^*) \right] + \nu \mathrm{E} \left[ \left\| w^t - x^t \right\|^2 \right] \right)$$

$$+ \frac{4\gamma\omega}{n} \left( \frac{1}{n} \sum_{i=1}^{n} \|\nabla f_i(x^*)\|^2 \right).$$

Recursively applying the last inequality and using $x^0 = w^0$, one obtains

$$\frac{1}{2\gamma} \mathrm{E} \left[ \left\| x^T - x^* \right\|^2 \right] + \mathrm{E} \left[ f(x^T) - f(x^*) \right] + \nu \mathrm{E} \left[ \left\| w^T - x^T \right\|^2 \right]$$

$$\leq \left( 1 - \frac{\gamma\mu}{2} \right)^T \left( \frac{1}{2\gamma} \mathrm{E} \left[ \left\| x^0 - x^* \right\|^2 \right] + \left( f(x^0) - f(x^*) \right) \right)$$

$$+ \sum_{i=0}^{T-1} \left( 1 - \frac{\gamma\mu}{2} \right)^i \frac{4\gamma\omega}{n} \left( \frac{1}{n} \sum_{i=1}^{n} \|\nabla f_i(x^*)\|^2 \right)$$

$$\leq \left( 1 - \frac{\gamma\mu}{2} \right)^T \left( \frac{1}{2\gamma} \mathrm{E} \left[ \left\| x^0 - x^* \right\|^2 \right] + \left( f(x^0) - f(x^*) \right) \right)$$

$$+ \sum_{i=0}^{\infty} \left(1 - \frac{\gamma\mu}{2}\right)^i \frac{4\gamma\omega}{n} \left(\frac{1}{n} \sum_{i=1}^{n} \|\nabla f_i(x^*)\|^2\right)$$

$$= \left(1 - \frac{\gamma\mu}{2}\right)^T \left(\frac{1}{2\gamma}\mathrm{E}\left[\left\|x^0 - x^*\right\|^2\right] + \left(f(x^0) - f(x^*)\right)\right) + \frac{8\omega}{n\mu} \left(\frac{1}{n} \sum_{i=1}^{n} \|\nabla f_i(x^*)\|^2\right)$$

Non-negativity of $\mathrm{E}\left[\left\|w^T - x^T\right\|^2\right]$ gives

$$\frac{1}{2\gamma}\mathrm{E}\left[\left\|x^T - x^*\right\|^2\right] + \mathrm{E}\left[f(x^T) - f(x^*)\right]$$

$$\leq \left(1 - \frac{\gamma\mu}{2}\right)^T \left(\frac{1}{2\gamma}\mathrm{E}\left[\left\|x^0 - x^*\right\|^2\right] + \left(f(x^0) - f(x^*)\right)\right) + \frac{8\omega}{n\mu} \left(\frac{1}{n} \sum_{i=1}^{n} \|\nabla f_i(x^*)\|^2\right).$$

$\square$

## H FUTURE WORK AND POSSIBLE EXTENSIONS

In this paper, many important features of distributed and federated learning were not investigated in detail. These include variance reduction of stochastic gradients (Horváth et al., 2022; Tyurin & Richtárik, 2022b), acceleration (Li & Richtárik, 2021; Li et al., 2020), local steps (Murata & Suzuki, 2021), partial participation (McMahan et al., 2017; Tyurin & Richtárik, 2022a) and asynchronous SGD (Koloskova et al., 2022). While some are simple exercises and can be easily added to our methods, many of them deserve further investigation and separate work. Further, note that several authors, including Szlendak et al. (2021); Richtárik et al. (2022); Condat et al. (2022), considered somewhat different families of compressors than those we consider here. We believe that the results and discussion from our paper can be adapted to these families.

