# OpenReview forum: "EF21-P and Friends: Improved Theoretical Communication Complexity for Distributed Optimization with Bidirectional Compression"
_ICLR.cc/2023/Conference — Submitted to ICLR 2023_

### Official Review · Reviewer_m3XR · 2022-10-24

**Confidence:** 3
**Correctness:** 2
**Technical Novelty And Significance:** 2
**Empirical Novelty And Significance:** 1
**Recommendation:** 1

**Clarity, Quality, Novelty And Reproducibility:**

I believe that the main algorithm proposed in this paper, EF21-P, is actually not novel and is equivalent to the original error feedback (or EF, or error compensation).
Here is some simple derivation:

First, recall that the formula in Equation (3) for EF21-P is (I ignore the step "t" for the compressor):

$x^{t+1} = x^t - \gamma \nabla f(w^t)$

$w^{t+1} = w^t + C(x^{t+1} - w^t)$

$w^0 = x^0$

Now, we add an auxiliary variable $a^t$, which is actually the message before compression, defined as follows:

$a^t $
$= x^{t+1} - w^t $
$= x^{t+1} - x^t + x^t - w^t $
$ = - \gamma \nabla f(w^t) + x^t - w^t $
$= - \gamma \nabla f(w^t) + x^t - w^{t-1} + w^{t-1} - w^t $
$= - \gamma \nabla f(w^t) + a^{t-1} + w^{t-1} - w^t $
$= - \gamma \nabla f(w^t) + a^{t-1} - C(x^t - w^{t-1}) $
$= - \gamma \nabla f(w^t) + a^{t-1} - C(a^{t-1}) $

$a^0 = x^{1} - w^0 = x^{1} - x^0 =  - \gamma \nabla f(w^0) = - \gamma \nabla f(x^0)$

By using $a^t$, we can totally get rid of $x^t$ and re-write the algorithm as follows:

$w^{t+1} = w^t + C(a^t)$

$a^t = - \gamma \nabla f(w^t) + a^{t-1} - C(a^{t-1}) $

Now, if you are familiar with error feedback, you would find that the algorithm above is exactly the original error feedback, and the term $a^{t-1} - C(a^{t-1})$ is exactly the "residual error" in error feedback (please refer to the EF-SignSGD paper: http://proceedings.mlr.press/v97/karimireddy19a/karimireddy19a.pdf).

Given that my derivations above are correct, i.e., EF21-P is equivalent to EF, then we would have even more issues in this paper:
1. EF21-P + DCGD is just EF-SGD using quantization as the compressor, which is not novel
2. EF21-P + DIANA is just EF + DIANA. There is already an algorithm called EC-SGD-DIANA (error compensation + DIANA, and error compensation is the same as error feedback) proposed in the following paper:
Gorbunov, Eduard, Dmitry Kovalev, Dmitry Makarenko, and Peter Richtárik. "Linearly converging error compensated SGD." NeurIPS 2020. (https://proceedings.neurips.cc//paper/2020/file/ef9280fbc5317f17d480e4d4f61b3751-Paper.pdf)
I'm not exactly sure whether EF + DIANA is equivalent to EC-SGD-DIANA, or maybe EF + DIANA is just another variant of DIANA that combines error feedback with DIANA. So I would recommend the authors to make a comparison and give some clarification.
3. The experiment results actually show that EF-SGD ( or EF21-P + DCGD) works better than DIANA and EC-SGD-DIANA (maybe EF21-P + DIANA), while DIANA and its variants are supposed to be newer algorithms with state-of-the-art performance in practice. So, the experiment results in this paper seem to contradict the results from the previous works. Now I wonder whether DIANA is really better than simple EF-SGD.

**Details Of Ethics Concerns:**

During the discussion stage, the authors have: 1) realized that EF21-P = EF which is not novel ("very shortly after we submitted our paper" as stated by the authors), but done nothing to fix this "minor blunder", not even during the 2 weeks of discussion period where revisions are allowed; 2) exaggerated the novelty of EF21-P + DCGD and EF21-P + DIANA without citing the previous works that proposed EF on server, while the authors clearly stated that "We do know of a very small number of papers where EF is applied on the server side". During the discussion period, the authors have every chance to fix the issue of EF21-P = EF, and add a citation of dist-EF-SGD. However, the authors made every effort to claim the novelty of EF21-P + DCGD and EF21-P + DIANA, and intentionally used the word "virtually" when referring to the previous works using EF on server, in order to minimize the credits given to the most related works. The very fact that the authors haven't made any revision shows that they either can not or will not fix the "minor blunder" as they claimed. Furthermore, exaggerating the novelty of the proposed algorithms without the citations of the essential previous works is clearly plagiarism, which raises serious concerns in ethics.

**Strength And Weaknesses:**

Strength:
1. This paper focuses on the theoretical analysis of EF21-P, which provides thorough theories for both convex and non-convex cases.
2. The empirical shows good performance.

Weakness:
1. For the empirical analysis, the model (logistic regression) and the datasets (libsvm datasets and cifar-10) are both too small and simple for distributed training. Typically, for such simple and small tasks, a single computer is enough for fast training.
2. I believe that the main algorithm proposed in this paper, EF21-P, is actually not novel and is equivalent to the original error feedback. See my discussion below for more details. However, I'm not 100% sure about this. So please make clarification and correct me if I'm wrong about this.

**Summary Of The Paper:**

This paper proposes a simple error-feedback mechanism called EF21-P, and some variants such as EF21-P + DCGD and EF21-P + DIANA. This paper focuses on the theoretical analysis of EF21-P. The empirical shows good performance.

**Summary Of The Review:**

Although this paper may provide some new insights in the theoretical analysis of distributed SGD with error feedback, the experiments are too simple, and more importantly there are some issues in the novelty of EF21-P.

---

> ### Author Response · Authors · 2022-11-09
> **Response to Reviewer m3XR (Part 1)**
>
> > This paper focuses on the theoretical analysis of EF21-P, which provides thorough theories for both convex and non-convex cases.
> The empirical shows good performance.
>
> Thanks for the positive evaluation of our work. **We wish to stress though that we not only provide thorough theory: we obtain new theoretical SOTA communication complexity rates for distributed optimization/training in the symmetric communication cost regime. Our complexity (of a new method we designed) is the first to decouple the variance coming from the uplink and downlink compressors from a multiplicative dependence to an additive dependence. We believe this is very significant, the key contribution of our work, and this was overlooked by the review.**
>
> > I believe that the main algorithm proposed in this paper, EF21-P, is actually not novel and is equivalent to the original error feedback (or EF, or error compensation). Here is some simple derivation...
>
> **Yes, you are right, and in fact, we found this our ourselves very shortly after we submitted our paper. We admit this was a bit embarrassing for us when we found out, especially since we know the EF field very well! However, this is a very minor issue, as we will explain. Because of this, we believe a score of 1 is wholly unjustified. Indeed, while score of 1 means "The contributions are neither significant nor novel", this is very far from being true. One particular claim we made early on in the paper is not true, yes, but this claim is easily fixable, has no bearing on the main results of our paper.**
>
> We will certainly mention that EF21 = EF21-P in the revised version of paper in a prominent way (and will change the title and the abstract as well)! However, this is easy to do, and is a minor issue as this can be handled by a few minor changes of the text.
>
> - **We still have theoretical SOTA for an important problem.** The issue is minor once you take into account the fact that this does not in any way diminish our main contribution: the new SOTA complexity for methods with bidirectional compression. Even though our methods contain a known method (EF/EF21-P) as a component, our methods are novel, and the theory we obtain is better and tighter. For example, we provide the first analysis of a method with bidirectional compression that guarantees communication complexity better than vanilla GD.
>
> - **We use EF/EF21-P in highly non-standard place - at the master.** What is also very important, however, is *how* we use EF/EF21-P to design a new SOTA method! Unlike the traditional approach, used virtually in all papers on error feedback, where EF is used on the client side and the master broadcasts uncompressed messages, we use EF/EF21-P on the server side. And instead of using EF/EF21-P or EF21 on the client side, we use DCGD or DIANA there. These choices are not arbitrary. We have an analysis of EF/EF21-P (used on the master) combined with EF21 on the client side, and the variances still remain in a multiplicative relationship. For this reason, we did not include this method nor theory in the paper. (We can include it here if you want to see it - just ask). Also, having EF21 on the master side does not seem to be helpful - Fatkhullin et al (https://arxiv.org/abs/2110.03294) tried this, and also suffer multiplicative dependence on the variances  (see their EF21-BC method - in their Table 1, the variance parameters $\alpha_W$ and $\alpha_M$ are multiplied).
>
> ---
>
> A historical comment:
>
> There is a large research field that analyzes the Asynchronous SGD - a method first proposed more than 20 years ago (see http://www.ifp.illinois.edu/~angelia/Parallel_Paper.pdf or https://papers.nips.cc/paper/2011/file/f0e52b27a7a5d6a1a87373dffa53dbe5-Paper.pdf). After decades, researchers continued to analyze this algorithm and its variants, further improving on the theoretical rates (see https://openreview.net/attachment?id=4_oCZgBIVI&name=supplementary_material). This later work was very important of course, and was in now way invalidated by the fact that the method was already proposed before. Indeed, the later work significantly advanced our knowledge about asynchronous methods. With bidirectional methods, we have a similar story: while EF = EF21-P was studied extensively before, we use it in a new way and obtain new SOTA rates!

---

> ### Author Response · Authors · 2022-11-09
> **Response to Reviewer m3XR (Part 2)**
>
> > EF21-P + DCGD is just EF-SGD using quantization as the compressor, which is not novel
>
> **No, it is not actually true.**
>
> 1. EF-SGD does not support bidirectional compression; it only supports uplink compression. However, our method EF21-P + DCGD does. So how can they be the same method? They are very clearly different.
>
> 2. The convergence rate of EF-SGD is worse than DCGD: $O(\frac{1}{\alpha} \kappa)$ vs $O(\frac{\omega}{n} \kappa).$ Note that DCGD improves with the number of workers $n$, while EF-SGD does not have that property.
>
> > EF21-P + DIANA is just EF + DIANA. There is already an algorithm called EC-SGD-DIANA (error compensation + DIANA, and error compensation is the same as error feedback) proposed in the following paper: Gorbunov, Eduard, Dmitry Kovalev, Dmitry Makarenko, and Peter Richtárik. "Linearly converging error compensated SGD." NeurIPS 2020. (https://proceedings.neurips.cc//paper/2020/file/ef9280fbc5317f17d480e4d4f61b3751-Paper.pdf) I'm not exactly sure whether EF + DIANA is equivalent to EC-SGD-DIANA, or maybe EF + DIANA is just another variant of DIANA that combines error feedback with DIANA. So, I would recommend the authors to make a comparison and give some clarification.
>
> These are somewhat relevant pointers, and we will include an explanation. However, **your claims here are also not correct either, and for the same reason.** EC-SGD-DIANA does not support downlink compression, and gets a rate that does not scale with the number of nodes $n.$ While indeed DIANA and EC-SGD are combined here as well, they are combined very differently, and mainly, their work addresses the asymmetric communication cost regime only. This can't be compared with our results - we address a very different problem: the symmetric communication cost regime.
>
> Perhaps the closest method to our method EF21-P + DIANA is Dore. However, the rates obtained in the Dore paper are much worse (see Table 1) than our rates.
>
> We wish to repeat a point we made before, since it is important and do not wish it to be missed: yes, we agree that EF21-P is indeed equivalent to EF. But the classical EF papers use it for uplink compression, and we realized it should really be used for downlink compression.
>
>
> ---
> **Summary**
> ---
>
> **We would be delighted if you could engage in a discussion with us as we believe our work was severely misjudged. Yes, we have committed a minor blunder by not realizing soon enough that EF = EF21-P, but we believe this minor blunder should certainly not be used as a reason to reject our paper. We believe we made a strong case explaining that our key contributions are not at all impacted by this - and this is what you have missed. We therefore kindly request that you re-evaluate our work in the light of this response and the subsequent discussion we are most happy to have with you.**
>
> **Thanks again for the effort you put into evaluating our work! We are certainly impressed that you noticed that EF = EF21-P. However, we are equally very surprised by how much you missed.**

---

> > ### Comment · Reviewer_m3XR · 2022-11-11
> > **I've increased the score of novelty and significance due to the theoretical analysis, but there is still almost no novelty in the algorithms**
> >
> > Thanks for the authors' feedback. Here is my additional comments:
> > 1. I don't really agree with the authors that EF21-P=EF is a "minor blunder". Apparently all the algorithms in this paper are based on EF21-P. After realizing that EF21-P is actually not novel, there will be a huge amount of revision for the writing of this paper by converting it to a paper of almost pure theoretical analysis. Because of that, I think this submission is simply not ready for publication at this time point.
> > 2. Thanks for the authors' clarification. I think the theoretical analysis in this submission may have some novelty, though I didn't actually have time to check the details of the proof. I've increased the score of "Technical Novelty And Significance" accordingly.
> > 3. Thanks for the authors' clarification. I agree that EF21-P + DCGD is different from the classic EF for uplink compression. Unfortunately, EF21-P + DCGD is still not novel. The authors claimed that using EF on the server side is novel. However, that is factually incorrect. Note that EF is studied extensively in the field of distributed machine learning. So, using EF on the server side is a very natural idea just like using EF21 on the server side, which has been proposed a long time ago. I believe the idea of using EF on the server side to achieve bidirectional communication compression was first proposed in this paper: Zheng, Shuai, Ziyue Huang, and James Kwok. "Communication-efficient distributed blockwise momentum SGD with error-feedback." NeurIPS 2019 (Algorithm 2, dist-EF-SGD). It is easy to check that the part of using EF on the server is equivalent to each other between EF21-P + DCGD and dist-EF-SGD when using a constant learning rate (the derivation is almost the same as I've done above, so I'm not going to write it down again). The readers may notice that dist-EF-SGD is using a variant of EF-SGD that puts the learning rate outside of the compressor. However, when using a constant learning rate as EF21-P + DCGD does, the position of the learning rate doesn't really matter. And it is trivial to put the learning rate back inside the compressor for dist-EF-SGD and change its convergence proof accordingly. Another difference between dist-EF-SGD and EF21-P + DCGD is that, dist-EF-SGD uses EF and biased compressors on the worker, while EF21-P + DCGD uses unbiased compressors without EF on the worker, which actually makes the theoretical analysis easier and simpler compared to dist-EF-SGD. Furthermore, since EF21-P + DIANA is just DIANA worker + EF server, its novelty is marginal because all these components are not novel, though the combination itself is novel.
> > The authors may think that it is just another "minor blunder" that EF on server side is not novel. However, again I have to disagree due to the heavy revision. Furthermore, dist-EF-SGD should be included as a baseline in the experiments (honestly, I think EF21-P + DCGD should be viewed as a special case of dist-EF-SGD by using a special unbiased compressor on the worker which no longer requires EF).
> > 4. Finally, I also agree with the other reviewers that the the experiment setup of this paper is too small for distributed training algorithms.
> >
> > In summary, unfortunately I'm not going to change the recommendation of rejection, mainly because both EF21-P and using EF on the server are not novel.

---

> > > ### Author Response · Authors · 2022-11-12
> > > **Response**
> > >
> > > 1.
> > >
> > > We understand your concern, but differ significantly in our estimate of how much revision is really needed. We believe a relatively minor revision would be sufficient. We will prepare it, and upload a new version of the paper. All we need to do is not to claim that EF21-P is a novel method, but that EF21-P can be seen as a new reformulation/reparameterization of EF which is useful on its own right. Virtually all text remains unchanged, but some paragraphs (and the abstract) indeed need to be redacted carefully. First, this way of looking at EF shows its very close connection to EF21, i.e., it's "dual" nature. This also enabled us to perform a strong analysis of EF21-P, which yields superior rates known analyses of EF. So, while the EF21-P method is not a new method per se, it provides a new point of view which is very fruitful. We get better rates, and we realize that EF should be employed on the master node, and not on the client notes, which is what virtually all EF papers do.
> > >
> > > 2.
> > >
> > > Thanks. We wish to point out again that our paper should best be judged by asking the following question: do the authors obtain new theoretical SOTA rates for methods with bidirectional compression (i.e., for communication-efficient distributed optimization in the asymmetric cost regime?) The answer to this question is YES - we offer the first method and analysis which decouples the multiplicative dependence of the compressor variances to an additive one. This is a very large contribution in our view, the key contribution of our paper, and remains, as we can see, completely overlooked. Yes, we do obtain this result by a careful combination of known tools and methods (e.g., EF, DCGD, DIANA), e.g., using the new insights that our EF21-P way of looking at EF provided to us. However, the way we assemble these tools is new, i.e., our final methods are indeed new. **Please note that even if they were not new at all, i.e., if we merely analyzed a known method, and still obtained new theoretical SOTA rates, especially with such a huge improvement as our theory offers, this should not be used as an argument to reject our paper in our view. The "novelty" you are seeking, we believe, is misplaced. Novelty can come in many forms. In our paper, the novelty is the new SOTA theoretical result. That is novel, and this was not disputed. Do we, as a community, care about theoretical SOTA rates? We should, just as we care about empirical SOTA results. Our methods are new, but less novel as they are composed of several known components. So, indeed, novelty on the algorithmic side is smaller than we originally though now that we are all in agreement that EF21-P = EF, after re-parameterization.**
> > >
> > > 3.
> > >
> > > We maintain that both EF+DCGD and EF+DIANA are novel methods. The method in the paper you point to does *not* reduce to DCGD when error feedback is removed from the master node. Indeed, the clients use EF instead of DCGD. This is a key difference, and our analysis is able capitalize on this. You can see this in Step 2 of their Algorithm 2: the workers do not merely compress the local (stochastic) gradients; they add error to the gradient information in a feedback loop - i.e., they apply EF. This happens even if constant stepsizes are used. So, their method is not EF+DCGD. Please also note that they consider the setup where the functions on all clients are the same; this is a very simple setup in which communication is not necessary at all to solve the problem.
> > >
> > > Re "The authors may think that it is just another "minor blunder" that EF on server side is not novel." No, we would not describe the situation this way. Please note that in our original response we said: "Unlike the traditional approach, used virtually in all papers on error feedback, where EF is used on the client side and the master broadcasts uncompressed messages, we use EF/EF21-P on the server side." We used the word "virtually", since indeed, in nearly all papers we have ever read on the EF topic, EF is applied on the worker side. We do know of a very small number of papers where EF is applied on the server side, but this fraction is very small. This is why we used the word "virtually". **Still, please note that what matters most is that whatever we do (even if we simply just re-analyzed a known method, which is not the case), we do obtain new SOTA rates. We would be happy if you could acknowledge this, and explain why you believe that obtaining new SOTA rates is no more enough of a contribution in your view. We believe this is the key to our paper.**
> > >
> > > 4.
> > >
> > > Our experiments are mainly designed to test our theoretical predictions, and we can do that without this requiring the problem sizes to be huge. We believe this should be seen as a plus. This is not an empirical piece of work, and standards for theory papers are different. Many theory papers are accepted without providing any experiments just like many experimental papers are accepted without containing any theory.

---

> > > > ### Comment · Reviewer_m3XR · 2022-11-19
> > > > **Thanks for the authors' reply**
> > > >
> > > > Thanks for the authors' reply.
> > > >
> > > > However, I need to clarify several points that I'm trying to make here:
> > > > 1. I do acknowledge that there is novelty in the theoretical analysis in this paper. Even if the main contribution of this paper is the SOTA theoretical rates of simple combination of existing algorithms, I do think that these contributions are meaningful and significant. However, as a reviewer, my duty is to judge the entire submission as a whole. Although the theoretical part is good, the fatal error in claiming EF21-P as a novel method, and the exaggeration of the novelty of EF21-P + DCGD and EF21-P + DIANA which are simple combinations of existing algorithms, cannot be  overlooked, because they are the main algorithms being studied in this paper. Thus, as a reviewer, I have to make tough choices and reject the **current version** of this paper, simply because it is not yet ready for publication. If this paper is rewritten into a purely theoretical paper, then I would tend to accept it. However, that would require heavy revision, as I mentioned before.
> > > > 2. The reason why I think the novelty of EF21-P + DCGD is extremely weak: As I mentioned before, it is trivial to use unbiased compressors on the workers for dist-EF-SGD, since it supports arbitrary compressors (the choice of blockwise compressor in the dist-EF-SGD is mainly driven by the concern of system overhead in real-world settings, if I understand it correctly). Then, obviously EF is not necessary for unbiased compressors, which is simply common sense. Thus, EF21-P + DCGD is almost just a special case of dist-EF-SGD with a special choice of compressors on the workers.
> > > > 3. I'm not exactly sure why the authors mentioned that "The method in the paper you point to does not reduce to DCGD when error feedback is removed from the master node" since obviously the main contribution of dist-EF-SGD is using EF on the master node and removing this part makes the entire dist-EF-SGD meaningless. Furthermore, if we follow the authors' logic here, then if we remove the compression methods from both dist-EF-SGD and EF21-P + DCGD, then they reduce to the same algorithm: EF on server, which turns out to support my point that the novelty of EF21-P + DCGD is limited.
> > > > 4. Even if we consider a version of dist-EF-SGD that uses both unbiased compressor and EF on the workers, it should be included in the experiments as a baseline, in order to verify the theoretical analysis that EF21-P + DCGD has a better rate than dist-EF-SGD, since the authors claimed that "Our experiments are mainly designed to test our theoretical predictions", and more importantly because dist-EF-SGD is the previous work closet to EF21-P + DCGD.
> > > > 5. I do think EF21-P + DIANA has some novelty, at least better than EF21-P + DCGD, since DIANA is much more complicated than DCGD and it would be non-trivial to combine it with EF on server and establish the corresponding theoretical analysis.
> > > > 6. If the revision is really that minor as claimed by the authors, the authors would have already done the revision and uploaded it. So far, the authors have: 1) realized that EF21-P = EF which is not novel ("very shortly after we submitted our paper" as stated by the authors), but done nothing to fix this "minor blunder", not even during the 2 weeks of discussion period where revisions are allowed; 2) exaggerated the novelty of EF21-P + DCGD and EF21-P + DIANA without citing the previous works that proposed EF on server, while the authors clearly stated that "We do know of a very small number of papers where EF is applied on the server side". During the discussion period, the authors have every chance to fix the issue of EF21-P = EF, and add a citation of dist-EF-SGD. However, the authors made every effort to claim the novelty of EF21-P + DCGD and EF21-P + DIANA, and intentionally used the word "virtually" when referring to the previous works using EF on server, in order to minimize the credits given to the most related works. The very fact that the authors haven't made any revision shows that they either can not or will not fix the "minor blunder" as they claimed. Furthermore, exaggerating the novelty of the proposed algorithms without the citations of the essential previous works is clearly plagiarism, which raises serious concerns in ethics.

---

> > > ### Author Response · Authors · 2022-11-12
> > > **Comment on Zheng, Shuai, Ziyue Huang, and James Kwok. "Communication-efficient distributed blockwise momentum SGD with error-feedback." NeurIPS 2019**
> > >
> > > We closely investigated "Communication-efficient distributed blockwise momentum SGD with error-feedback." NeurIPS 2019, and it is clear from Corollary 1 that Algorithm 2 of this paper requires at least $O(\frac{1}{\alpha^2\varepsilon^{3/2}})$ iterations. Assuming that the algorithm uses Top$K$ or Rand$K$ compressor, the communication complexity at least equals $K * O(\frac{1}{\alpha^2\varepsilon^{3/2}}) = K * O(\frac{d^2}{K^2 \varepsilon^{3/2}}) = O(\frac{d^2}{K\varepsilon^{3/2}}).$ Note that the communication complexity of the vanilla GD without compression equals $d * O(\frac{1}{\varepsilon}) = O(\frac{d}{\varepsilon}).$
> > >
> > > Now, we want to ask a simple question: ***Which method guarantees us better communication speed: GD or dist-EF-SGD?***
> > > From these bounds, it is clear that the GD method is a winner, GD method $\frac{d}{K\varepsilon^{1/2}}$ faster!
> > >
> > > Similarly, we also can ask: ***Which method guarantees us better communication speed in the convex setting: GD or EF21-P + DIANA?***
> > > In Section 3, we prove that EF21-P + DIANA is a winner! This is the first method that gives such an optimistic answer!
> > >
> > > "Communication-efficient distributed blockwise momentum SGD with error-feedback." is a good paper, but from the view of theoretical bounds, the vanilla GD (or DIANA) is a better candidate to be a part of experiments.

---

### Official Review · Reviewer_HQee · 2022-10-25

**Confidence:** 3
**Correctness:** 2
**Technical Novelty And Significance:** 2
**Empirical Novelty And Significance:** 2
**Recommendation:** 1

**Clarity, Quality, Novelty And Reproducibility:**

The paper is well-written and well-organized, and the method is new.
Codes are provided, so it should not be difficult to reproduce the experimental results.

[update] during discussion, reviewers find that EF=EF21-P, which is also acknowledged by the authors and severely damages the novelty. Therefore I decided to lower all my scores.

**Strength And Weaknesses:**

Strength:
1. The proposed EF21-P is an interesting way to look at error feedback. The compression error seem to be modeled as the difference between $x$ and $w$, so the optimization objective seem to be make $x$ and $w$ more coherent, which is novel.
2. The theoretical results are comprehensive, including both the convex and non-convex cases, and the combination with DIANA/DCGD to achieve the best results.

Weaknesses:
1. The experimental settings are small-scale. #features is 20k and #samples is 72k. A single worker should be sufficient to train logistic regression on this dataset.

**Summary Of The Paper:**

The authors propose EF21-P in eq(3), which is an interesting and novel variant of error feedback. Theoretical analysis is comprehensive and solid. Empirical results on logistic regression task show the benefits of EF21-P.

**Summary Of The Review:**

The author proposes a novel variant of error feedback, and it seems very interesting compared with most of the existing error feedback variants. The theoretical analysis looks comprehensive and solid. The only concern is how this method works in practice with larger-scale datasets and more complicated models.

---

> ### Author Response · Authors · 2022-11-09
> **Response to Reviewer HQee**
>
> Thank you for the very positive review!!
>
> > The experimental settings are small-scale. #features is 20k and #samples is 72k. A single worker should be sufficient to train logistic regression on this dataset.
>
> Our goal was to obtain new theoretical results for bidirectional compression. The experiments are provided merely to support them; and we believe the size of the problems does not matter for the purposes of our tests. That is to say, the questions we aimed to ask and answer could be answered using the problem sizes we consider. Our experiments were mainly aimed to test the predictive power of our theory.
> Further large scale experiments are of course possible to do, but our work is not of an empirical nature, and such work would really require a dedicated paper.

---

> > ### Comment · Reviewer_HQee · 2022-11-10
> > **novelty**
> >
> > From reviewer m3xr's comment, the proposed EF21-P is identical to EF. There is no novelty from the algorithm side. Therefore, I will lower my score.
> >
> > For the experiment, I do not agree that this small dataset can empirically verify your conclusion.

---

> > > ### Author Response · Authors · 2022-11-10
> > > **Response**
> > >
> > > What you say is factually incorrect. Of course our methods are novel!
> > >
> > > Indeed, several building blocks in our method is known: EF = EF21-P, which we did not notice before submission, and DCGD and DIANA are also known methods. But the key methods in our paper -- the methods that use bidirectional compression (EF21-P + DCGD and EF21-P + DIANA) -- are new as such, and most importantly, they obtain new theoretical SOTA!
> > >
> > > Even though EF=EF21-P, our point of view at EF is different, and our theory for EF21-P is superior to EF theory.
> > >
> > > Also, we use EF in a new way: on the master rather than on the clients - this is very nonstandard. The EF literature virtually exclusively uses EF on the  the clients to compress uplink communication. We realized this was not the way to go; and employ DCGD and DIANA on the clients instead.

---

> > > > ### Comment · Reviewer_HQee · 2022-11-12
> > > > **EF on master is not novel**
> > > >
> > > > Please note that EF on master and client sides are not novel. For example, check [1]. A lot of follow-up works incorporate compression on both sides. If the authors are familiar with the related works in EF, they should not be making such statements, hoping that the reviewer might not be familiar with this topic.
> > > >
> > > > As the author admits that EF=EF21-P, it becomes a fatal drawback regarding the novelty. It's the authors' responsibility to avoid this before submission. If the paper is more about improving the theoretical results of EF, then it needs a rewrite.
> > > >
> > > > [1] Zheng, Shuai, Ziyue Huang, and James Kwok. "Communication-efficient distributed blockwise momentum SGD with error-feedback." Advances in Neural Information Processing Systems 32 (2019).

---

> > > > > ### Author Response · Authors · 2022-11-12
> > > > > **Re: EF on master is not novel**
> > > > >
> > > > > > Please note that EF on master and client sides are not novel. For example, check [1]. A lot of follow-up works incorporate compression on both sides. If the authors are familiar with the related works in EF, they should not be making such statements, hoping that the reviewer might not be familiar with this topic.
> > > > >
> > > > > **We did not say that there are 0 papers which apply EF on the master. We said that **virtually all** literature on EF focuses on applying EF on the client side. We stand by this. We never meant to deceive, and we always admit when a valid point is made. In any case, while the components that make up our methods are known, our way of combining them into the final algorithm is novel. The reviewer did not point out any prior work in which either EF+DCGD nor EF+DIANA was proposed. Note that our method explicitly does not use EF on the clients, and it uses unbiased compressors there. This is important in our theory and guarantees. Now, and this is the point the reviewer does not appreciate in our view: even if we merely analyzed an existing method, what matters are the theoretical SOTA rates we establish. This is what matters the most, by far. A novel method with bad theoretical guarantees is surely much much less interesting than an existing method with better, let alone SOTA theoretical guarantees.**
> > > > >
> > > > > > As the author admits that EF=EF21-P, it becomes a fatal drawback regarding the novelty. It's the authors' responsibility to avoid this before submission.
> > > > >
> > > > > We agree it's an issue (we clearly explained it), but we differ in the judgement of how severe it is. It our view, it is a minor issue. Certainly not fatal. Why do we claim this? Because it does not in any way invalidate any of the theory, and hence the main results and conclusions of the paper persist: we obtain new SOTA rates for bidirectional compression. Can you acknowledge this? If you do not believe this is the case, can you explain why? This is the heart of the paper, and we do not seem to have a discussion about this at all.
> > > > >
> > > > > The issue which you portray as fatal is minor as a minor change to the paper wording in strategic places makes it all OK. We just need to state that EF21-P is a new way of looking at EF, instead of claiming it's a new method. That is all. It's still the case that this novel point of view enables us to construct a new SOTA method in terms of theoretical communication complexity. The re-parameterization enables us use different proof techniques than those used to analyze EF. And our choice of DCGD and DIANA as the client compression methods is also key to the success of pur approach and the strength of our bounds.
> > > > >
> > > > > > If the paper is more about improving the theoretical results of EF, then it needs a rewrite.
> > > > >
> > > > > **The paper is not mainly about improving the theoretical results of EF. We never claimed this. The paper is about obtaining new SOTA theoretical communication complexity for the asymmetric communication cost setup. We do succeed here, and the reviewer still seems to ignore this, despite the fact that we pointed this out several times already.** What we said is that en route to establishing these results, it is important for us to view EF as EF21-P as this enables us to use a different proof technique from that used to analyze classical EF. The proof technique resembles that of EF21 rather than EF - we realized this once we saw that EF21-P is in some sense the "primal" variant of EF21 (these are different methods). Yes, some minor rewriting is needed. But it is minor.
> > > > >
> > > > > ---
> > > > >
> > > > > **Thanks for engaging with us! We appreciate it even though we do not seem to agree. Thanks for the time you are putting into this!**

---

> > > > > ### Author Response · Authors · 2022-11-12
> > > > > **Comment to Reviewer m3XR about "Communication-efficient distributed blockwise momentum SGD with error-feedback"**
> > > > >
> > > > > Reviewer HQee, please take a look at our comment "Comment on Zheng, Shuai, Ziyue Huang, and James Kwok. "Communication-efficient distributed blockwise momentum SGD with error-feedback." NeurIPS 2019 " to Reviewer m3XR

---

### Official Review · Reviewer_7QTA · 2022-10-26

**Confidence:** 3
**Correctness:** 4
**Technical Novelty And Significance:** 2
**Empirical Novelty And Significance:** 2
**Recommendation:** 5

**Clarity, Quality, Novelty And Reproducibility:**

Paper is easy to follow.


**Strength And Weaknesses:**

Strength:

I think the most interesting part of this paper is the proposed compression technique and the section “Unified SGD analysis framework with EF21-P mechanism” which is placed in appendix.

Weakness:

While there is theoretical value in bidirectional distributed SGD, I  do not think there is much practical gain in using bidirectional compression as there is a number of references that claim that Broadcasting one message to n users is much cheaper than n poin-to-point communication. Therefore, my main criticism is the motivation for bidirectional compression.

Another question is how do you compare the additional storage cost of distributed algorithm?

Additionally, I do not think the Assumption on $\hat{L}$ is standard. Can author provide any justification for that?


**Summary Of The Paper:**

This paper suggests a new compression method based on the recent EF21 scheme and then used this compression method in the bidirectional distributed SGD method. Authors provide convergence rates for strongly convex, convex, and non-convex objectives. The theory is verified using logistic regression experiments.


**Summary Of The Review:**

Please see the comments above!

---

> ### Author Response · Authors · 2022-11-09
> **Reply to Reviewer 7QTA**
>
> > While there is theoretical value in bidirectional distributed SGD, I do not think there is much practical gain in using bidirectional compression as there is a number of references that claim that Broadcasting one message to n users is much cheaper than n poin-to-point communication. Therefore, my main criticism is the motivation for bidirectional compression.
>
> **We are convinced that that this is invalid criticism. Please read our response to Reviewer Vk1R (Part 1) where we address the same concern.** We hope our response there settles this question. We are happy to elaborate -- please do ask if anything remains unclear.
>
> > Another question is how do you compare the additional storage cost of distributed algorithm?
>
> **The additional storage cost we require is relatively small, and does not pose any real problem.** Indeed, compared to the DIANA method, where the server holds one state vector, our method requires the server to hold two state vectors in total. We believe that it is not a significant drawback because we preserved $O(d)$ memory complexity. We explain this in more detail in our response to Reviewer Vk1R (Part 2); please refer to that.
>
> > Additionally, I do not think the Assumption on $\hat{L}$ is standard. Can author provide any justification for that?
>
> Note that this assumption automatically holds, i.e., $\widehat{L}$ is finite, under the very standard assumption requiring all the functions $f_i$ to be $L_i$-smooth. In particular, $\widehat{L} \leq \max_i L_i = L_{\max}$. We prove this in Lemma 2.4. So, this is not really an additional assumption in the sense that it would limit the scope of our theory. The reason why we work with $\widehat{L}$ is that it naturally appears in our analysis, and it can be significantly smaller than the naive upper bound $max_i L_i$. So, our method provably benefits when the quantity $\widehat{L}$ is small, and we hence want to make the dependence on this quantity explicit.
>
> We also need to point out that while indeed the quantity $\widehat{L}$ is not in widespread use, this is because it does not naturally appear in the context of many algorithms. However, it does naturally appear in some methods, such as in (https://openreview.net/pdf?id=GugZ5DzzAu). We work with this extra smoothness constant because it allows us to prove that EF21-P + DIANA gets better rates than vanilla GD (see Section 3), without actually assuming more than $L_i$-smoothness of each $f_i$. That is, it allows us to get better complexity for free in cases when this quantity is small.
>
> **In summary, the fact that we are able to work with $\widehat{L}$ is an advantage of our analysis, not a weakness.**

---

### Official Review · Reviewer_Vk1R · 2022-10-31

**Confidence:** 3
**Correctness:** 4
**Technical Novelty And Significance:** 3
**Empirical Novelty And Significance:** Not applicable
**Recommendation:** 5

**Clarity, Quality, Novelty And Reproducibility:**

The overall idea is novel and clearly presented but the writing is very heavy on notation. I would suggest adding a table of symbols and their meanings in the appendix to help readers keep track. It might also be beneficial to provide a brief primer on gradient compression with error-feedback to enable to paper to reach a broader audience. Specifically, the steps in Algorithm 1 may not be self-explantaory to readers unfamiliar with the intuition behind error feedback.

**Strength And Weaknesses:**

Strengths:

1. To the best of my understanding, this is a novel application of error feedback for server-to-worker compression in distributed machine learning. The fact that it can be used to augment any unbiased worker-to-server compression method also broadens its scope.

2. The theoretical and empirical results presented show improvements over prior work.

Weaknesses:

1. My main concern with the paper is that the overall gain with this approach is not entirely clear. As the authors themselves acknowledge, the received wisdom in this space is that downloading data is faster/cheaper than uploading it which reduces the need for compressing data being sent by the server. Moreover, the proposed approach appears to require the workers to a) maintain local state/memory and b) use an unbiased compressor. While the requirement of maintaining local state is a bit restrictive, that is not a big issue since there are scenarios where such state can be maintained. However, as far as I know, if a worker maintains state, it can use a biased compressor with error feedback to improve significantly over unbiased compressors. As the authors don't compare against any such method, it is unclear if their approach leads to any significant improvement over these approaches, especially when the need for server-side compression may not be that acute to begin with.

2. A second minor concern is the limited empirical evaluation. The authors only consider logistic regression on a few datasets. I would like to see evaluation on some other machine learning models to see if the gains are as apparent in other models as well. Also please clarify why results with the baseline approaches (DIANA, MCM, DASHA) are not presented for all datasets (some have DIANA as the baseline, some have DASHA, some have MCM).

**Summary Of The Paper:**

The paper proposes a new error-feedback based gradient compression algorithm for distributed machine learning. The key difference with prior work is that the compression is bidirectional i.e. data sent by both the workers and the server is compressed unlike most prior works which only focus on compressing data sent by the workers. The authors show that a novel application of error feedback at both the server and the workers enables them to improve over the communication complexity of several prior methods and converge at the same rate as gradient descent despite sending significantly lesser data in each direction at each communication round.

**Summary Of The Review:**

The paper proposes a new application of error feedback to bidirectional gradient compression. The application is novel and the theoretical results show improvement over prior work. However the evaluation is somewhat limited and the overall gains of the approach for different machine learning models and as compared to methods using error feedback and biased gradient compression at workers aren't clear yet.

---

> ### Author Response · Authors · 2022-11-09
> **Response to Reviewer Vk1R (Part 1)**
>
> > My main concern with the paper is that the overall gain with this approach is not entirely clear. As the authors themselves acknowledge, the received wisdom in this space is that downloading data is faster/cheaper than uploading it which reduces the need for compressing data being sent by the server.
>
> There are two communication regimes studied in the literature, and both are well motivated: A) **asymmetric** and B) **symmetric**.
>
> A) **Asymmetric communication.** The cost of broadcast is assumed to be so fast, that it is neglected (in the theoretical analysis, but not always in the practical implementation of the algorithms). Admittedly, this scenario is considered in the vast majority of papers that consider communication compression. The theoretical communication complexity rates in these works do not account for the bits transferred from the server to the workers and hence essentially assume that the cost of downlink communication is zero. Clearly, this regime is a useful abstraction to model the situation where the gap between the uplink and downlink speeds is very large, and makes sense to a certain point only: to the point where the uplink compression level is so high that the uplink speed per communication round matches the fast downlink speed per communication round. Beyond this compression level, it would be inaccurate to not account for downlink communication. Such a breaking point can, however, be reached very quickly. For example, if the downlink speed is 10 times faster than the uplink speed, then applying the Rand-K compressor with $K=0.1d$ to compress the uplink messages equals the cost of compressed uplink and uncompressed downlink communication. However, in modern ML practice, $d$ can be very large, so large that it may be desirable to compress far beyond the $10:1$ ratio obtained this way. beyond this point, further progress can only be made by compressing the downlink communication as well.
>
> For these reasons, we believe it is not particularly helpful to characterize the (simplifying and certainly useful, to a certain degree) theoretical assumptions in the asymmetric communication regime literature as "received wisdom". This would be unscientific. Instead, we argue that it is important to reexamine the underlying assumptions, find the point where they are no longer useful, and propose a remedy. And they certainly break, as we explained above.
>
> B) **Symmetric communication.** In this paper we consider the much more algorithmically and theoretically challenging symmetric regime, in which the cost of downlink communication is not assumed to be negligible. Please note that symmetry here does not necessarily mean that the cost of uplink is the same as the cost of downlink, even though this is the simplest way of thinking about this regime. This symmetric regime is largely overlooked by the community, mainly due to the fact that practical compute systems such as supercomputers and clusters are to a certain degree asymmetric, and also because the design and analysis of algorithms which have the luxury to ignore the downlink speed is substantially easier. Indeed, as we have explained in the paper, **existing methods which support bidirectional compression suffer from a multiplicative dependence on the variances that come from the uplink and downlink compressors**, and this makes these methods less effective and interesting from a practical and theoretical point of view. Indeed, such methods can have worse communication complexity than methods which do not compress messages at all. **In our work, we propose the first methods that break this multiplicative dependence, and replace it by an additive dependence.** We believe this is a major advance in this field, and we also believe this will inspire many others to propose further improvements.

---

> ### Author Response · Authors · 2022-11-09
> **Response to Reviewer Vk1R (Part 2)**
>
> > Moreover, the proposed approach appears to require the workers to a) maintain local state/memory and b) use an unbiased compressor. While the requirement of maintaining local state is a bit restrictive, that is not a big issue since there are scenarios where such state can be maintained. However, as far as I know, if a worker maintains state, it can use a biased compressor with error feedback to improve significantly over unbiased compressors. As the authors don't compare against any such method, it is unclear if their approach leads to any significant improvement over these approaches, especially when the need for server-side compression may not be that acute to begin with.
>
> **Please note that the extra state we hold does not change the amount of storage significantly; just by a small constant factor.** First, in the situation where the server does not compress at all, it would still require $O(d)$ memory to aggregate the $d$-dimensional vectors it receives from the nodes. Moreover, in the original DIANA algorithm, the server holds another $d$-dimensional state vector, $h^t$. In our approach, we use an additional state $p^t$ that has size $d,$ so the memory complexity is still $O(d)$. So, we hold two state vectors instead of one state vector.
>
> **This is not true. The best methods specialized to unbiased compressors have much better theoretical convergence rates than the best methods that work with biased (i.e., contractive) compressors.** This is natural since any unbiased compressor turns into a biased (contractive) compressor after appropriate scaling, and hence the class of contractive compressors is broader. Clearly, it is not possible to prove better rates for a broader class of compressors, other things equal. For example, let us consider the result from https://arxiv.org/pdf/2010.12292.pdf (see Table 1): In the strongly convex regime, the methods with biased compressors and error feedback have complexity $O(\frac{1}{\alpha} \kappa)$, while DIANA has the convergence rate $O(\frac{\omega}{n} \kappa)$. In order to understand the difference, let us fix Rand-$K$ as the compressor. For this compressor, we know that $1/\alpha = \omega = d/K.$ This means that DIANA can be $n$ times faster!
>
> The reviewer may perhaps ask: Why didn’t we try to combine EF and EF21-P and get a new method that can work biased compressors? The answer is that we don’t know how to do that (or whether it is even possible to do that) in a way to get better complexities than with the approach we actually chose. For instance, the authors of the paper (https://arxiv.org/abs/2110.03294) tried to do that, but they got bad dependencies on the variances of compressors (see also our Table 2).
>
> We can add EF to Table 1, but it will get the complexity worse than DIANA. That is the main reason why we didn’t add it.
>
> > A second minor concern is the limited empirical evaluation. The authors only consider logistic regression on a few datasets. I would like to see evaluation on some other machine learning models to see if the gains are as apparent in other models as well. Also please clarify why results with the baseline approaches (DIANA, MCM, DASHA) are not presented for all datasets (some have DIANA as the baseline, some have DASHA, some have MCM).
>
> **Our paper is mainly a theoretical work: we obtain theoretical SOTA communication complexity for distributed training in the symmetric communication cost regime.** So, we agree with you that this is a minor point. We can add some more experiments on some other machine learning tasks, but it is not clear what the community might get from such experiments. We believe that our theory and Tables 1 and 2 are convincing on their own.
>
> > The overall idea is novel and clearly presented but the writing is very heavy on notation. I would suggest adding a table of symbols and their meanings in the appendix to help readers keep track. It might also be beneficial to provide a brief primer on gradient compression with error-feedback to enable to paper to reach a broader audience. Specifically, the steps in Algorithm 1 may not be self-explantaory to readers unfamiliar with the intuition behind error feedback.
>
> Good suggestions, we will apply them.

---

### Decision · Program_Chairs · 2023-01-20

**Decision:**

Reject

**Justification For Why Not Higher Score:**

The paper received significant criticisms regarding novelty, which required a big overhaul of the presentation of current results and algorithms, especially in comparison to current literature.

**Justification For Why Not Lower Score:**

N/A

**Metareview: Summary, Strengths And Weaknesses:**

This paper presents a simple error-feedback mechanism, referred to as EF21-P, to address errors introduced by a contractive compressor. EF21-P operates in the primal space of models, and is used as a key building block for communication-efficient distributed optimization methods. The authors provide theoretical bounds on communication complexity for convex and nonconvex problems, and the results are corroborated by experiments.

Summarized Strengths:
-  Unified SGD analysis framework with EF21-P mechanism
- Comprehensive theoretical results including convex and non-convex cases
- Results easily reproducible as codes are provided
- Theoretical and empirical results show improvements over prior work
- Can be used to augment any unbiased worker-to-server compression method

Summarized Weaknesses:
- Main algorithm proposed in paper is not novel and is equivalent to original error feedback (EF) method. The authors allege that they were not aware of that during the submission of the paper, yet at the same time explain how they are extremely well versed in the EF scheme. I personally give benefit of a doubt to the authors, but at the very least the discussion during the rebuttal phase would demand that the paper is significantly rewritten to address this (arguably major) issue.
- Overall gain is unclear due to need for workers to maintain local state and use an unbiased compressor
- Limited empirical evaluation on only logistic regression with few datasets
- Little practical gain from bidirectional compression
- Some nonstandard assumptions (eg on \hat{L})

There was extensive engagement with the most critical of the reviewers, during which the authors realized that one of the claims (eg novelty of proposes EF21-P scheme) is not present, as it is equivalent with another method (EF). This of course can happen, and is not a major sign of concern, especially in the presence of novelty at other parts of the paper. However, given the present way that the results, and contributions are presented, the authors would have to significantly overhaul their manuscript to a point that would significantly deviate from the current version, that would require a complete and separate review phase.

The authors should take thorough steps to address the significant concerns of novelty raised by the reviewers, and if wish to resubmit the current paper to a future venue explain in detail how EF21-P is not a novel algorithm, and the main points of novelty are the theoretical results, and experimental findings, that many of the reviewers are in agreement that are novel.


**Summary Of Ac-Reviewer Meeting:**

N/A